# Learning Task-Sufficient World Models via Intervention-Curriculum Co-Design

## Abstract

We study how agents learn world models with latent representations that are *task-specific, minimal, and sufficient* for sequential decision making. Rather than predicting pixels or relying on generic embeddings, we aim to learn representations that retain exactly the information needed for control across tasks. We model the problem end-to-end as a closed loop of agent–environment interaction, enabling the agent to sequentially acquire minimal and sufficient latent representations over a series of tasks. On the agent side, for each new task, it begins with *active intervention* to acquire informative trajectories that implicitly reveal task-relevant latent factors, and then trains the world model to learn a latent space that is both minimal and task-sufficient. On the environment side, learning is facilitated through an *adaptive curriculum* that co-evolves with the agent. By tailoring environment settings and task order to the agent's learning progress, the curriculum exposes control-relevant mechanisms at the right level of difficulty, while jointly scheduling world-model updates with policy learning. This co-design of intervention and curriculum leads to a compact, structured latent space that supports efficient, transferable policy learning and generalization. Empirically, our approach improves sample efficiency and generalization across skills, object–skill compositions, and unseen tasks on standard continuous control and robotic manipulation benchmarks.

## 1 Introduction

World models (Ha & Schmidhuber, 2018) are generative or predictive models that capture environment observation functions, dynamics, and rewards; in model-based RL (MBRL), it benefits planning, policy improvement, and simulated rollouts to reduce sample complexity (Sutton, 1991). One way of learning the world model is to compress raw observations into meaningful latent states that support accurate prediction and control, facilitating reuse across tasks and improving generalization, e.g., Dreamer-series (Hafner et al., 2020; 2021; 2025), Efficient-Zero (Schrittwieser et al., 2020; Ye et al., 2021), and TD-MPC (Hansen et al., 2022; 2024). Recent work also leverages pre-trained foundation models as strong perceptual encoders: using embeddings as the observation space, and the agent then learns dynamics on top of these embeddings via self-predictive manners (Zhou et al., 2025; Baldassarre et al., 2025; Kapoor et al., 2025; Wang et al., 2025; Assran et al., 2025).

These models often excel at learning visual dynamics and supporting downstream control. Yet their latent representations frequently entangle perceptual factors unrelated to control, which can yield brittle policies in complex or changing environments (Wang et al., 2022a; Liu et al., 2023). By contrast, humans typically do not plan using pixel-level predictions or by tracking redundant perceptual details; instead, our planning relies on compact task-relevant representations (Mastrogiuseppe & Ostojic, 2018; Ho et al., 2022; Rajalingham et al., 2022; Nayebi et al., 2023). Motivated by this, we argue that learning world models with *task-specific, minimal sufficient* latents, derived from raw observations or foundation-model priors, is essential for efficient and generalizable policy learning, as such representations preserve exactly the information needed for decision-making.

In pursuit of this minimal and sufficient representation in the world model, we resort to the analogy of how humans build a model of the world through their interactions with the *environment*: a child probes the world, and subconsciously chooses useful experiences, and gradually masters useful concepts (Schulz et al., 2008; Bonawitz et al., 2011). Similarly for agents: we can (i) shape the *environment* via a suitable curriculum over settings and task order to expose control-relevant mechanisms progressively, and (ii) equip the *agent* to probe the environment actively: treating the previously learned world model as a prior, the agent designs and executes interventions to identify

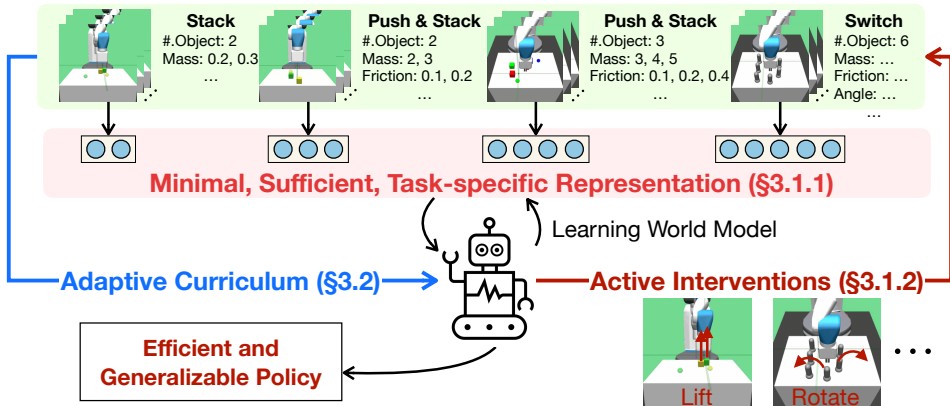

Figure 1: **Overview on `MIST-WM`**: co-design on the agent and environment. An adaptive curriculum (tasks, environment parameters) will be learned to provide an optimal order for the agent to learn the world model and policy. And the agent conducts active interventions to identify the key latent factors in each new environment in a self-supervised way. On top of that, a world model with minimal, sufficient, and task-specific representation will be learned.

task-relevant latent representation and collect informative trajectories. This closed loop yields representations that are sufficient for control yet minimal, enabling efficient policy learning.

Building on this, we propose `MIST-WM` (Minimal, Sufficient, and Task-specific World Model), a framework that co-evolves the environment and the agent (Fig. 1). The agent learns world models sequentially across tasks, using each previously learned model as a prior for the next, yielding a latent space that becomes progressively expandable and identifiable. For each task, `MIST-WM` estimates an observation model, latent dynamics, and a reward model. Crucially, on encountering a new task, the agent actively designs and executes *interventions* via self-supervised skills to collect trajectories that maximize expected information gain about task-specific latents. The resulting interaction data will then be used to update the world model under objectives that learn structured and compact latents through minimality and sufficiency objectives.

On the environment side, we provide an adaptive curriculum for the agent that co-schedules world-model learning and downstream decision making under a unified objective, coupling the world model and the policy. Inspired by unsupervised environment design (UED) (Dennis et al., 2020; Rigter et al., 2024; Teoh et al., 2024), we optimize over environment settings, i.e., tasks and scene parameters (e.g., object properties, distractors), to reveal control-relevant mechanisms at the right order and difficulty. Together, these components enable `MIST-WM` to (i) learn task-specific latent representations that are minimal and sufficient across a sequence of tasks, and (ii) adapt quickly to novel tasks by leveraging and composing previously learned representations.

Overall, `MIST-WM` offers an end-to-end recipe for what data to collect, how to learn the world model, in what order to encounter tasks, and how to generalize across tasks via compositional reuse of learned representation. We evaluate on locomotion and robotic control benchmarks. `MIST-WM` learns task-specific representations that are both minimal and sufficient, aligning with ground-truth system variables. With these compact representations, it achieves sample-efficient policy learning and robust generalization to novel settings, at the skill level, in object–skill compositions, and on unseen tasks.

## 2 PRELIMINARIES

`MIST-WM` models the environment as a factored POMDP with an *expandable* latent state across tasks. We first formalize the generative process by adapting the Factored MDP framework of Boutilier et al. (2000) to *time-varying* structures and an expandable state space. We then define the *Minimal, Sufficient, Task-specific* (MIST) states and outline the computational flow for identifying and using MIST within the world model.

### 2.1 FACTORED POMDP WITH CHANGING STRUCTURE AND EXPANDABLE STATES

We model the system as a factored POMDP, which can be represented as a tuple $\mathcal{M} = \left(\mathcal{S}, \mathcal{A}, \mathcal{O}, \gamma, \mathcal{G}, \mathbb{P}_s, \mathbb{P}_o, \mathcal{R}, \rho_0\right)$, where $\mathcal{S}$ is the (latent) state space, $\mathcal{A}$ the action space, $\mathcal{O}$ the ob-

Figure 2: (a) Example DBNs that describe the generative process of two tasks. The grey and white nodes indicate observable and latent variables. The blue nodes are the newly expanded variables for Task 2. Red edges are the links from state variables to rewards. For simplicity, we omit the factorization of the state–observation mapping $\mathbf{s} \to \mathbf{o}$. (b) Computational flow of our proposed framework compared with the Dreamer-based world model.

servation space, $\gamma \in (0, 1)$ the discount, and $\rho_0$ an initial-state distribution. $\mathcal{G}$ is a Dynamic Bayesian Network (DBN) (Murphy et al., 2002) over state factors $\mathbf{s}_t = (\mathbf{s}_t^1, \ldots, \mathbf{s}_t^d)$, action $\mathbf{a}_t = (\mathbf{a}_t^1, \ldots, \mathbf{a}_t^m)$ and reward $r_t$. Let $\mathrm{pa}(X)$ denote the set of parents of node $X$ in $\mathcal{G}$. Hence, the state transitions are $\mathbb{P}_s(\mathbf{s}_t \mid \mathbf{s}_{t-1}, \mathbf{a}_{t-1}) = \prod_{i=1}^{d} \mathbb{P}_s(\mathbf{s}_t^i \mid \mathrm{pa}(S_t^i))$, where $\mathrm{pa}(S_t^i) \subseteq \{\mathbf{s}_{t-1}^1 : \mathbf{s}_{t-1}^d, \mathbf{a}_{t-1}^1 : \mathbf{a}_{t-1}^m\}$. The task-specific reward is a function of the parents of $R_t$: $\mathcal{R}(s_t, a_t) = \mathbb{E}[R_t \mid \mathrm{pa}(R_t)]$, where $\mathrm{pa}(R_t) \subseteq \{\mathbf{s}_t^1 : \mathbf{s}_t^d, \mathbf{a}_t^1 : \mathbf{a}_t^m\}$. The observation function is modeled as $\mathbb{P}_{\mathbf{o}}(\mathbf{o}_t \mid \mathbf{s}_t) = \prod_{j=1}^{p} \mathbb{P}_o(\mathbf{o}_t^j \mid \mathrm{pa}(\mathbf{o}_t^j))$, where $\mathrm{pa}(\mathbf{o}_t^j) \subseteq \{\mathbf{s}_t^1 : \mathbf{s}_t^d\}$.

Then we consider a sequence of tasks $\{T_i\}_{i=1}^N$, where the latent state space is allowed to grow over time. Specifically, when the agent transitions to a new task, new state variables may be introduced that were absent in previous tasks. Thus, in task $T_i$, the latent state is $\mathbf{s}_t^i = (\mathbf{s}_t^{i,1}, \ldots, \mathbf{s}_t^{i,d_i})$, where $d_1 \leq \cdots \leq d_N$, allowing the state dimensionality to *expand across tasks*. Each task is equipped with a task-specific DBN $\mathcal{G}^{(i)}$ over $(\mathbf{s}_t^i, \mathbf{a}_t, \mathbf{o}_t, r_t^i)$. The corresponding state transition $\mathbb{P}_s^{(i)}$, observation function $\mathbb{P}_o^{(i)}$, and reward function $\mathcal{R}^{(i)}$ are also task-specific, yielding the task-specific DBNs (Bilmes, 2000). Fig. 2(a) illustrates two tasks with different state–transition ($\mathbf{s}_t \to \mathbf{s}_{t+1}$) and state–reward ($\mathbf{s}_t \to r_t$). In addition, task 2 introduces an additional state factor $\mathbf{s}^{2,3}$.

## 2.2 Minimal, Sufficient, and Task-Specific (MIST) States

Once the factored structure is obtained, the MIST states can be directly identified from it. Clearly, we can consider MIST states as the state parents of the variables that correspond to the decision-making objectives (e.g., rewards). Two kinds of state variables are included in this set: (i) *current-time* factors that directly affect the reward, and (ii) *previous-time* factors that affect the reward *indirectly* via their children at time $t$. Formally, we have:

**Definition 1** (MIST indices for task $T_i$). *Let $T_i$ have factored state $\mathbf{s}_t^i = (\mathbf{s}_t^{i,1}, \ldots, \mathbf{s}_t^{i,d_i})$ and DBN $\mathcal{G}^{(i)}$ with parent operator $\mathrm{pa}_{\mathcal{G}^{(i)}}(\cdot)$. Define $I_{i,1}^{(t)} := \{ k \in [d_i] \mid \mathbf{s}_t^{i,k} \in \mathrm{pa}_{\mathcal{G}^{(i)}}(r_t) \}$, $I_{i,2}^{(t)} := \{ k \in [d_i] \mid \exists j \in I_{i,1}^{(t)} \text{ s.t. } \mathbf{s}_{t-1}^{i,k} \in \mathrm{pa}_{\mathcal{G}^{(i)}}(\mathbf{s}_t^{i,j}) \}$. Let $U_i := I_{i,1}^{(t)} \cup I_{i,2}^{(t)}$.*

**Definition 2** (Sufficiency). *The index set $U_i$ is* sufficient *for $T_i$ if $r_t^i \perp\!\!\!\perp (\mathbf{s}_{t-1}^i, \mathbf{s}_t^i) \setminus (\mathbf{s}_{t-1}^i|_{I_{i,2}^{(t)}}, \mathbf{s}_t^i|_{I_{i,1}^{(t)}}) \mid (\mathbf{a}_t, \mathbf{s}_{t-1}^i|_{I_{i,2}^{(t)}}, \mathbf{s}_t^i|_{I_{i,1}^{(t)}})$. Equivalently, given $\mathbf{a}_t$, the reward depends on the state only through the current MIST states and their one-step parents.*

**Definition 3** (Minimality). *A sufficient set $U_i$ is* minimal *if there exists no strict sub-indices $J^{(t)} \subsetneq I_i^{(t)}$ and $J^{(t-1)} \subsetneq I_{i,2}^{(t)}$ such that $r_t \perp\!\!\!\perp (\mathbf{s}_{t-1}^i, \mathbf{s}_t^i) \setminus (\mathbf{s}_{t-1}^i|_{J^{(t-1)}}, \mathbf{s}_t^i|_{J^{(t)}}) \mid (\mathbf{a}_t, \mathbf{s}_{t-1}^i|_{J^{(t-1)}}, \mathbf{s}_t^i|_{J^{(t)}})$. Equivalently, no variable in $U_i$ can be removed without violating sufficiency.*

An illustrative example of identifying the minimal sufficient states for each task is shown in Fig. 2(a). The MIST states for task 1 and 2 are $\{\mathbf{s}^{1,1}, \mathbf{s}^{1,2}\}$ and $\{\mathbf{s}^{2,2}, \mathbf{s}^{2,3}\}$, respectively, traced by red edges.

**Using MIST for World Model and Policy Learning** Given the MIST states, we can employ them directly for policy learning. For a concise comparison, Fig. 2(b) contrasts our approach with Dreamer-based methods (Hafner et al., 2021; 2025). While Dreamer learns unstructured latent states, we obtain compact, task-specific representations by projecting Dreamer-style latents through our structured decomposition and masking unrelated components. Consequently, our policy can be produced solely from the MIST states (pink dashed nodes in the right panel). Similarly, reward reconstruction uses only the reward's parent variables (blue nodes), thereby respecting the identified structure.

## 3 APPROACH: MIST-WM

With the formal definition of MIST states in place, we now present the algorithmic framework of learning MIST-WM. The framework consists of two *interleaved* stages. In the first stage, agents actively explore the environment to learn a world model and identify minimal, sufficient, and task-specific state representations. In the second stage, the environment and task are scheduled through an adaptive curriculum. The remainder of this section details both stages. Fig. 3 provides an overview of our methodology, illustrating the roles of agent and environment within the MIST-WM framework.

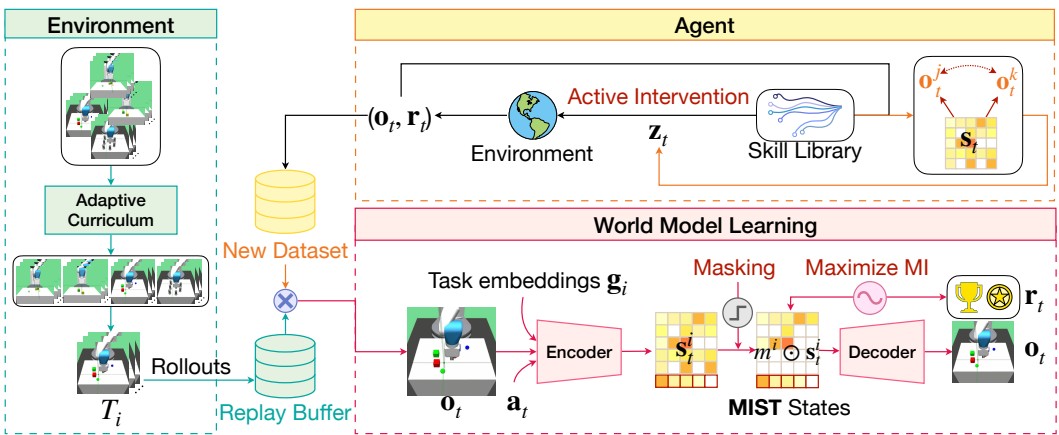

Figure 3: **Method overview.** On the agent side, for each new environment, the agent selects skills from its library to perform *active interventions*, collects informative data, and then learns a *world model with MIST states*. On the environment side, an adaptive *curriculum* schedules the next task/environment for the agent.

### 3.1 AGENT LEARNING: LEARNING MINIMAL SUFFICIENT TASK-SPECIFIC REPRESENTATIONS

We enable the agent to identify underlying states in a self-supervised, open-ended manner, analogous to human learning through active interaction. The target is to learn a world model with a MIST state representation (§3.1.1). To achieve this, the agent must acquire active intervention that discover and probe the environment, so as to gather informative data that supports the learning target (§3.1.2).

#### 3.1.1 LEARNING TASK-SPECIFIC MINIMAL SUFFICIENT WORLD MODELS

First, we focus on world-model learning. For each task $T_i$ (with task embedding $\mathbf{g}_i$), we learn a world model with the standard variational objective (Hafner et al., 2025):

$$
\mathcal{L}_{\text{WM}}^i(\theta) = \mathbb{E}_{\mathcal{D}} \Bigg[ - \sum_{t=0}^{T-1} \log p_\theta(\mathbf{o}_t \mid \mathbf{s}_t^i) + \beta \, \text{KL}\big( q_\theta(\mathbf{s}_t^i \mid \mathbf{o}_t, \mathbf{s}_{t-1}^i, \mathbf{a}_t) \,\|\, p_\theta(\mathbf{s}_t^i \mid \mathbf{s}_{t-1}^i, \mathbf{a}_t) \big)
$$

$$
- \log p_\theta(r_t^i \mid \mathbf{s}_t^i) - \log p_\theta(\gamma_t^i \mid \mathbf{s}_t^i) \Bigg],
$$

(1)

where $\theta$ denotes the world model parameters. However, this objective alone generally cannot identify the MIST states, as the latent space has no explicit structure (left of Fig. 2(b)). We therefore use *structure learning* to (i) recover a MIST subspace by learning a factored generative structure over latents and observations, and (ii) couple this structure with reward prediction through sufficiency and minimality scores. Let $\mathbf{s}_t^i \in \mathbb{R}^d$ be the learned latent state for task $T_i$ and $m^i \in (0,1)^d$ be a mask

selecting the task-specific subspace $\mathcal{S}_i^{\min}$ for $T_i$, i.e., $m^i \odot \mathbf{s}_t^i \in \mathcal{S}_i^{\min}$. We define the **Sufficiency Score** $\mathcal{R}_{\text{suff}}^{(i)}(\theta)$ and the **Minimality Score** $\mathcal{R}_{\min}^{(i)}(\theta)$ as follows.

The sufficiency score uses the masked states to maximize the likelihood of the observed rewards:

$$\mathcal{R}_{\text{suff}}^{(i)} = \mathbb{E}_{\mathcal{D}}\left[\sum_{t=0}^{T-1} \log p_\theta\big(r_t \mid m^i \odot \mathbf{s}_t^i, \mathbf{a}_t\big)\right]. \tag{2}$$

For the minimality score, we encourage the mask to retain only the necessary state dimensions by maximizing the mutual information between the masked states and the task descriptor, while penalizing the mask's $\ell_1$ norm:

$$\mathcal{R}_{\min}^{(i)} = \text{MI}\big(m^i \odot \mathbf{s}_{1:T}^i, \mathbf{g}^i\big) - \lambda_M \|m^i\|_1, \tag{3}$$

where $\lambda_M$ is a balancing coefficient. $\mathbf{g}_i$ is the task information. When no such auxiliary information (e.g., language instructions) is available, we set $\mathbf{g}^i$ to be a function of the reward sequence $r^i$.

We instantiate the world model with Dreamer-v3 (Hafner et al., 2025) to obtain the latent states, learn the soft mask $m^i$ via a gating-mask objective adapted from Rajamanoharan et al. (2024), and estimate the mutual information term using MINE (Belghazi et al., 2018). Both approximated objectives are provided in Appendix B.1. When moving to a new task (Fig. 2(a)), the dimensionality of the latent state is allowed to increase every time. Concretely, when transitioning from task $T_i$ to $T_{i+1}$, we fine-tune the encoder parameters learned for $T_i$ and adapt the final layer to accommodate the additional state dimensions required by $T_{i+1}$. This preserves the previously learned subspace while expanding the representation for new task-specific factors. Details are in Appendix B.4.

With these structure-learning objectives, we identify a minimal sufficient subspace from the learned latents and restrict downstream control to this subspace; that is, both the policy and value function operate on $\mathcal{S}_i^{\min}$. For task $T_i$, we optimize the cumulative return $\mathcal{J}_{\text{RL}}^{(i)}(\psi) =$
$$\mathbb{E}\left[\sum_{t=0}^{T-1} \gamma^t \, r^i(\mathbf{s}_t^i, \mathbf{a}_t)\right], \quad \text{with } \mathbf{a}_t \sim \pi_\psi\big(\cdot \mid m^i \odot \mathbf{s}_t^i\big).$$

### 3.1.2 LEARNING ACTIVE INTERVENTION SKILLS

Identifying such MIST states first requires informative data for each task; without it, the learned latents $\mathbf{s}^i$ cannot reliably capture the underlying factors. Here, "informative" means trajectories that expose how latent factors influence observations and rewards, rather than passively reflecting a narrow slice of the state space. While offline datasets can sometimes provide this, in a new environment an agent without purposeful exploration will generally struggle to uncover task-relevant states. Therefore, upon entering a novel environment, the agent actively collects data through *active interventions*, by executing meaningful skills $\mathbf{z}$ that are designed to probe and differentiate latent factors. Here, by *intervention* we mean that the skill policy deliberately perturbs the environment to induce distinguishable trajectories for different latent factors, rather than the safety-oriented notion of intervention commonly used in robotics. Our goal is to leverage these skills to identify informative interaction trajectories that facilitate the discovery of task-specific latent states (Sec. 3.1.1).

These skills are high-level latent variables sampled from a distribution $p_\eta$, and actions are drawn from a skill-conditioned policy $\pi_\varepsilon(\mathbf{a} \mid \mathbf{s}, \mathbf{z})$. Accordingly, we seek skills that *maximize* the information gain about the world-model parameters $\theta$ in the new environment while *ensuring* broad, task-agnostic state coverage, following the standard skill-discovery surrogate (Eysenbach et al., 2019). Consequently, the agent maintains a diverse library of skills. Accordingly, our objective decomposes into two consecutive procedures: (i) building a skill library, and (ii) selecting informative skills to intervene on the environment to enhance information gain for world model learning.

**Learning the Skill Library** $\mathcal{Z}$ Following the mutual information skill learning (MISL) framework (Eysenbach et al., 2019; Park et al., 2024; Zheng et al., 2025), we learn the *exploration policy* $\pi_\varepsilon(\mathbf{a} \mid \mathbf{s}, \mathbf{z})$ to maximize state–skill mutual information and action entropy. We select the prior distribution of skills as a uniform distribution over the d-dimensional unit hypersphere (a uniform von Mises–Fisher distribution (Mardia & Jupp, 2009), $p(\mathbf{z}) = \text{Unif}\big(\mathbb{S}^{d-1}\big)$) and learn the *discriminator* $q_\phi(\mathbf{z} \mid \mathbf{s})$ to make skills identifiable from states.

Specifically, we maximize the lower bound of mutual information with the objectives:

$$\max_{\varepsilon, \eta, \phi} \mathbb{E}\Big[\mathcal{H}\big(\pi_\varepsilon(\cdot \mid \mathbf{s}, \mathbf{z})\big) + \log q_\phi(\mathbf{z} \mid \mathbf{s}) - \log p_\eta(\mathbf{z})\Big] \tag{4}$$

where $\mathcal{H}$ computes the conditional entropy. Empirically, we sample a skill $\mathbf{z} \sim p_\eta$ per $k$ steps, roll out $\pi_\varepsilon(\cdot \mid \mathbf{s}, \mathbf{z})$, treat $\log q_\phi(\mathbf{z} \mid \mathbf{s})$ as an intrinsic reward and maximize policy entropy analytically. To capture temporally extended effects, we replace $\mathbf{s}$ with a segment encoder $\phi_{\mathrm{seg}}(\mathbf{s}_{t+1:t+k}, \mathbf{s}_t)$.

**Maximizing the Information Gain** Within the skill library, we select those could potentially provide the most information gain for world-model learning. We instantiate the information gain term by *selecting* the skill that best facilitates the discovery of latent representations. Let $\mathbf{s}_t^{n_i}$ denote the $i$-th latent factor at time $t$, taking values in the space $\mathcal{V}_i$. For two distinct values $v, v' \in \mathcal{V}_i$, under the skill-conditioned exploration policy $\pi_\varepsilon(\cdot \mid \mathbf{s}, \mathbf{z})$, we define the $k$-step observation segments as

$$\mathbf{o}_{t:t+k}^{(v)} \sim \mathcal{P}\Big( \mathbf{o}_{t:t+k} \,\Big|\, \mathbf{s}_t^{(n_i)} = v, \, \pi_\varepsilon(\cdot \mid \mathbf{z}) \Big), \qquad \mathbf{o}_{t:t+k}^{(v')} \sim \mathcal{P}\Big( \mathbf{o}_{t:t+k} \,\Big|\, \mathbf{s}_t^{(n_i)} = v', \, \pi_\varepsilon(\cdot \mid \mathbf{z}) \Big).$$

These rollouts can be generated directly by the agent experimenting in the environment, e.g., lifting two cubes with different weights using different skills. A skill is informative if it induces observation segments whose distributions differ markedly for two distinct latent-factor values $v$ and $v'$. When $\mathcal{V}_i$ is continuous, $v$ and $v'$ can be understood as the valuations of two independent samples from $\mathcal{V}_i$, chosen such that $v \neq v'$. In particular, when $\mathbf{o}_{t:t+k}^{(v)}$ and $\mathbf{o}_{t:t+k}^{(v')}$ are clearly distinguishable, the resulting contrast reveals the latent factors responsible for the difference.

Therefore, we encourage this "separability of segments" produced from different values of $\mathbf{s}_t^{(n_i)}$ via a distance metric. Empirically, we encode each sequence with $\phi_\xi(\cdot)$. For each anchor $\mathbf{o}_{t:t+k}^v$, we choose a positive sample $\mathbf{o}_{t:t+k}^{(v^+)}$ from the same valuation of latent variables (i.e., $\mathbf{s}_t^{(n_i)} = v$ but with another rollout segment), and a set of negatives $\{\mathbf{o}_{t:t+k}^{(v')}\}_{v' \in \mathcal{N}(v)}$. We utilize an InfoNCE-style contrastive loss (Oord et al., 2018):

$$\mathcal{L}_{\mathrm{contrast}}^{(n_i)}(\varepsilon, \xi) = \mathbb{E}_{a_{t:t+T-1} \sim \pi_\varepsilon(\cdot|z)} \left[ -\log \frac{\mathrm{d}(\mathbf{x}_v, \mathbf{x}_{v^+})}{\mathrm{d}(\mathbf{x}_v, \mathbf{x}_{v^+}) + \sum_{v' \in \mathcal{N}(v)} \mathrm{d}(\mathbf{x}_v, \mathbf{x}_{v'})} \right], \tag{5}$$

where $j^+$ denotes the positive index and $\mathcal{N}(j)$ the negative set. $\mathbf{x}$ is the encoded trajectory output, and $\mathrm{d}(u, v) := \exp(\mathrm{cosine}(u, v))$. Hence, for any task $T$ under the environment and curriculum specification (§3.2), the learned skills guide interaction to maximize expected information gain about task-relevant factors, yielding data most useful for world-model learning. Other than functioning as "scoring", this objective also updates the exploration policy $\pi_\varepsilon$ to make the agents grasp the skills that prioritize these behaviors.

For learning the skill library, we adopt off-the-shelf MISL algorithms; in particular, DIAYN (Eysenbach et al., 2019) and METRA (Park et al., 2024). Details are given in Appendix B.2. Each environment provides minimal design scaffolding to ensure that such informative segments can be discovered. For example, to encourage the discovery of object mass, we place two cubes with different weights on the workstation and allow the agent to infer the latent factors through interactions. Details are given in Appendix C.2.

## 3.2 Environment Learning: Adaptive Curriculum for World Models and Tasks

We have discussed above, for a given sequence of environments and tasks, how to learn `MIST-WM` by identifying minimal sufficient representations and how to obtain informative interactions via exploration. Let us now address how to schedule environments and tasks for the agent.

We use an adaptive curriculum that selects and orders environment–task pairs to expose the agent to informative transitions for world-model learning while progressively staging difficulty to accelerate downstream control. We optimize the ordering of environment and tasks (a permutation $\sigma$ over $\{T_i\}_{i=1}^N$). Since each task is instantiated in a specific environment, $\sigma$ is selected to maximize *data efficiency* for both world-model learning and policy learning.

To achieve our goal that learns the world model and downstream tasks as data-efficiently as possible, we operationalize it as minimizing world-model learning error and maximizing the cumulative rewards across all environments and tasks.

$$\min_\sigma \sum_{j=1}^N \mathcal{L}_{\mathrm{WM}}\big(T_{\sigma(j)} \mid \mathcal{D}_{\sigma(<j)}\big) - \lambda_R \sum_{j=1}^N \mathcal{J}_{\mathrm{RL}}^{(\sigma(j))}(\psi). \tag{6}$$

Specifically, for world-model learning, since the latent dynamics are unknown, following Sekar et al. (2020), we estimate the error via an ensemble $\{f_\theta^{(m)}\}_{m=1}^M$ and define a disagreement. For downstream task learning, let $p_\theta(r \mid \mathbf{s}, \mathbf{a})$ be the reward head conditioned on the current learned latent states. We use the reward loss to measure it directly. Hence, we have the evaluation proxy:

$$C_T(\mathcal{D}) := \mathbb{E}_{(\mathbf{s},\mathbf{a}) \sim \mathcal{D}} \left[ d_{\mathrm{var}}\Big(\{f_\theta^{(m)}(\mathbf{s}, \mathbf{a})\}_{m=1}^M\Big) - \log p_\theta(r \mid \mathbf{s}, \mathbf{a}) \right], \tag{7}$$

where $d_{\mathrm{var}}$ is the variance. This induces a natural notion of *task difficulty*. Following Unsupervised Experiment Design (UED) (Dennis et al., 2020; Rigter et al., 2024), we select the next environment to minimize worst-case error in world-model and task learning. Concretely, we explore all environments with the policy $\pi_\varepsilon$ and then have a buffer of per-task error scores $C_T(\mathcal{D})$. We select the next environment to reduce the worst-case error in world-model and task learning. Concretely, we first explore all environments with the policy $\pi_\varepsilon$ and maintain a per-task error buffer $C_T(\mathcal{D})$. At each step, we choose the hardest environment, that is, the one with the largest current error, and collect additional data there: $T^\star = \arg\max_T C_T(\mathcal{D})$. We then select $T^\star$ for learning. By repeating this procedure iteratively, we progressively reduce the maximum error across environments. In effect, this approximates solving a min–max objective, i.e., minimizing the maximum of the total error in world-model and policy learning.

## 4    RELATED WORKS

**Structures in World Model** Identifying meaningful structure in model-based reinforcement learning (MBRL) and world models is critical for generalization. Prior work can be broadly grouped into three lines. The first line focuses on compact state abstractions in (PO)MDPs. Representative examples include bisimulation-based methods (Ferns et al., 2004; Castro, 2020; Zhang et al., 2021), which aggregate states that are equivalent in return and transition dynamics under the abstraction. Other model-free approaches instead incorporate contrastive objectives to learn invariant features, as in CURL (Laskin et al., 2020). The second line aims to learn task- or control-aware representations in a model-based manner. Methods such as TIA (Fu et al., 2021) and Denoised MDP (Wang et al., 2022a) learn representations tailored to controllability or task-specific representations, thereby improving planning. The third line recovers structured world models by identifying graph structure in MDPs or POMDPs, i.e., learning factored MDPs (Boutilier et al., 2000) from data. These approaches assume and exploit an underlying causal graph, either over observable states (Pitis et al., 2020; Wang et al., 2022b; Ji et al., 2024) or latent states (Liu et al., 2023; Huang et al., 2022), to support more generalizable decision making. Our work is most closely related to the last two lines but differs from them: rather than learning generic controllable representations or full graph structures, we explicitly target *task-specific minimal sufficient* representations. In particular, we recover a factored structure but focus only on the components causally connected to the task, yielding compact, control-sufficient latents that can be integrated directly into policy learning.

**Unsupervised RL** Unsupervised reinforcement learning (RL) aims to acquire meaningful behavior without external rewards, and prior work largely follows two directions. One direction focuses on *intrinsic motivation*, where agents optimize surrogate signals that capture knowledge or competence about the environment. Examples include modeling prediction error as a novelty signal (Burda et al., 2019), using curiosity-based measures (Pathak et al., 2017; Kauvar et al., 2023), estimating disagreement among ensemble models (Sekar et al., 2020), and increasing empowerment or controllability across environments (Eysenbach et al., 2019; Tiomkin et al., 2024). The other direction emphasizes *unsupervised skill discovery*, where those methods maximize the mutual information between latent skills and states to learn diverse, meaningful behaviors (Eysenbach et al., 2019; Park et al., 2024; Hu et al., 2024; Zheng et al., 2025). Our work is most closely aligned with the latter: the *active intervention* component of MIST-WM learns latent skills specifically to probe the environment and expose task-relevant latent factors, thereby enabling the identification of minimal, sufficient representations for downstream control. For curriculum learning, we draw inspiration from unsupervised environment design (UED) (Dennis et al., 2020; Rigter et al., 2024), which provides curricula over sets of tasks for either policy learning or world-model learning. The work of Dennis et al. (2020) focuses on designing environments specifically for policy learning, while Rigter et al. (2024) studies reward-free cases. In contrast, our approach considers both settings simultaneously: we leverage the learned world model itself to progressively guide task selection to shape the curriculum.

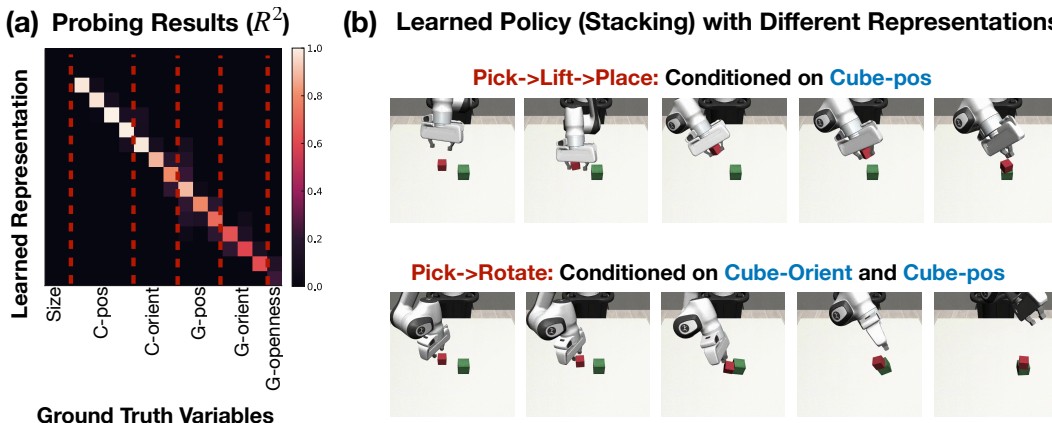

Figure 4: Results on Robosuite Stacking environment. (a) The probing results ($R^2$) matrix ("C" and "G" mean "cube" and "gripper" respectively), and (b) Different policies conditioned on different learned representation variables. (c) Average probing results on the Robosuite of baselines.

## 5 EXPERIMENTAL STUDY

We evaluate `MIST-WM` along four axes that mirror our design goals: (i) *Representation quality*: does the learned latent recover a minimal, task-sufficient representation? (ii) *Task performance*: given a fixed task sequence, does `MIST-WM` achieve strong sample efficiency relative to baseline world models? (iii) *Generalization*: can previously learned representations be recombined to solve novel tasks via skill-level, compositional, and unseen-task generalization? (iv) *Ablations*: how do the components of `MIST-WM` contribute across different stages?

For a comprehensive evaluation, we test `MIST-WM` on DMControl(Cheetah, Walker, Reacher in Tunyasuvunakool et al. (2020)), RoboSuite (Zhu et al., 2020), and Meta-World Yu et al. (2020), and compare against baselines spanning the design space: world-model backbones (Dreamer-V3 (Hafner et al., 2025), DINO-WM (Zhou et al., 2025), TD-MPC2 (Hansen et al., 2023)), factorization methods (Factored Dreamer (Liu et al., 2023)), intrinsic-motivation approaches (Curiosity Replay (Kauvar et al., 2023), Plan2Explore (Sekar et al., 2020)), and MISL methods (METRA (Park et al., 2024)).

**> Learning Minimal Sufficient Representation** We use a controlled RoboSuite setup (Zhu et al. (2020), Fig. 4a) where ground-truth simulator states can be found in simulators, allowing us to directly assess whether the learned representation recovers the underlying latent factors. Specifically, we assess whether `MIST-WM` can recover a minimal, task-sufficient subspace by mapping learned coordinates to ground truth via linear probes and reporting alignment $R^2$ (coefficient of determination Wright (1921)). The procedure for mapping the learned representations to ground-truth variables is detailed in Appendix B.7. We also compare the learned representations with ablations and baselines through analyses along two axes: *(i) Data collection:* Random (matched budget), MISL-only (METRA (Park et al., 2024)), and intrinsic-reward methods (Curiosity Replay (Kauvar et al., 2023) and Plan2Explore (Sekar et al., 2020)). *(ii) World model:* using foundation-model (DINO-WM (Zhou et al., 2025)) and factorized models (I-Factor (Liu et al., 2023), Denoised MDP (Wang et al., 2022a)).

**Take-Away 1: `MIST-WM` can recover task-relevant representations.** As shown in Fig. 4(a), we obtain near one-to-one alignment for task-relevant factors, including cube position/orientation and gripper position/orientation. However, the static or non-intervenable variables (e.g., object size) are not identified. Likewise, very fine-grained cues (i.e., the gripper open/close states) are not identified: though such signals exist in the data, they are hard to capture with contrastive pairs. In general, `MIST-WM` identifies a compact subspace that retains what matters for the task and discards others. Fig. 4(c) shows the average $R^2$ (diag) and Appendix Table A9 gives the full results. We find that learned `MIST-WM` attains the highest alignment. Within data collection, aside from ours, MISL

(METRA) yields the best alignment, while intrinsic-reward methods are comparable. For world models, structural representation learning (I-Factor and Denoised MDP) outperform DINO-WM on average, suggesting the effectiveness of learning disentangled representations of these two.

**Take-Away 2: Different learned state representations induce qualitatively different policy behaviors**. In RoboSuite Stacking (Fig. 4(c)), conditioning the policy on position features alone yields an optimal pick–lift–place sequence. In contrast, conditioning on both orientation and position enables a shortcut: when the objects are close, the agent can directly rotate one object onto the other. The indicates that an aligned latent space provides more controllable and interpretable behaviors.

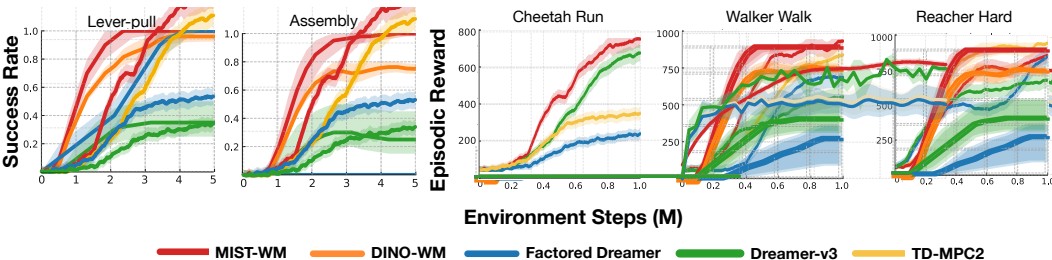

Figure 5: Results (across 5 seeds) on single-task learning, including **robotic manipulation** tasks (Meta World) and **locomotion** (DMControl). Shaded areas indicate the standard deviation.

**> Single-Task Policy Learning** We consider evaluating the single task policy learning (no curriculum, only with MIST states learning and active intervention skills) on benchmarks, including robotic control (Meta-World (Yu et al., 2020)) and locomotion tasks from DMControl (Tunyasuvunakool et al., 2020). We compare against DINO-WM (Zhou et al., 2024), I-Factor (a Dreamer-based factored model; labeled *Factored Dreamer*) (Liu et al., 2023), Dreamer-V3 (Hafner et al., 2025), and TD-MPC2 (Hansen et al., 2024). Regarding policy learning, for DMControl and Franka-kitchen, all methods use Proximal Policy Optimization (PPO, Schulman et al. (2017)), whereas for Meta-World and RoboSuite, they use Soft Actor-Critic (SAC, Haarnoja et al. (2018)). Results are in Fig. 5 with full results in Appendix Fig. A3, Table A10-A11. Computational costs are given in Appendix C.4.

**Take-Away 3: `MIST-WM` consistently improves data efficiency and policy learning performances across most tasks, except for Reacher-Hard.** On Meta-World, DINO-WM is typically the second-best, while Factored Dreamer and Dreamer-V3 often lag behind DINO-WM. On locomotion, `MIST-WM` outperforms TD-MPC2 on Cheetah and Walker but underperforms on Reacher-Hard, while still exceeding Dreamer-based baselines. We attribute this to differing objectives: TD-MPC methods optimize a model-free control objective, whereas `MIST-WM` builds on Dreamer objectives.

**> Generalization on Skills, Compositions, and Unseen Tasks** We evaluate `MIST-WM` on three types of generalization using the learned MIST representations. Summary results are in Fig. 6 and full results are in Fig. A4.

**Take-Away 4: The learned MIST representations consistently improve skill, compositional, and unseen-task generalization.** *(i) Skill generalization.* Can the learned MIST states facilitate better skill-level generalization? We study this in RoboSuite (Zhu et al., 2020) and Franka-Kitchen (Gupta et al., 2019), where source tasks are learned and then transferred to target tasks that require composing multiple the learned representations. Results show that `MIST-WM` achieves large improvements on both source and target tasks, with especially pronounced gains in RoboSuite target tasks and harder Kitchen tasks (e.g., Kitchen-light). This suggests that the learned representation help scale to complex, long-horizon behaviors. *(ii) Compositional generalization.* Here we combine learned "objects" and "skills" to create novel tasks. In Meta-World (Yu et al., 2020), we train on (door-unlock, drawer-open, faucet-close, handle-pull) and evaluate on (door-open, drawer-close). `MIST-WM` outperforms all baselines on both base and target tasks, with especially strong gains on the door-open target. *(iii) Unseen-task generalization.* Finally, in Meta-World, we test on coffee-push, a task where neither "coffee" nor "push" appeared during training. `MIST-WM` generalizes well, indicating that the learned latent representations capture transferable structure that extends to previously unseen factors. We learn an expandable state space that supports new tasks. When a task involves new compositions of skills or objects, the agent can initialize from previously learned states and learn the corresponding policy with fewer samples.

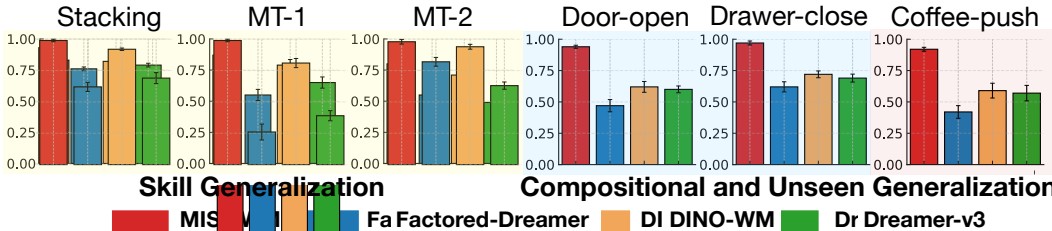

Figure 6: Success rate (across 5 seeds) on skill and compositional generalization, including tasks in Stacking (Robosuite), MT-1 & 2 (kitchen), and Door-open, Door-close, Coffee-push (Meta-World). Error bars indicate the standard deviation.

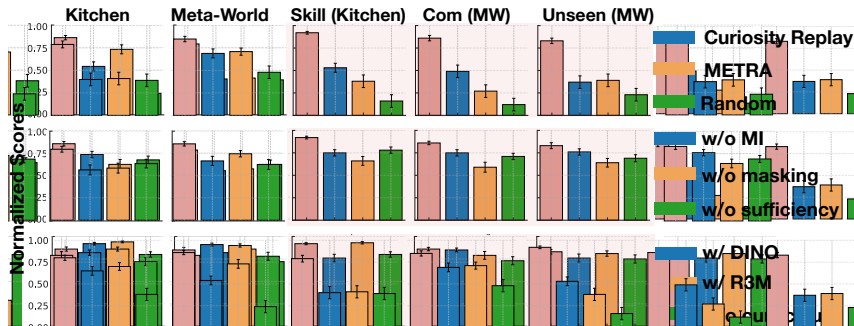

Figure 7: **Ablation studies**. We consider studies on exploration policy (Row 1); World model terms (Row 2); and Backbones and curriculum (Row 3). Pink bars are original MIST-WM and error bars indicate the standard deviation.

**> Ablation Studies** To evaluate each component of MIST-WM, we conduct ablations along three axes: *(i) Exploration policy* for data collection: *Random*, *Intrinsic motivation* (Curiosity Replay (Kauvar et al., 2023)), and *MISL* (METRA); *(ii) World Model terms*: removing the MI term in the *minimality* score, removing *masking* in the minimality score, and removing the *sufficiency* term; and *(iii) Backbone and curriculum*: replacing the Dreamer-v3 encoder with *DINO-WM or R3M*, and training w.r.t. a *no-curriculum* variant (i.e., random order). We report both single-task learning and cross-environment generalization (Fig. 7, full results are in Appendix Fig. A7). All scores are normalized to an *Oracle* that directly receives ground-truth states in each environment.

**Take-Aways for Ablations:** The default MIST-WM achieves the best overall. In generalization settings (pink panels), performance approaches the Oracle (often $\approx 0.9$), and exceeds single-task learning, indicating that the *learned representations transfer across tasks*. For case (i), METRA is typically second-best, while Random is consistently worst. On generalization tasks, Curiosity Replay sometimes surpasses METRA, consistent with its broader, task-agnostic coverage, improving transfer. For case (ii), Removing MI, masking, or sufficiency uniformly degrades performance. The effects are similar in single-task settings; in generalization, masking has the largest impact, suggesting that enforcing compactness improves transfer. For case (iii), using DINO-WM (Zhou et al., 2024) or R3M (Nair et al., 2023) features can help on some tasks, but is not consistently superior. Removing the curriculum causes larger drops in generalization than in single-task learning, reflecting the curriculum's role in generalization across tasks.

## 6 CONCLUSION

In this work, we study how agents can learn world models whose latent space provides a *minimal, sufficient* representation for each task. Our aim is to distill compact latents directly from pixels, focusing the agent on task-relevant factors. To this end, we co-design *active interventions* and an *adaptive environment curriculum*: interventions discover meaningful behaviors and gather informative interactions; structure-aware objectives learn minimal and sufficient representations; and the curriculum schedules environments to match the agent's progress. Together, this co-design offers a principled route to learning meaningful latent spaces for world models, supporting more efficient and generalizable policy learning, mirroring aspects of the human learning process and decision-making.

REPRODUCIBILITY STATEMENT

For our method, details of the model choices and hyperparameters are provided in Appendix C.3. Modifications to the benchmarks are described in Appendix C.2. Reproduction details for other methods and the specific baseline designs are also provided in Appendix C.3.

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

# Appendix

# A  NOTATIONS AND CONCEPTS

## A.1  NOTATIONS

Table A1 collects the notations used in the theorem proofs for clarity and consistency.

| Variable | Explanation | Support |
|---|---|---|
| $\mathcal{S}$ | (latent) state space | $\mathcal{S} \subseteq \mathbb{R}^{d_s}$ |
| $\mathcal{A}$ | action space | $\mathcal{A} \subseteq \mathbb{R}^{d_a}$ |
| $\mathcal{O}$ | observation space | $\mathcal{O} \subseteq \mathbb{R}^{d_o}$ |
| $\mathcal{D}$ | dataset / replay buffer distribution | trajectories over $(\mathbf{s}, \mathbf{a}, \mathbf{o}, r)$ |
| $\mathcal{Z}$ | skill library (set of learnable skills) | $\mathcal{Z} \subseteq \mathbb{R}^{d_z}$ |
| $\gamma$ | discount factor | $\gamma \in (0,1)$ |
| $\rho_0$ | initial-state distribution | $\rho_0 : \mathcal{S} \to [0,1]$ |
| $\{T_i\}_{i=1}^N$ | set of tasks | $i = 1, \ldots, N$ |
| $\mathbf{s}_t^i = (\mathbf{s}_t^{i,1}, \ldots, \mathbf{s}_t^{i,d_i})$ | factored (learned) state of task $i$ | $d_i$ state factors |
| $I_i^{(t)}$ | indices of state factors influencing reward $r_t$ | $I_i^{(t)} \subseteq [d_i]$ |
| $I_i^{(t-1)}$ | indices of one-step parents of $I_i^{(t)}$ | $I_i^{(t-1)} \subseteq [d_i]$ |
| $U_i$ | union of current and one-step parents | $U_i = I_i^{(t)} \cup I_i^{(t-1)}$ |
| $\mathbf{s}_t^i\big|_{I_i^{(t)}}$ | current-time MIST states | subset of $\mathbf{s}_t^i$ |
| $\mathbf{s}_{t-1}^i\big|_{I_i^{(t-1)}}$ | one-step parent MIST states | subset of $\mathbf{s}_{t-1}^i$ |
| $\mathbf{g}_i$ | task descriptor / embedding for $T_i$ (can be rewards in simple cases) | $\mathbf{g}_i \in \mathbb{R}^{d_g}$ |
| $m^i$ | soft/binary mask selecting task-specific subspace for $T_i$ | $m^i \in (0,1)^{d_i}$ |
| $\mathcal{S}^{\min}$ | minimal sufficient subspace for task $T_i$ | $\mathcal{S}^{\min} \subseteq \mathcal{S}$ |
| $T$ | horizon length | $T \in \mathbb{N}^+$ |
| $\mathbf{z}$ | latent skill variable | $\mathbf{z} \in \mathbb{R}^{d_z}$ |
| $\mathcal{N}(v)$ | negative sample index set for contrastive learning | $\mathcal{N}(v) \subseteq \{1, \ldots, N\}$ |
| $\mathbf{x}_v, \mathbf{x}_{v^+}, \mathbf{x}_{v'}$ | encoded representations of anchor, positive, and negative samples | $\mathbf{x} \in \mathbb{R}^{d_h}$ |
| **Function** | | |
| $\mathcal{G}$ | dynamic Bayesian network (DBN) over $(\mathbf{s}_t, \mathbf{a}_t, r_t)$ | graph structure |
| $\mathcal{G}^{(i)}$ | task-specific DBN over $(\mathbf{s}_t^i, \mathbf{a}_t, \mathbf{o}_t, r_t^i)$ | graph structure |
| $\mathbb{P}_s(\mathbf{s}_t \mid \mathbf{s}_{t-1}, \mathbf{a}_{t-1})$ | state transition probability | $\mathcal{S} \times \mathcal{A} \to \Delta(\mathcal{S})$ |
| $\mathbb{P}_o(\mathbf{o}_t \mid \mathbf{s}_t)$ | observation function | $\mathcal{S} \to \Delta(\mathcal{O})$ |
| $\mathcal{R}(\mathbf{s}_t, \mathbf{a}_t)$ | reward function | $\mathcal{S} \times \mathcal{A} \to \mathbb{R}$ |
| $\mathbb{P}_s^{(i)}(\mathbf{s}_t \mid \mathbf{s}_{t-1}, \mathbf{a}_{t-1})$ | transition probability of task $i$ | $\mathcal{S} \times \mathcal{A} \to \Delta(\mathcal{S})$ |
| $\mathbb{P}_o^{(i)}(\mathbf{o}_t \mid \mathbf{s}_t)$ | observation function of task $i$ | $\mathcal{S} \to \Delta(\mathcal{O})$ |
| $\mathcal{R}^{(i)}(\mathbf{s}_t, \mathbf{a}_t)$ | reward function of task $i$ | $\mathcal{S} \times \mathcal{A} \to \mathbb{R}$ |
| $\mathrm{pa}_{\mathcal{G}^{(i)}}(\cdot)$ | parent operator in task-specific DBN $\mathcal{G}^{(i)}$ | node $\mapsto$ set of parents |
| $\mathcal{L}_{\mathrm{WM}}^i(\theta)$ | task-conditioned world model objective for $T_i$ | variational training loss |
| $\mathcal{R}_{\mathrm{suff}}^{(i)}(\theta)$ | sufficiency score | $\mathbb{E}_{\mathcal{D}}\left[\sum_{t=0}^{T-1} \log p_\theta(r_t \mid m^i \odot \mathbf{s}_t^i, \mathbf{a}_t)\right]$ |
| $\mathcal{R}_{\min}^{(i)}(\theta)$ | minimality score for selecting $\mathcal{S}^{\min}$ | model-dependent criterion |
| $p_\theta(r_t \mid m^i \odot \mathbf{s}_t^i, \mathbf{a}_t)$ | conditional reward likelihood (task $i$) | scalar density |
| $\mathcal{J}_{\mathrm{RL}}^{(i)}(\psi)$ | expected cumulative return objective for task $i$ | $\mathbb{E}\left[\sum_{t=0}^{T-1} \gamma^t r^i(\mathbf{s}_t^i, \mathbf{a}_t)\right]$ |
| $\pi_\psi(\cdot \mid m^i \odot \mathbf{s}_t^i)$ | task-conditioned policy based on masked latent states | distribution over $\mathcal{A}$ |
| $p_\eta$ | skill prior distribution | $\mathbf{z} \sim p_\eta$ |
| $\pi_\varepsilon(\mathbf{a} \mid \mathbf{s}, \mathbf{z})$ | skill-conditioned policy | distribution over $\mathcal{A}$ |
| $p(\mathbf{z})$ | uniform skill prior on unit hypersphere | $\mathrm{Unif}(\mathbb{S}^{d-1})$ |
| $q_\phi(\mathbf{z} \mid \mathbf{s})$ | discriminator for skill identification from states | $\mathcal{S} \to \Delta(\mathcal{Z})$ |
| $\phi_{\mathrm{seg}}(\mathbf{s}_{t+1:t+k}, \mathbf{s}_t)$ | segment encoder mapping state segments to embeddings | $\mathbb{R}^{(k+1)\times d_s} \to \mathbb{R}^{d_h}$ |
| $\phi_\xi(\cdot)$ | sequence encoder for contrastive learning | trajectories $\to \mathbb{R}^{d_h}$ |
| $\mathcal{L}_{\mathrm{contrast}}^{(n_i)}$ | InfoNCE-style contrastive loss for skill discrimination | scalar objective |
| $\mathrm{d}(\mathbf{x}, \mathbf{x}')$ | similarity/distance function between embeddings | $\mathbb{R}^{d_h} \times \mathbb{R}^{d_h} \to \mathbb{R}$ |
| $C_T(\mathcal{D})$ | evaluation proxy combining disagreement and reward likelihood | $\mathbb{E}_{(\mathbf{s}, \mathbf{a}) \sim \mathcal{D}}[\cdot]$ |
| $d_{\mathrm{var}}(\{f_\theta^{(m)}\}_{m=1}^M)$ | variance-based disagreement measure across ensemble predictions | $\mathbb{R}^M \to \mathbb{R}$ |
| $f_\theta^{(m)}(\mathbf{s}, \mathbf{a})$ | $m$-th ensemble dynamics model prediction | $\mathcal{S} \times \mathcal{A} \to \mathbb{R}$ |
| **Symbol** | | |
| $\odot$ | Hadamard (element-wise) product | $(\mathbb{R}^d \times \mathbb{R}^d) \to \mathbb{R}^d$ |
| $\mathbb{S}^{d-1}$ | $(d-1)$-dimensional unit sphere | $\{\mathbf{z} \in \mathbb{R}^d : \|\mathbf{z}\|_2 = 1\}$ |

Table A1: List of notations, explanations, and corresponding value ranges.

## A.2  CONCEPTS

Here we explain a few key concepts used in our framework as a complement of the main paper.

**Unstructured latent states** We instantiate the world model with a recurrent state-space model (RSSM) as in Dreamer-v3, which produces latent states $\mathbf{s}_t \in \mathbb{R}^d$ from observations $\mathbf{o}_t$ and actions $\mathbf{a}_t$ via

$$q_\theta(\mathbf{s}_{t+1} \mid \mathbf{o}_t, \mathbf{s}_t, \mathbf{a}_t), \qquad p_\theta(\mathbf{o}_t \mid \mathbf{s}_t), \quad p_\theta(\mathbf{s}_{t+1} \mid \mathbf{s}_t, \mathbf{a}_t).$$

These latent states are *unstructured* in the sense that no explicit factorization or task-specific subspace is imposed.

**Structure learning** On top of the unstructured latents, we perform *structure learning* to extract a task-specific subspace. For each task $T_i$, let $\mathbf{s}_t^i \in \mathbb{R}^d$ denote the corresponding Dreamer latent and $\mathbf{g}^i$ a task descriptor. We learn a mask $m^i \in (0, 1)^d$ and a factored generative model over $(\mathbf{s}_t^i, \mathbf{o}_t, r_t^i)$ conditioned on $\mathbf{g}^i$ to optimize Eq. 2-3 to encourage a low-dimensional, task-sufficient subspace.

**MIST states** For task $T_i$, the *Minimal, Sufficient, Task-specific (MIST) states* are the masked latents

$$\tilde{\mathbf{s}}_t^i = m^i \odot \mathbf{s}_t^i \in \mathcal{S}_i^{\min},$$

where $\mathcal{S}_i^{\min} \subseteq \mathbb{R}^d$ denotes the minimal sufficient subspace recovered by structure learning. Intuitively, MIST states are those coordinates of $\mathbf{s}_t^i$ that are (i) sufficient for predicting rewards $r_t^i$ given $\mathbf{a}_t$, and (ii) cannot be removed without violating sufficiency.

Concretely, the connection between *masked states*, *latent factors*, and the *expandable state space* is as follows. Dreamer provides an unstructured latent vector $\mathbf{s}_t^i \in \mathbb{R}^{d_i}$ for each task $T_i$, whose coordinates we interpret as latent factors. As we move to new tasks and the environment introduces additional mechanisms, we allow the latent dimensionality $d_i$ to grow (expandable state space). For each task $T_i$, we then learn a soft mask $m^i \in (0, 1)^{d_i}$ that gates these coordinates and defines the task-specific subspace $m^i \odot \mathbf{s}_t^i$, which we call the MIST states. Thus: (i) *latent factors* are the coordinates of $\mathbf{s}_t^i$, (ii) the *expandable state space* means that the dimensionality $d_i$ can increase across tasks, and (iii) the *masked states* $m^i \odot \mathbf{s}_t^i$ are the subset of those factors that our structure-learning objectives identify as minimal and sufficient for predicting rewards.

**Active intervention skills** To obtain informative data for structure learning, the agent executes *intervention skills* parameterized by a latent variable $\mathbf{z}$. We maintain a skill prior $p_\eta(\mathbf{z})$ (e.g., uniform on a hypersphere) and a skill-conditioned exploration policy

$$\mathbf{a}_t \sim \pi_\varepsilon(\cdot \mid \mathbf{s}_t, \mathbf{z}),$$

trained to maximize a mutual-information style objective (Eq. 4) and a contrastive objective (Eq. 5). Rolling out $\pi_\varepsilon$ yields *active interventions* that probe how latent factors affect observations and rewards, producing data that is informative for discovering MIST states.

**Segment separability** Given a latent factor $s_t^{(n)}$ taking values in some space $\mathcal{V}_n$, and a fixed skill $\mathbf{z}$, we consider $k$-step observation segments

$$\mathbf{o}_{t:t+k}^{(v)} \sim P\big(\mathbf{o}_{t:t+k} \mid s_t^{(n)} = v, \, \pi_\varepsilon(\cdot \mid \cdot, \mathbf{z})\big), \qquad \mathbf{o}_{t:t+k}^{(v')} \sim P\big(\mathbf{o}_{t:t+k} \mid s_t^{(n)} = v', \, \pi_\varepsilon(\cdot \mid \cdot, \mathbf{z})\big),$$

for two distinct values $v \neq v' \in \mathcal{V}_n$. We encode segments with $\mathbf{x}^{(v)} = \phi_\xi(\mathbf{o}_{t:t+k}^{(v)})$ and define *segment separability* via an InfoNCE-style contrastive loss. Skills that induce highly separable segments (low $\mathcal{L}_{\text{contrast}}^{(n)}$) provide strong evidence about the underlying latent factors and are therefore preferred for active interventions.

# B IMPLEMENTATION DETAILS ON MIST-WM

## B.1 MUTUAL INFORMATION AND MASKING LEARNING

We use MINE (Belghazi et al., 2018) to estimate the mutual information. Following Wang & Huang (2025), to avoid instability from inaccurate estimates early in training, the MI weight is annealed from a small value to its target with a cosine schedule:

$$\alpha_{\text{MI}}(t) = \begin{cases} \alpha_{\text{start}} + \frac{1}{2}(\alpha_{\text{end}} - \alpha_{\text{start}})(1 - \cos(\pi t)), & \alpha_{\text{end}} > \alpha_{\text{start}}, \\ \alpha_{\text{end}} + \frac{1}{2}(\alpha_{\text{start}} - \alpha_{\text{end}})(1 + \cos(\pi t)), & \text{otherwise}, \end{cases} \tag{A1}$$

where $t \in [0, 1]$ is normalized training time. This smoothly activates the MI constraint and mitigates variance-driven min–max oscillations.

To induce compact latents without the shrinkage artifacts of plain $\ell_1$ penalties, we use a modified gated mask inspired by Wang & Huang (2025); Rajamanoharan et al. (2024). The mask is applied to the latent state $\mathbf{s}$; the stochastic component is left unchanged. Let $\mathbf{s}$ denote the latent vector. We define a binary gate and a magnitude reparameterization:

$$\text{Gate:} \quad \text{Gate}(\mathbf{s}) := \mathbf{1}\big(\,|\mathbf{s}| + b_{\text{gate}} > 0\,\big), \tag{A2}$$

$$\text{Magnitude:} \quad f_{\text{mag}}(\mathbf{s}) := \exp(r_{\text{mag}})\,\mathbf{s} + b_{\text{mag}}, \tag{A3}$$

and the masked latent is

$$\boldsymbol{m} \odot \mathbf{s} := \text{Gate}(\mathbf{s}) \,\odot\, f_{\text{mag}}(\mathbf{s}). \tag{A4}$$

We couple this with an adaptive sparsity objective on the gate:

$$\mathcal{L}_{\ell_1}(\boldsymbol{m}) = \big\|\text{ReLU}\big(\text{Gate}(\mathbf{s})\big)\big\|_1, \tag{A5}$$

where the target sparsity threshold is increased smoothly using the cosine schedule in equation A1. This schedule provides fine-grained control over the active fraction of latent dimensions while preserving exploration efficiency.

## B.2 Details on Skill Library Learning

We use DIAYN (Eysenbach et al., 2019) and METRA (Park et al., 2024) as our skill library learning framework.

For DIAYN, let $z \sim p_\eta(z)$ be a skill sampled per episode/segment and $s \sim d^{\pi_\psi}(\cdot \mid z)$ the (discounted) state distribution induced by the skill–conditioned policy $\pi_\psi(a \mid s, z)$. Define

$$\mathcal{F}(\psi, \eta) = \underbrace{\mathcal{H}[A \mid S, Z]}_{\text{max-ent exploration under skills}} - \mathcal{H}[Z \mid S] + \mathcal{H}[Z], \tag{A6}$$

where entropies are taken under $z \sim p_\eta$, $s \sim d^{\pi_\psi}(\cdot \mid z)$, and $a \sim \pi_\psi(\cdot \mid s, z)$. Expanding the KL terms gives

$$\mathcal{F}(\psi, \eta) = \mathcal{H}[A \mid S, Z] + \mathbb{E}_{z \sim p_\eta,\, s \sim d^{\pi_\psi}(\cdot \mid z)}\big[\log p(z \mid s)\big] - \mathbb{E}_{z \sim p_\eta}\big[\log p_\eta(z)\big]. \tag{A7}$$

Using a variational classifier $q_\phi(z \mid s)$, we obtain the tractable lower bound

$$\mathcal{F}(\psi, \eta) \geq \mathcal{G}(\psi, \eta, \phi) := \mathbb{E}_{z \sim p_\eta,\, s \sim d^{\pi_\psi}(\cdot \mid z)}\Big[\underbrace{\mathcal{H}\big(\pi_\psi(\cdot \mid s, z)\big)}_{\text{entropy of } \pi} + \log q_\phi(z \mid s) - \log p_\eta(z)\Big]. \tag{A8}$$

For METRA, we consider the skill reward as:

$$r_\phi(\mathbf{s}, \mathbf{z}, s\mathbf{v}') = \big(\phi(\mathbf{s}') - \phi(\mathbf{s})\big)^\top \mathbf{z}. \tag{A9}$$

and the total objective is then as:

$$\mathcal{L}_{\text{METRA}}(\phi, \lambda) = \mathbb{E}_{(\mathbf{s}, \mathbf{z}, \mathbf{s}') \sim \mathcal{D}}\Big[\big(\phi(\mathbf{s}') - \phi(\mathbf{s})\big)^\top \mathbf{z} + \lambda \operatorname{clip}_\varepsilon\big(1 - \|\phi(\mathbf{s}') - \phi(\mathbf{s})\|_2^2\big)\Big], \quad \lambda \geq 0. \tag{A10}$$

Both are optimized using SAC (Haarnoja et al., 2018) to learn the exploration policy. Hyperparameters are given in Section C.3.

## B.3 Details on Curriculum Learning

We follow a UED-style scheduler for environment selection. At each step, we compute an ensemble-disagreement proxy $\Delta_T$ (model-error hardness) per environment from its buffer:

$$\Delta_T = \mathbb{E}_{(s, a) \sim \mathcal{B}_T}\big[\text{Tr}\big(\text{Cov}_{m=1}^M\big[\,f_{\theta^{(m)}}(s, a)\,\big]\big)\big]. \tag{A11}$$

We then normalize scores before sampling:

$$\tilde{\Delta}_T = \frac{\Delta_T - \min_{T'} \Delta_{T'}}{\max_{T'} \Delta_{T'} - \min_{T'} \Delta_{T'} + \varepsilon}. \tag{A12}$$

A Boltzmann distribution with temperature $\eta$ emphasizes harder environments:

$$p_\eta(T) \;=\; \frac{\exp\big(\tilde{\Delta}_T/\eta\big)}{\sum_{T'\in\mathcal{T}}\exp\big(\tilde{\Delta}_{T'}/\eta\big)}. \tag{A13}$$

Then, we mix domain randomization with error-driven selection: with probability $p_{\mathrm{DR}}$ (or if any $\mathcal{B}_T$ is empty), we sample environment parameters at random; otherwise, we sample according to $p_\eta$:

$$u \sim \mathcal{U}[0,1], \qquad T_{\mathrm{next}} \sim \begin{cases} \mathrm{DomainRandomisation}(\Theta), & \text{if } u < p_{\mathrm{DR}} \text{ or } \exists\, T : |\mathcal{B}_T| = 0, \\ \mathrm{Cat}\big(p_\eta(\cdot)\big), & \text{otherwise.} \end{cases} \tag{A14}$$

To compute the disagreement term, we use an ensemble of transition functions $f_\theta(\mathbf{s}, \mathbf{a})$ corresponding to the RSSM dynamics in the Dreamer model, following Ball et al. (2020); Sekar et al. (2020); Mendonca et al. (2021). Concretely, we train an ensemble of one-step predictors that map the current model state and action to the next model state. The ensemble is optimized jointly with the world model on latent transitions $\mathbf{s}' \sim p_\theta(\mathbf{s}' \mid \mathbf{s}, \mathbf{a})$, as defined in Eq. 1.

## B.4 EXPANDABLE LATENT SPACE

When transitioning from task $T_i$ to $T_{i+1}$, we allow the latent dimensionality to grow, $d_{i+1} > d_i$. We follow a simple over-parameterized heuristic that is consistent with practice and then rely on structure learning to select the task-relevant subspace. For each benchmark we initialize the Dreamer latent dimensionality $d_1$ to a fixed value and keep it shared across tasks at the beginning (e.g., $d_1 = 512$ for Meta-World, RoboSuite, and Kitchen, which have high-dimensional visual observations, and $d_1 = 256$ for DMControl). When moving from task $T_i$ to $T_{i+1}$, we allow the latent dimensionality to grow by a fixed increment, $d_{i+1} = d_i + \Delta d$. where we set $\Delta d = 32$ in all experiments. The choice of $d_1$ and $\Delta d$ is therefore a heuristic design decision rather than an oracle choice of the "correct" dimension. In practice, this slight over-parameterization is benign because our structure-learning module (through the sufficiency and minimality objectives and the learned mask $m^i$) identifies and uses only the task-relevant subset of dimensions (the MIST states), while irrelevant coordinates are effectively suppressed.

Concretely, we fine-tune the encoder and RSSM parameters up to the latent layer for $T_i$ and adapt the final linear layer that maps the encoder/RSSM features into $\mathbb{R}^{d_{i+1}}$. The additional $d_{i+1} - d_i$ coordinates are initialized randomly and trained for $T_{i+1}$. The decoders (observation, reward, and continuation) are similarly extended to read the expanded latent vector, but we fine-tune their parameters for the first $d_i$ dimensions and learn the additional weights for the new coordinates.

## B.5 ALGORITHMIC FRAMEWORKS

Algorithm 1 gives the full algorithm of `MIST-WM`.

## B.6 ADDITIONAL DETAILS ON REPRESENTATION LEARNING COMPONENTS

**Unstructured Latent States.** The Dreamer world model produces latent states $\mathbf{s}_t \in \mathbb{R}^d$ through the RSSM encoder $q_\theta(\mathbf{s}_t \mid \mathbf{o}_t, h_t, \mathbf{a}_{t-1})$. These latents are *unstructured*: no constraints are imposed on their factorization, and all coordinates may encode arbitrary mixtures of task-relevant and task-irrelevant features. During structure learning, the encoder parameters generating $\mathbf{s}_t$ remain **frozen** so that all subsequent structure-learning objectives operate over a fixed latent basis.

**Structure Learning.** Given the unstructured latents $\mathbf{s}_t$, we identify a structured subspace $_i^{\min}$ for each task $T_i$ via a learnable mask $m^i \in (0,1)^d$ and two task-specific objectives (Eq. 2 & Eq. 3). During this stage, the RSSM encoder and Dreamer dynamics are **frozen**; only the mask $m^i$ and the MI estimator parameters receive gradients.

**MIST States.** For each task $T_i$, the MIST representation is defined as

$$\tilde{\mathbf{s}}_t^i = m^i \odot \mathbf{s}_t^i \in_i^{\min},$$

which retains only the coordinates necessary for reward prediction and decision making. Policies and value functions $\pi_\psi(\mathbf{a}_t \mid \tilde{\mathbf{s}}_t^i)$ operate exclusively on this subspace, and gradients through RL losses $\nabla_\psi^{(i)}{}_{\mathrm{RL}}(\psi)$ do not update the world model.

**Active Intervention Skills.** When encountering a new task, the agent collects informative trajectories via latent skills $\mathbf{z} \sim p_\eta(\mathbf{z})$. Actions follow a skill-conditioned policy $\pi_\varepsilon(\mathbf{a}_t \mid \mathbf{s}_t, \mathbf{z})$, trained using the mutual-information based MISL objective (Eq. 4). Here only the skill policy $\eta$ and discriminator $q_\phi$ receive gradients; the world model and encoders stay **fixed** during skill learning.

**Segment Separability.** To select informative skills, we measure how distinguishable two segments generated from different latent-factor values are. Let $\mathbf{o}_{t:t+k}^{(v)}$ and $\mathbf{o}_{t:t+k}^{(v')}$ be observation segments produced under skill $\mathbf{z}$. After encoding them using $\phi_\xi$, we apply an InfoNCE objective (Eq. 5). Gradients update the segment encoder $\phi$ and the skill parameters $\varepsilon$.

Table A2 summarizes which objectives update which components.

| Module | WM Loss | Structure Losses | Skill Losses |
|---|---|---|---|
| Dreamer encoder / RSSM | ✓ | frozen | frozen |
| Reward / obs decoders | ✓ | ✓ | frozen |
| Mask $m^i$ | frozen | ✓ | frozen |
| MI estimator (MINE) | frozen | ✓ | frozen |
| Skill policy $\varepsilon$ | frozen | frozen | ✓ |
| Skill discriminator $q_\phi$ | frozen | frozen | ✓ |
| Segment encoder $\phi_\xi$ | frozen | frozen | ✓ |
| Policy / value $\psi$ | frozen | frozen | ✓ (RL) |

Table A2: Summary of gradient flow across modules. A checkmark indicates parameters updated under the corresponding losses.

## B.7 EVALUATION ON THE LEARNED REPRESENTATIONS

As illustrated in Fig. 4, we assess representation quality by measuring how well the learned latents predict the ground-truth simulator factors. Since the latent space can be over-parameterized, a single factor may be encoded across multiple coordinates. We therefore first fix all model parameters and only use the encoders to map observations to latent vectors $\mathbf{s}_t^i$, so that no further learning happens in the world models during this evaluation.

Given the assignment of latent dimensions to each ground-truth variable, we group the corresponding coordinates and attach a separate probe network to every group. Each probe is a two-layer MLP with hidden size 128, trained to regress all ground-truth factors associated with that group, and we report the coefficient of determination ($R^2$) between the predictions and the true values.

To train the probes, we randomly split the evaluation rollouts into two parts: 40% of the data is used to fit the MLPs and the remaining 60% is held out for computing the correlation metrics. The rollouts themselves are obtained by sampling random configurations of the underlying ground-truth latent factors. We use MSE loss for continuous variables and cross entropy loss for categorical variables.

After training the MLPs, we use the coefficient of determination ($R^2$) to determine the "matching quality" of learned representation and the ground-truth ones. The $R^2$ coefficient quantifies how well the predicted values $\hat{x}_i$ explain the variance of a ground-truth variable $x_i$. For continuous variables, it is defined as

$$R^2 = 1 - \frac{\sum_i (x_i - \hat{x}_i)^2}{\sum_i (x_i - \bar{x})^2}, \tag{A15}$$

where $\bar{x} = \frac{1}{N} \sum_i x_i$ denotes the empirical mean of the ground-truth values. Similarly, for categorical variables, we can use the most frequent category in the dataset to approximate $\bar{x}$, and then measure the distance by equal: $\delta_{x_i = \hat{x}_i}$.

## C    DETAILS ON TASK SETTINGS AND HYPERPARAMETERS

### C.1    TASK SETTINGS

We evaluate our framework across a diverse set of standard continuous control (locomotion) and robotic manipulation benchmarks. From DMControl (Tunyasuvunakool et al., 2020), we consider three representative locomotion tasks—*Cheetah Run*, *Walker*, and *Reacher*. To assess manipulation and generalization, we further include Meta-World (Yu et al., 2020), a widely used multi-task benchmark covering diverse object-centric skills, and extend to more realistic robotic platforms such as Franka-Kitchen (Gupta et al., 2019) and Robosuite (Zhu et al., 2020).

#### C.1.1    BENCHMARKS

**DMControl**    The DeepMind Control Suite (DMControl) (Tunyasuvunakool et al., 2020) is a standard continuous-control benchmark based on MuJoCo physics. We evaluate on three representative tasks:

- Cheetah-Run: a planar cheetah agent optimized for speed, testing fast locomotion and stability.
- Walker: a bipedal agent balancing and walking forward, emphasizing coordination.
- Reacher: a 2-link arm reaching for randomly placed targets, measuring precise control and sample efficiency.

| DMControl | Robosuite | Franka-kitchen | Meta World |
| --- | --- | --- | --- |

Figure A1: Visualization on tested benchmarks.

**Meta-World**    Meta-World (Yu et al., 2020) is a large multi-task benchmark for robotic manipulation. Each task involves a 7-DoF robotic arm manipulating the same object but with opposite goals, providing an ideal testbed for multi-task interference.

**Franka-kitchen**    The Franka-Kitchen benchmark (Gupta et al., 2019) simulates a realistic Franka Emika Panda robot arm interacting with common kitchen appliances (microwave, lights, burners, cabinets). Tasks are composed from multiple sub-goals, requiring long-horizon planning, object interaction, and compositional skill reuse.

**Robosuite**    Robosuite (Zhu et al., 2020) is a suite of MuJoCo-based robotic benchmarks designed for general-purpose robot learning. It provides a diverse set of manipulation tasks with realistic object dynamics and robot arms (including Sawyer and Panda).

#### C.1.2    SINGLE TASK LEARNING

We evaluate single-task learning across four suites. In **DMControl**, we use *Cheetah Run*, *Walker*, and *Reacher*. In **Meta-World**, we use *Box-Close*, *Lever-Pull*, *Assembly*, *Bin-Picking*, *Handle-Pull*, *Door-Unlock*, *Faucet-Close*, and *Drawer-Open*. In **RoboSuite**, we use *Lift* and *Pick & Place*. In **Franka Kitchen**, we use *Microwave*, *Kettle*, *Stove*, *Light*, and *Cabinet*.

We train for $10^6$ environment steps on DM Control, $5 \times 10^6$ on Meta-World, $5 \times 10^6$ on RoboSuite, and $5 \times 10^4$ on Franka Kitchen.

### C.1.3 GENERALIZATION TASKS

**Skill generalization.** For **RoboSuite**, we evaluate stacking tasks that can be composed from skills learned on *Lift* and *Pick & Place*. For **Franka Kitchen**, we transfer previously learned skills to long-horizon sequences using the extended setup from Chen et al. (2024). We consider two multi-task sequences: **MT-1**: *turn on microwave → move kettle → turn on stove → turn on light*; **MT-2**: *turn on microwave → turn on stove → turn on light → slide cabinet to the right*. We use $10^6$ adaptation steps for RoboSuite and $1.25 \times 10^5$ for Franka Kitchen.

**Compositional & unseen generalization.** For **Meta-World**, we test compositional generalization to *door-open* and *drawer-close*, and unseen-task generalization to *coffee-push*, *sweep-into*, *push-wall*, *reach-wall*, and *plate-slide* (randomly selected). All adaptations use $2.3 \times 10^5$ environment steps.

**Training**

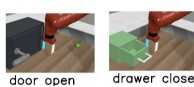

**Compositional Generalization**

**Unseen Tasks**

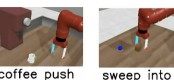 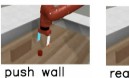 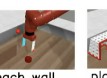 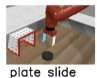

Figure A2: Visualization on the tasks used for compositional generalization and unseen tasks.

### C.2 ENVIRONMENT SETTINGS

**RoboSuite.** We design controlled multi-object scenes to encourage recovery of system variables and enable active interventions. Each episode initializes *three* cubes, with two cubes sharing identical color and texture. We construct a bank of ten environment variants that systematically vary a single physical factor at a time—(i) mass (body_mass), (ii) friction coefficients, and (iii) orientation components (e.g., Euler angles)—while keeping all other factors fixed. Within any episode, only *two* of the three cubes differ along the currently controlled factor. This parametrization yields clear intervention targets and facilitates learning an active intervention policy.

**Meta-World & Franka Kitchen.** Given the more limited low-level control in *Meta-World*, we adopt multi-object settings analogous to *Franka Kitchen*, which naturally contains diverse, manipulable objects. For Meta-World, we reuse the *Continual Bench* setup from Liu et al. (2025) to explore environments featuring drawers, doors, boxes, bins, and handles, in order to learn object-centric latent representations before downstream tasks. All baselines are granted the same phase-one interaction protocol and budget to ensure fairness. Unless otherwise specified, we allocate $10\%$ of the total environment steps for this representation-learning phase across suites.

### C.3 HYPERPARAMETERS

We build on the official *Dreamer-v3* codebase (including *DMControl*/MuJoCo tasks). For *Meta-World*, we adopt the Dreamer-v3 hyperparameters as specified in the *TD-MPC2* paper. For *Franka Kitchen* and *RoboSuite*, we use the same configuration as Meta-World due to task similarity. Baseline implementations follow their public releases wherever possible: *DINO-WM* is based on its official learning setup, but we replace the original MPC component with either PPO or SAC to match our environments; *TD-MPC2* is run with its original settings on *DMControl*. Beyond these, our method introduces phase-specific hyperparameters for (i) the exploration policy, (ii) MIST learning, and (iii) adaptation. The specification for (i) - (iii) is provided in Table A4-A5. The hyperparameters of SAC and PPO are specified in Table A6-A7.

### C.4 Details on Computational Requirements

All experiments are conducted on 4 NVIDIA A100 GPUs with 80GB memory. We report the training cost in GPU days and compare against the base model (Dreamer-v3). As shown in Table A3, our method requires slightly more compute while achieving better generalization.

| Task Suite | Ours | Dreamer-v3 |
|---|---|---|
| DM Control | 0.35 | 0.25 |
| Meta-World | 0.75 | 0.50 |
| Franka Kitchen | 0.35 | 0.30 |
| RoboSuite | 0.75 | 0.60 |

Table A3: Training cost (GPU days) comparison with Dreamer-v3 on different suites.

## D  Full Results

### D.1 Full Results on Representation Probing

Table A9 reports the full representation–probing results; the averaged scores appear in Fig. 4(c) of the main paper. The trends mirror our main findings. MIST-WM achieves the best overall performance. In the yellow block (exploration skills), METRA is consistently second-best. In the green block (world models), factored approaches (*I-Factor* and *Denoised MDP*) outperform DINO-WM on average. Notably, DINO-WM excels on the *size* factor, likely due to its strong pretrained visual prior.

### D.2 Full Results on Single Task Learning

Fig. A3 presents the complete single-task learning results, covering 8 tasks from Meta-World and 3 tasks from DMControl, complementing the summary shown in Fig. 5.

### D.3 Full Results on Generalization Tasks

Figs. A6-A6 show the complete results on skill generalization and on compositional/unseen tasks, respectively. The summary results are given in Fig. A4. We provide some visualized rollouts at https://sites.google.com/view/mist-wm-iclr.

### D.4 Full Results on Ablation Studies

Fig. A7 gives the full results on ablation studies, and the summary results are given in Fig. 7 in the main paper.

**Discussion: Choice of Skill-Learning Baselines.**  In the ablation studies, our primary goal is to examine how replacing our intervention-skill module with different unsupervised skill-learning methods affects overall performance. While the original experiments used METRA as a representative baseline, we have additionally included DIAYN following the reviewer's suggestion. Table A12 summarizes the performance across three representative settings (single-task, multi-task, and compositional generalization). We find that METRA generally outperforms DIAYN, both as a standalone skill-learning algorithm and when used as the skill-library initializer for MIST-WM. However, we emphasize that the choice between METRA and DIAYN is not central to our framework: both are only used to initialize the skill library, while the intervention skills are subsequently refined using our separability objective in Eq. 5. As a result, MIST-WM remains largely agnostic to the choice of initial skill-learning algorithm.

| Hyperparameter | Value |
|---|---|
| **General** | |
| Batch size | 128 (DMControl); 512 (Others) |
| Ratio of imagined samples | 0.5 (DMControl); 0.2 (Others) |
| Replay buffer size | $10^6$ (DMControl); $10^5$ (Others) |
| **Active Intervention** | |
| Learning rate | 0.0001 |
| Replay buffer size | $10^6$ (DMControl); $10^5$ (Others) |
| Contrastive encoder layers | 3 |
| Contrastive hidden units | 256 |
| Episodes per batch | 16 (DMControl); 32 (Others) |
| Discount factor $\gamma$ | 0.99 |
| Policy network layers | 2 |
| Policy network hidden units | 1024 |
| Gradients per episode | 50 (DMControl); 100 (Others) |
| MISL algorithm | DIAYN (DMControl); METRA (Others) |
| Target smoothing coef. | 0.995 |
| Entropy coef. | 0.01 |
| **World Model** | |
| Dreamer deterministic size | 512 for Meta World, 256 for Others |
| Dreamer stochastic size | 16 |
| Reward prediction layers | 2 |
| Reward pred. hidden units | 256 (DMControl); 512 (Others) |
| Policy network layers | 3 |
| Policy net hidden units | 256 (DMControl); 512 (Others) |
| Minimal score coef. | $10^{-3}$ (DMControl); $10^{-2}$ (Others) |
| Sufficiency score coef. | $10^{-3}$ (DMControl); $10^{-2}$ (Others) |

Table A4: Agent intervention exploration and world model hyperparameters for `MIST-WM`.

| Hyperparameter | Value |
|---|---|
| **Training** | |
| Training steps | $5\times10^6$ (MW, RS); $5\times10^4$ (Kitchen); $10^6$ (DMControl) |
| Maximize MINE after | 10% of total steps |
| **Adaptation** | |
| Adaptation steps | $2.3\times10^5$ (Meta-World) |
| Sparsity | 0.35 |
| Maximize MINE (start) | 0 |

Table A5: Training, adaptation, and curriculum hyperparameters for `MIST-WM`.

| Hyperparameter | Value |
|---|---|
| Batch size | 128 for Robsouite, 256 for Meta-World |
| Network architecture | MLPs |
| Actor / critic size | 3 fully connected layers with 256 units |
| Non-linearity | ReLU |
| Policy initialization | Standard Gaussian |
| Policy learning rate | $3 \times 10^{-4}$ |
| Optimizer | Adam |
| Adam $\beta$ (policy) | $(0.9, 0.999)$ |
| Adam $\beta$ (Q-function) | $(0.9, 0.999)$ |
| Discount | 0.99 |

Table A6: SAC hyperparameters for Meta-World and RoboSuite.

---

**Algorithm 1** MIST-WM

---

**Require:** Task set $\{T_i\}_{i=1}^N$, environments $\{\mathcal{E}_i\}_{i=1}^N$, world-model parameters $\theta$, policy parameters $\psi$, exploration policy $\pi_\varepsilon(\mathbf{a} \mid \mathbf{s}, \mathbf{z})$, skill prior $p_\eta(\mathbf{z})$, curriculum buffer $\{C_T(D_i)\}_{i=1}^N$

1: Initialize replay buffers $D_i \leftarrow \emptyset$ for all $i$
2: Initialize world model $\theta$, task-specific masks $\{m_i\}$, MI estimator, and policy $\psi$
3: Pretrain skill library (MISL) by maximizing Eq. 4 on randomly sampled environments to obtain $\pi_\varepsilon(\mathbf{a} \mid \mathbf{o}, \mathbf{z})$

**Outer loop over curriculum**
4: **for** training iteration $k = 1, 2, \dots$ **do**
5:     For each task $T_i$, compute difficulty proxy

$$C_T(D_i) \leftarrow \mathbb{E}_{(s,a,r)\sim D_i}\Big[\mathrm{Var}_m\big(f_\theta^{(m)}(\mathbf{s}, \mathbf{a})\big) - \log p_\theta(r \mid \mathbf{s}, \mathbf{a})\Big]$$

6:     Select next task by worst-case error: $i^\star \leftarrow \arg\max_i C_T(D_i)$
7:     Reset environment to $\mathcal{E}_{i^\star}$

**Active intervention and data collection on** $T_{i^\star}$
8:     **for** $e = 1$ to $N_{\mathrm{int}}$ **do**
9:         Sample a skill $\mathbf{z} \sim p_\eta(\mathbf{z})$
10:        Roll out skill-conditioned exploration policy $\mathbf{a}_t \sim \pi_\varepsilon(\mathbf{a} \mid \mathbf{s}_t, \mathbf{z})$ for $H$ steps in $\mathcal{E}_{i^\star}$
11:        Collect transitions $(\mathbf{o}_t, \mathbf{a}_t, r_t, \mathbf{o}_{t+1})_{t=0}^{H-1}$ and compute its contrastive score according to Eq. 5
12:        Update $\pi_\epsilon$ using Eq. 4
13:     **end for**
14:     Update $\pi_\epsilon$ using Eq. 5 on collected segments to favor skills with high information gain
15:     With high probability $p$, store segments among the top-$k$ scores into $D_{i^\star}$, and $1 - p$ to store the remaining segments

**World model and policy learning with MIST objectives on** $T_{i^\star}$
16:     **for** $u = 1$ to $N_{\mathrm{wm}}$ **do**
17:        Sample a minibatch of sequences from $D_{i^\star}$
18:        Encode $\mathbf{o}_{0:T}^{i^\star}$ and $\mathbf{a}_{0:T}^{i^\star}$ with $p_\theta$ to obtain $\mathbf{s}_{0:T}^{i^\star}$ and recurrent states $\mathbf{h}_{0:T}^{i^\star}$
19:        Compute Dreamer-based loss $\mathcal{L}_{\mathrm{WM}}^{(i^\star)}(\theta)$ as in Eq. 1
20:        Compute task-specific sufficiency score

$$R_{\mathrm{suff}}^{(i^\star)}(\theta) = \mathbb{E}\Big[ \sum_{t=0}^{T-1} \log p_\theta(r_t^{i^\star} \mid m_{i^\star} \odot \mathbf{s}_t^{i^\star}, \mathbf{a}_t)\Big]$$

21:        Estimate mutual information term $MI(m_{i^\star} \odot \mathbf{s}_{1:T}^{i^\star}, g_{i^\star})$ using MINE and minimality score
$$R_{\mathrm{min}}^{(i^\star)}(\theta) = MI(m_{i^\star} \odot \mathbf{s}_{1:T}^{i^\star}, g_{i^\star}) - \lambda_M \|m_{i^\star}\|_1$$
22:        Update $\theta$, $m_{i^\star}$, and MI-estimator parameters by minimizing

$$\mathcal{L}_{\mathrm{total}}^{(i^\star)} = \mathcal{L}_{\mathrm{WM}}^{(i^\star)} - \alpha_{\mathrm{suff}} R_{\mathrm{suff}}^{(i^\star)} - \alpha_{\mathrm{min}} R_{\mathrm{min}}^{(i^\star)}$$

23:     **end for**
24:     **for** $u = 1$ to $N_{\mathrm{rl}}$ **do**
25:        Sample trajectories from $D_{i^\star}$ and encode them to latent states $\mathbf{s}_t^{i^\star}$
26:        Form task-specific MIST representation $\tilde{\mathbf{s}}_t^{i^\star} = m_{i^\star} \odot \mathbf{s}_t^{i^\star}$
27:        Update policy $\pi_\psi(\mathbf{a} \mid \tilde{\mathbf{s}}_t^{i^\star})$ to maximize $J_{\mathrm{RL}}^{(i^\star)}(\psi) = \mathbb{E}\big[\sum_{t=0}^{T-1} \gamma^t r_t^{i^\star}\big]$
28:        Roll out the updated policy $\pi_\psi$ in $\mathcal{E}_{i^\star}$ for $H$ steps to obtain new trajectories $\tau_{\mathrm{new}}$
29:        Add $\tau_{\mathrm{new}}$ into $D_{i^\star}$ and update $\theta$ using minibatches sampled from $D_{i^\star} \cup \tau_{\mathrm{new}}$
30:     **end for**

**Update curriculum statistics**
31:     Recompute $C_T(D_{i^\star})$ with the updated $\theta$
32: **end for**

---

| Parameter | Value |
|---|---|
| Network architecture | MLPs |
| Actor / critic size | $3 \times 256$ units for DMC, $3 \times 400$ units for Kitchen |
| Non-linearity | ReLU |
| Observation normalization | Yes |
| Reward normalization | Yes |
| Reward clipping (stddev.) | 10 |
| Batch size | 128 |
| Policy trust region | 0.2 |
| Value trust region | No |
| Advantage normalization | Yes |
| Entropy penalty scale | 0.01 |
| Discount factor | 0.997 |
| GAE $\lambda$ | 0.95 |
| Learning rate | $3 \times 10^{-4}$ |
| Gradient clipping | 0.5 |
| Adam $\varepsilon$ | $10^{-5}$ |

Table A7: PPO hyperparameters for DMC and Kitchen.

| Hyperparameter | Value |
|---|---|
| Temperature $\eta$ | 0.1 |
| Domain randomization prob. $p_{DR}$ | 0.3 |
| Normalization constant $\varepsilon$ | $10^{-6}$ |
| Ensemble size $M$ | 5 |

Table A8: Curriculum learning hyperparameters.

| Methods | Size | C-pos | C-Orient | G-pos | G-orient | G-openness | C-Mass | Average |
|---|---|---|---|---|---|---|---|---|
| Random | 0.01 | 0.92 | 0.16 | 0.48 | 0.21 | 0.06 | 0.03 | 0.31 |
| METRA | 0.03 | 0.95 | 0.85 | 0.51 | 0.49 | 0.20 | 0.55 | 0.59 |
| Curiosity Replay | 0.02 | 0.96 | 0.53 | 0.23 | 0.36 | 0.15 | 0.39 | 0.44 |
| Plan2Explore | 0.01 | 0.93 | 0.59 | 0.41 | 0.26 | 0.10 | 0.40 | 0.45 |
| DINO-WM | **0.76** | 0.75 | 0.09 | 0.53 | 0.42 | 0.00 | 0.61 | 0.40 |
| I-Factor | 0.00 | 0.79 | 0.41 | 0.56 | 0.47 | 0.08 | 0.59 | 0.48 |
| Denoised MDP | 0.00 | 0.83 | 0.30 | 0.72 | 0.65 | 0.16 | 0.68 | 0.56 |
| MIST-WM | 0.02 | **1.00** | **0.94** | **0.86** | **0.69** | **0.21** | **0.96** | **0.78** |

Table A9: **Probing results ($R^2$) under different settings**. The pink region corresponds to varying the exploration data using different collection methods, while the green region corresponds to varying the world-modeling methods while keeping the data fixed. The average is computed across all factors excluding the size.

| Environment | Ours | DINO-WM | Factored Dreamer | Dreamer-v3 | TD-MPC2 |
|---|---|---|---|---|---|
| Box-close | $\mathbf{0.94} \pm 0.02$ | $0.75 \pm 0.05$ | $0.15 \pm 0.06$ | $0.26 \pm 0.13$ | $0.09 \pm 0.02$ |
| Lever-pull | $\mathbf{0.91} \pm 0.05$ | $0.72 \pm 0.04$ | $0.38 \pm 0.08$ | $0.29 \pm 0.11$ | $0.06 \pm 0.03$ |
| Assembly | $\mathbf{0.79} \pm 0.12$ | $0.61 \pm 0.03$ | $0.06 \pm 0.02$ | $0.22 \pm 0.08$ | $0.12 \pm 0.06$ |
| Bin-picking | $\mathbf{0.88} \pm 0.05$ | $0.71 \pm 0.04$ | $0.10 \pm 0.06$ | $0.31 \pm 0.12$ | $0.08 \pm 0.03$ |
| Handle-pull | $\mathbf{0.84} \pm 0.06$ | $0.80 \pm 0.02$ | $0.61 \pm 0.09$ | $0.53 \pm 0.08$ | $0.15 \pm 0.04$ |
| Door-unlock | $0.96 \pm 0.02$ | $0.91 \pm 0.04$ | $0.55 \pm 0.08$ | $\mathbf{0.98} \pm 0.05$ | $0.13 \pm 0.02$ |
| Faucet-close | $\mathbf{0.91} \pm 0.06$ | $0.68 \pm 0.08$ | $0.23 \pm 0.12$ | $0.18 \pm 0.11$ | $0.10 \pm 0.04$ |
| Draw-open | $\mathbf{0.97} \pm 0.02$ | $0.86 \pm 0.04$ | $0.62 \pm 0.09$ | $0.54 \pm 0.08$ | $0.17 \pm 0.05$ |

Table A10: Success rate at 2M environment steps (mean $\pm$ std over 5 seeds).

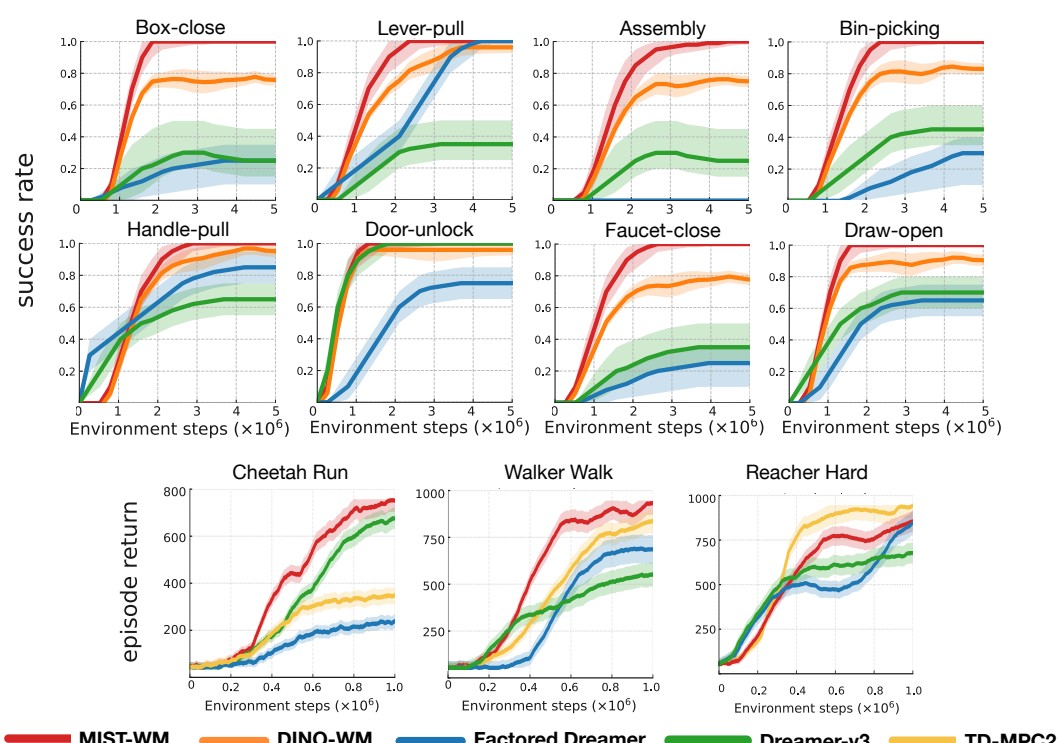

Figure A3: Results (across 5 seeds) on single-task learning, including **robotic manipulation** tasks (Meta World) and **locomotion** (DMControl).

| Environment | Ours | DINO-WM | Factored Dreamer | Dreamer-v3 | TD-MPC2 |
|---|---|---|---|---|---|
| Cheetah Run | $\mathbf{731.6}_{\pm 32.9}$ | $165.2_{\pm 41.8}$ | $217.5_{\pm 16.4}$ | $661.0_{\pm 42.3}$ | $339.8_{\pm 31.6}$ |
| Walker Walk | $\mathbf{904.2}_{\pm 42.5}$ | $286.5_{\pm 37.4}$ | $557.2_{\pm 56.4}$ | $698.2_{\pm 52.5}$ | $867.5_{\pm 51.2}$ |
| Reacher Hard | $871.6_{\pm 38.5}$ | $346.2_{\pm 29.8}$ | $869.3_{\pm 33.9}$ | $702.4_{\pm 43.3}$ | $\mathbf{915.6}_{\pm 23.3}$ |

Table A11: DMControl performance at 1M steps (mean $\pm$ std).

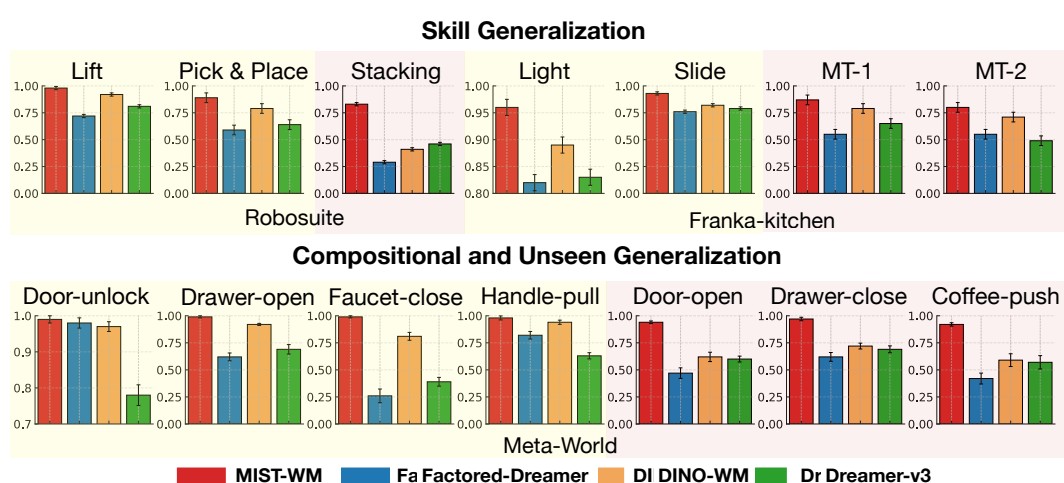

Figure A4: Success rate (across 5 seeds) on skill and compositional generalization, including tasks in Robosuite, Franka-kitchen, and Meta-World. Results in light yellow boxes are those on source tasks, and light pink are those on generalization tasks. Error bars indicate the standard deviation.

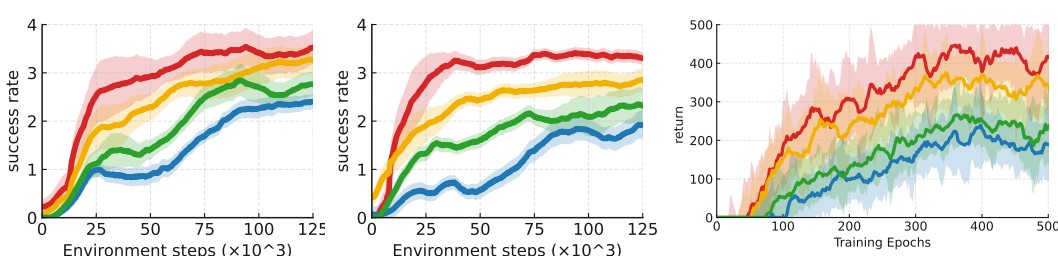

Figure A5: Results (across 5 seeds) on skill generalization tasks, where the left two figures are the skill generalization curves on Franka Kitchen MT-1 and MT-2; and the right figure is the generalization results on robosuite-stacking. The yellow curve indicates the results using DINO-WM.

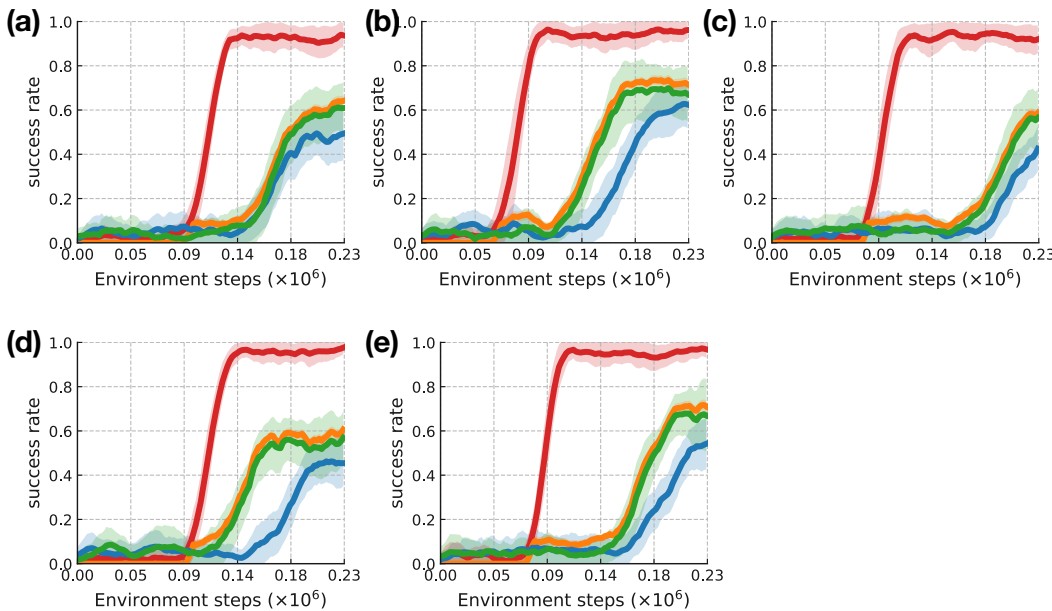

Figure A6: Results (across 5 seeds) on unseen tasks in Meta-World, including (a) push-wall; (b) reach-wall; (c) coffee-push; (d) sweep-into; and (e)plate-slide.

| Environment | Ours (METRA) | Ours (DIAYN) | METRA | DIAYN | Curious Replay |
|---|---|---|---|---|---|
| Cheetah | **741.0 ± 25.6** | 731.6 ± 32.9 | 680.8 ± 32.3 | 664.5 ± 39.6 | 635.8 ± 46.8 |
| Box-close | **0.94 ± 0.02** | 0.79 ± 0.04 | 0.81 ± 0.06 | 0.59 ± 0.10 | 0.48 ± 0.12 |
| Coffee-push | **0.89 ± 0.05** | 0.82 ± 0.03 | 0.76 ± 0.07 | 0.62 ± 0.05 | 0.30 ± 0.16 |

Table A12: Ablation comparing METRA and DIAYN as skill-library initializers.

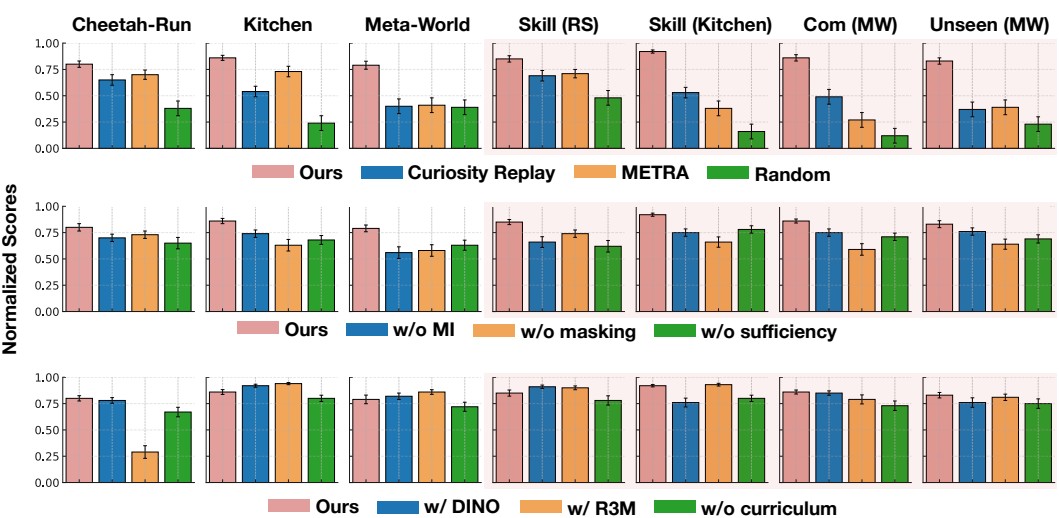

Figure A7: **Ablation studies. Row 1—Exploration policy:** compare our skill-conditioned exploration against *Random*, *Intrinsic motivation* (Curiosity Replay), and *MISL* (METRA). **Row 2—Structure-learning terms:** remove the MI term in the *minimality* score (w/o MI), remove *masking* in the minimality score (w/o Mask), and remove the *sufficiency* term (w/o Suff). **Row 3—Backbones and curriculum:** replace our backbone with *DINO-WM* or *R3M*, and evaluate w/o curriculum. Error bars indicate the standard deviation.

# E    LIMITATIONS AND FUTURE WORKS

Several limitations remain. Most notably, our experiments are confined to simulated domains, and extending the framework to real-world systems such as robotics platforms is an important direction for future work. In addition, the current framework is more centralized in the RL domain. However, the idea that an agent should continually seek minimal and sufficient information may also provide insights for self-improving, open-ended learning in other domains, such as LLM-based agents for scientific discovery (Piriyakulkij et al., 2024; Zhang et al., 2025; Lu et al., 2024), an exciting direction for future work.

# F    LLM USAGE STATEMENT

LLMs were used only to correct grammatical issues in this paper. No part of the research ideas, experimental design, implementation, or empirical analysis relied on LLMs.

