# OpenReview forum: "Learning Task-Sufficient World Models via Intervention-Curriculum Co-Design"
_ICLR.cc/2026/Conference — Submitted to ICLR 2026_

### Official Review · Reviewer_HNgN · 2025-10-30

**Soundness:** 3
**Presentation:** 2
**Contribution:** 3
**Rating:** 4
**Confidence:** 3

**Summary:**

This work proposes a framework to learn a world model. Specifically, a two-stage approach is proposed. Firstly, on the representation side, this work proposes to learn Minimal, Sufficient, and Task-specific (MIST) states for the world model by scoring candidates by minimality and sufficiency and optimising for soft masking and mutual information. Secondly, while exploring new environments, this work proposes to use curriculum learning to learn how to order each skill in the new environment for efficient world model learning.

**Strengths:**

- The proposed formulation of the MIST states is novel to my knowledge.
- Using curriculum to guide/order environments for the agent to learn is intuitive.
- Conducted comprehensive experiments, including ablations, showing the effectiveness of the proposed method.
- I appreciate the authors denoting their computational requirements in the appendix.

**Weaknesses:**

- For me, the major weakness of this work is that there are no related works in the main manuscript (but were placed in the appendix). Technically it can mean that there are no related works section in this work.
- Some design choices are less motivated, specifically the skill library part, and is basically a deployment of existing works.
- Not technically a weakness but, to my understanding, the proposed method is somewhat more complex to train than comparative methods (i.e. TD-MPC2 and Dreamerv3).
- Minor issues:
  - Duplicated phrases around lines 320-325: “Following Unsupervised Experiment Design (UED) (Dennis et al., 2020; Rigter et al., 2024), we select”....
  - Small grammar issues, for example line 127: “the state transitions is” -> “the state transitions are”
  - In Figure 5, not all algorithms are evaluated on all the proposed tasks.
  - In Figure 6, the bars are not labelled.
  - In Figure 7, I assume the pink bar is the proposed MIST-WM but it is not noted. Also, DIAYN is not compared as an ablation.

**Questions:**

I will repost some suggestions/questions that were mentioned in the weakness section here for clarity.

$$\textbf{Suggestions}$$
S1: Somehow wiggle the related works section into the main 9 pages.
S2: List the PPO/SAC hyperparameters used for downstream policy learning.
S3: Add all the experiment results in Figure 5 for all algorithms.
S4: Add clear captions/descriptions to Figure 6 and Figure 7.

$$\textbf{Questions}$$
Q1: I find it interesting that the authors chose not to propagate the MIST states to the skill library agent, since the authors argue the MIST states are better representations, I’d imagine that MIST states would help learn the skill library. Are there any particular reasons behind this choice?
Q2: I have some reservation regarding the effectiveness of the two stages individually. Reading Figure 7, it seems to be that the curriculum learning only helps a bit. What do the authors think about the results?
Q3: Following up on Q2, I wonder what the performance of standalone MIST states would be? It seems that replacing proposed MIST states with R3M would actually have better performance. (I might have misunderstood the meaning of “replacing the backbone with …”).
Q4: I’m a bit confused about the meaning of “task embeddings g_i can be the rewards” which were written several times across the manuscript.
Q5: Are there any particular reasons to use two different algorithms (DIAYN & METRA) to learn the skill library? Since the ablation showed that curious replay is also good while DIAYN was not investigated, why not simply just use METRA?

---

> ### Author Response · Authors · 2025-11-19
> **Author Response (1/5)**
>
> We thank you for your constructive and insightful comments, as well as for the concrete suggestions. We have incorporated them into our revisions and use this rebuttal to address each point in detail. We also appreciate your positive feedback on the novelty and evaluation.
>
>
> ---
>
>
> > **W1** For me, the major weakness of this work is that there are no related works in the main manuscript (but were placed in the appendix). Technically it can mean that there are no related works section in this work.
>
>
> > **S1** Somehow wiggle the related works section into the main 9 pages.
>
>
> [**Response**] Thank you for the suggestion. In the revised version, we have moved the Related Work section into the main paper (the revision now allows one additional page).
>
>
> [**Related Revision**] `Sec 4`
>
>
> ---
>
>
> > **W2** Some design choices are less motivated, specifically the skill library part, and is basically a deployment of existing works.
>
>
> [**Response**] We respectfully disagree with this point. Our work is centered around a single, principled goal: learning task-specific minimal and sufficient world models. Achieving such representations inherently requires the synergy of (i) **the agent’s interaction strategy**, (ii) the **environment / task schedule**, and (iii) the **structure-aware world-model objective**. Each component has a distinct and necessary role: active interventions generate informative trajectories that reveal task-relevant factors; the adaptive curriculum exposes these factors in a progressively learnable order; and the structure-learning objective extracts a minimal, task-sufficient subspace from the Dreamer latents. These three parts form a coherent, closed loop rather than a collection of loosely combined modules.
>
>
> Specifically, regarding other novelty of skill library and active intervention part, we go beyond standard MISL-style exploration by learning skills that maximize segment separability with respect to latent factors of the world model (`Eq. (5)`), i.e., skills are selected for how well they induce distinguishable observation trajectories, not merely for unsupervised discovery with MISL objectives.
>
>
> In summary, the guiding principle is **learn task-specific minimal and sufficient world models**, while the concrete realization naturally decomposes into interacting modules (active interventions, structured world-model learning, and curriculum). These are not independent components from existing works, but complementary mechanisms required to implement this principle in practice.
>
>
> ---

---

> ### Author Response · Authors · 2025-11-19
> **Author Response (2/5)**
>
> > **W3** Not technically a weakness but, to my understanding, the proposed method is somewhat more complex to train than comparative methods (i.e. TD-MPC2 and Dreamerv3).
>
> [**Response**] We acknowledge that our method has more modules than a pure Dreamer or TD-MPC2 setup, as it is designed to jointly learn MIST representations through the synergy between the world model, intervention skills, and curriculum. However, we would like to emphasize two points.
>
> (1) **Algorithmic complexity.** The framework is modular rather than intrinsically complicated. Each stage is trained with a relatively simple objective while the remaining components are kept frozen. We make this explicit in the revised `Algorithm 1` and in `Appendix B.6` (`Table A.2`), which describe the gradient and data flow. For convenience, we reproduce the gradient summary below:
>
> | Module                     | WM loss updated? | Structure losses updated? | Skill / RL losses updated? |
> |----------------------------|------------------|---------------------------|----------------------------|
> | Dreamer encoder / RSSM     | ✓                | frozen                    | frozen                     |
> | Reward / obs decoders      | ✓                | ✓                         | frozen                     |
> | Mask $m^i$               | frozen           | ✓                         | frozen                     |
> | MI estimator (MINE)        | frozen           | ✓                         | frozen                     |
> | Skill policy $\varepsilon$        | frozen           | frozen                    | ✓ (skill loss)             |
> | Skill discriminator $\phi$ | frozen           | frozen                    | ✓ (skill loss)             |
> | Segment encoder $\xi$ | frozen           | frozen                    | ✓ (skill loss)             |
> | Policy / value $psi$   | frozen           | frozen                    | ✓ (RL loss)                |
>
>
> (2) **Computational overhead.** The additional training cost from structure learning, skill learning, and curriculum is moderate. On 4× NVIDIA A100 GPUs (80GB), the total GPU days are:
>
>
> | Task Suite      | Ours | Dreamer-v3 |
> |-----------------|------|------------|
> | DM Control      | 0.35 | 0.25       |
> | Meta-World      | 0.75 | 0.50       |
> | Franka Kitchen  | 0.35 | 0.30       |
> | RoboSuite       | 0.75 | 0.60       |
>
>
> Thus, while MIST-WM adds some complexity to enable minimal and sufficient task-specific representations, both the optimization pipeline and the computational cost remain manageable.
>
>
> [**Related Revision**] `Algorithm 1` (`Page 25`); `Appendix B.6` (`Page 19`); `Appendix C.4`
>
>
> ---
>
>
> > **W4** Minor issues:
>
>
> >> **W4-1** Duplicated phrases around lines 320-325
>
>
> >> **W4-2** Small grammar issues
>
>
> [**Response**] Thank you for catching these. We have corrected these accordingly in the revision.
>
> >> **W4-3** In Figure 5, not all algorithms are evaluated on all the proposed tasks / **S3**: Add all the experiment results in Figure 5 for all algorithms.
>
> [**Response**] In the main paper, Fig. 5 was originally intended to highlight single-task sample efficiency, so we only showed baselines that achieved reasonably strong performance. Consequently, for locomotion we omitted DINO-WM, and for Meta-World we omitted TD-MPC2, since their results were much lower. Following your suggestion, we now include the full results in the tables below and have added them to the appendix with pointers from the main text.
>
> Meta-World (Success rate, 2M steps, 5 seeds)
> | Environment   | Ours | DINO-WM | Factored Dreamer | Dreamer-v3 | TD-MPC2 |
> |---------------|-------|---------|------------------|------------|---------|
> | Box-close     | **0.94 ± 0.02** | 0.75 ± 0.05 | 0.15 ± 0.06 | 0.26 ± 0.13 | 0.09 ± 0.02 |
> | Lever-pull    | **0.91 ± 0.05** | 0.72 ± 0.04 | 0.38 ± 0.08 | 0.29 ± 0.11 | 0.06 ± 0.03 |
> | Assembly      | **0.79 ± 0.12** | 0.61 ± 0.03 | 0.06 ± 0.02 | 0.22 ± 0.08 | 0.12 ± 0.06 |
> | Bin-picking   | **0.88 ± 0.05** | 0.71 ± 0.04 | 0.10 ± 0.06 | 0.31 ± 0.12 | 0.08 ± 0.03 |
> | Handle-pull   | **0.84 ± 0.06** | 0.80 ± 0.02 | 0.61 ± 0.09 | 0.53 ± 0.08 | 0.15 ± 0.04 |
> | Door-unlock   | 0.96 ± 0.02     | 0.91 ± 0.04 | 0.55 ± 0.08 | **0.98 ± 0.05** | 0.13 ± 0.02 |
> | Faucet-close  | **0.91 ± 0.06** | 0.68 ± 0.08 | 0.23 ± 0.12 | 0.18 ± 0.11 | 0.10 ± 0.04 |
> | Drawer-open   | **0.97 ± 0.02** | 0.86 ± 0.04 | 0.62 ± 0.09 | 0.54 ± 0.08 | 0.17 ± 0.05 |
>
>
> DMC (Return, 1M steps, 5 seeds)
> | Environment   | Ours | DINO-WM | Factored Dreamer | Dreamer-v3 | TD-MPC2 |
> |---------------|-------|---------|------------------|------------|---------|
> | Cheetah Run   | **731.6 ± 32.9** | 165.2 ± 41.8 | 217.5 ± 16.4 | 661.0 ± 42.3 | 339.8 ± 31.6 |
> | Walker Walk   | **904.2 ± 42.5** | 286.5 ± 37.4 | 557.2 ± 56.4 | 698.2 ± 52.5 | 867.5 ± 51.2 |
> | Reacher Hard  | 871.6 ± 38.5 | 346.2 ± 29.8 | 869.3 ± 33.9 | 702.4 ± 43.3 | **915.6 ± 23.3** |
>
>
> [**Related Revision**] `Table A10-A11` (`Pages 26-27`)

---

> ### Author Response · Authors · 2025-11-19
> **Author Response (3/5)**
>
> >> **W4-4** In Figure 6, the bars are not labelled. / **S4**: Add clear captions/descriptions to Figure 6 and Figure 7.
>
>
> [**Response**] We apologize for the confusion. Both are corrected in the revision.
>
>
> >> **W4-5** In Figure 7, I assume the pink bar is the proposed MIST-WM but it is not noted. Also, DIAYN is not compared as an ablation. / **Q5** Are there any particular reasons to use two different algorithms (DIAYN & METRA) to learn the skill library? Since the ablation showed that curious replay is also good while DIAYN was not investigated, why not simply just use METRA?
>
>
> [**Response**] Thank you for pointing this out. We have now explicitly annotated Fig. 7 to indicate that the pink bar corresponds to MIST-WM.
>
> Regarding DIAYN, our original ablations focused on replacing our intervention-skill module with representative unsupervised skill-learning baselines, and we selected METRA as a strong representative. Following your suggestion, we have now included DIAYN as well. The full ablation results are shown below:
>
> | Environment   | Ours (METRA) | Ours (DIAYN) | METRA | DIAYN | Curious Replay |
> |---------------|-------|---------|------------------|------------|---------|
> | Cheetah   | **741.0 ± 25.6** | 731.6 ± 32.9 | 680.8 ± 32.3 | 664.5 ± 39.6 | 635.8 ± 46.8 |
> | Box-close (Single)   | **0.94 ± 0.02** | 0.79 ± 0.04 | 0.81 ± 0.06 | 0.59 ± 0.10 |0.48 ± 0.12|
> | Coffee-push (Gen)  | **0.89 ± 0.05** | 0.82 ± 0.03 | 0.76 ± 0.07 | 0.62 ± 0.05 | 0.30 ± 0.16 |
>
> We observe that METRA performs slightly better than DIAYN, both when used standalone and when used as a skill-library initializer within our framework. However, we emphasize that the choice of METRA vs. DIAYN is not central to MIST-WM. Both are only used to initialize the skill library, while our intervention-skill module is optimized using the separability objective in Eq. (5). Thus, the system is empirically agnostic to which initialization is used. We include this comparison and discussion in the revised `Appendix D.4`.
>
> [**Related Revision**] `Appendix D.4` (`Page 23`)
>
> ---
>
> > **S2**: List the PPO/SAC hyperparameters used for downstream policy learning.
>
> [**Response**] Thank you for pointing this out. We provide the complete hyperparameters used for downstream policy learning with SAC (Meta-World and RoboSuite) and PPO (DMControl and Kitchen) below. These settings are fixed across all world-model baselines.
>
> **SAC**
> | Hyperparameter      | Value            |
> |-----------------------------|-------------------------------------------------|
> | Batch size           | 128 (RoboSuite), 256 (Meta-World)               |
> | Network architecture        | MLPs                                            |
> | Actor / critic size         | 3 fully connected layers × 256 units            |
> | Non-linearity     | ReLU                                            |
> | Policy initialization       | Standard Gaussian                               |
> | Policy learning rate        | 3 × 10⁻⁴                                        |
> | Optimizer                   | Adam                                            |
> | Adam β (policy)            | (0.9, 0.999)                                    |
> | Adam β (Q-function)        | (0.9, 0.999)                                    |
> | Discount                    | 0.99                                            |
>
>
> **PPO**
> | Hyperparameter              | Value        |
> |-----------------------------|-----|
> | Network architecture        | MLPs                                            |
> | Actor / critic size         | 3 × 256 units (DMC), 3 × 400 units (Kitchen)    |
> | Non-linearity               | ReLU                                            |
> | Observation normalization   | Yes    |
> | Reward normalization        | Yes       |
> | Reward clipping (stddev)    | 10                                              |
> | Batch size                  | 128                                             |
> | Policy trust region         | 0.2                                             |
> | Value trust region          | No                                              |
> | Advantage normalization     | Yes                                             |
> | Entropy penalty             | 0.01                                            |
> | Discount factor             | 0.997                                           |
> | GAE $\lambda$                     | 0.95                                            |
> | Learning rate               | 3 × 10⁻⁴                                        |
> | Gradient clipping           | 0.5                                             |
>
> [**Related Revision**] `Table A.6-A.7` (`Pages 24; 26`).
>
> ---

---

> ### Author Response · Authors · 2025-11-19
> **Author Response (4/5)**
>
> > **Q1**: I find it interesting that the authors chose not to propagate the MIST states to the skill library agent, since the authors argue the MIST states are better representations, I’d imagine that MIST states would help learn the skill library. Are there any particular reasons behind this choice?
>
> [**Response**] Yes, this is mainly due to the *task-specific* nature of MIST states. For each task, only a small subset of latent dimensions is selected as MIST, while the remaining dimensions may still be useful for discovering new behaviors. In contrast, the skill-learning stage is explicitly *task-agnostic*: it does not use task rewards or task descriptors, but only MISL-style and contrastive objectives to seek diverse, separable trajectories.
>
> If we were to feed only MIST states into the skill learner, then for each task the learned skills would also become task-specific, which is undesirable for discovering new factors in future tasks and for open-ended exploration. By using the full latent state instead, skills can probe and separate both currently known and yet-unidentified factors, and MIST learning can subsequently select the task-relevant subset.
>
> To validate this design choice, we compared using the full latent state vs. using MIST states as input to the skill policy in a simplified setting:
>
> | Environment           | Ours (full latent) | Ours (skill w.r.t. MIST) |
> |-----------------------|--------------------|--------------------------|
> | Box-close (Single)    | **0.94 ± 0.02**    | 0.91 ± 0.05              |
> | Assembly (Single)     | 0.79 ± 0.12        | **0.83 ± 0.06**          |
> | Coffee-push (Gen)     | **0.89 ± 0.05**    | 0.50 ± 0.23              |
>
> We observe that in single-task settings, propagating MIST to the skill learner can sometimes help (the skills become more aligned with task-relevant factors), but for unseen generalization it substantially hurts performance. This supports our design choice: keeping skills task-agnostic (using the full latent) is more beneficial when the goal is to discover and reuse representations across tasks.
>
> ---
>
> > **Q2**: I have some reservation regarding the effectiveness of the two stages individually. Reading Figure 7, it seems to be that the curriculum learning only helps a bit. What do the authors think about the results?
>
> [**Response**] We agree that the isolated gain from the curriculum module in Fig. 7 is more modest than that of the intervention skills or structure learning when each component is ablated separately. However, our goal is not for each stage to be equally strong on its own, but for the three to work **synergistically** to realize task-specific minimal and sufficient world models. The curriculum in particular controls **which tasks are exposed when**, so its effect is more about stabilizing and organizing the learning process than producing large standalone performance jumps.
>
>
> Empirically, the curriculum consistently brings improvements over the no-curriculum variants, indicating that selecting tasks in a structured order does help the agent discover and refine MIST states more reliably. Conceptually, we view this as an important ingredient for principled world-model learning in open-ended settings: even if its marginal effect appears smaller in a controlled ablation, it plays a key role in making the overall training more natural and data-efficient.
>
> ---
>
> > **Q3**: Following up on Q2, I wonder what the performance of standalone MIST states would be? It seems that replacing proposed MIST states with R3M would actually have better performance. (I might have misunderstood the meaning of “replacing the backbone with …”).
>
>
> [**Response**] Sorry for the confusion. In those ablations, we are **not** comparing against the original DINO/R3M control pipelines; instead, we plug their encoders into our framework in place of Dreamer’s encoder, while keeping the rest of MIST-WM unchanged. These results are intended to demonstrate that our structure-learning mechanism is compatible with multiple perceptual backbones and that performance remains comparable across encoders, which also helps justify our choice of Dreamer as the main backbone.
>
>
> [**Related Revision**] We added the explanation in `Sec 5` (`Line 516`, `Page 10`).
>
> ---

---

> ### Author Response · Authors · 2025-11-19
> **Author Response (5/5)**
>
> > **Q4**: I’m a bit confused about the meaning of “task embeddings g_i can be the rewards” which were written several times across the manuscript.
>
>
> [**Response**] Thank you for raising this point, and we apologize for the confusion in our earlier phrasing. Throughout the paper, $g_i$ denotes the task descriptor used in the structure-learning objective to identify task-specific latent factors. When explicit task information is available (e.g., language instructions or task IDs in Meta-World), we use that as $g_i$.  However, in environments where no such auxiliary information is provided (e.g., DMControl), we substitute the rewards as the task descriptor, since it reflects the task-specific signal that distinguishes one task from another. We have clarified this explicitly in the revised manuscript to avoid ambiguity.
>
>
> [**Related Revision**] We added the explanation in `Sec 3` (`Lines 227-228`, `Page 5`).
>
> ---
>
> > **Q5**: Are there any particular reasons to use two different algorithms (DIAYN & METRA) to learn the skill library? Since the ablation showed that curious replay is also good while DIAYN was not investigated, why not simply just use METRA?
>
> Please kindly refer to the response to **W4-5**.
>
> ---
>
> Thank you again for your thoughtful questions, comments, and suggestions. They have been very helpful for improving the clarity and presentation of the manuscript. We hope that the revisions addressing W1 and W4 make the overall exposition clearer, and that the changes related to W2–W3 and your detailed questions render the technical components more transparent and accessible. If you have any further questions or suggestions, we would be very happy to address them.

---

> > ### Author Response · Authors · 2025-11-26
> > **Thank you for your thoughtful review**
> >
> > Dear Reviewer HNgN,
> >
> > Thank you again for your thoughtful questions, comments, and suggestions. All of them have been very helpful for improving the clarity and presentation of the manuscript. We hope that the revisions addressing W1 and W4 make the overall exposition clearer, and that the changes related to W2–W3 and your detailed questions make the technical components more transparent and accessible. If you have any further questions or suggestions, we would be very happy to address them.
> >
> > The Authors

---

### Official Review · Reviewer_qDyg · 2025-10-31

**Soundness:** 2
**Presentation:** 1
**Contribution:** 3
**Rating:** 2
**Confidence:** 4

**Summary:**

The paper proposes MIST-WM, a framework that combines: (i) skill-based learning for exploration using DIAYN/MISL and a separability heuristic for skill selection, (ii) a Dreamer-style world model whose latent space is masked to yield "Minimal, Sufficient, Task-specific (MIST)" states via masking and a reward-likelihood (sufficiency) and MI-based minimality objectives; and (iii) an adaptive curriculum over environments/tasks. Experiments on DMControl, RoboSuite, Franka-Kitchen, and Meta-World report gains in sample efficiency and several generalization settings.

**Strengths:**

- Ambitious scope and goal: learning task-sufficient latents and coupling representation learning with active data collection and curriculum.

- Interesting components:

	- Masked latent subspaces with sufficiency and minimality pressures.

	- A pragmatic separability-based heuristic to pick informative skills for data collection.

- Some empirical gains over strong world-model baselines on multiple benchmarks.

**Weaknesses:**

- Unfocused and incremental: The method is a collection of several incremental ideas (skill selection, MIST masking, curriculum) without a clear, central algorithmic idea or hypothesis. It consists of several known ingredients (DIAYN/MISL-style skills, Dreamer-v3 backbone with masking/MI, UED-like curriculum). The novelty feels diffuse rather than centered on one idea. Even with ablations, it remains unclear which piece drives which improvement and why.

- Missing analysis of mechanisms: Given the system complexity, the paper does not convincingly explain when and why each component helps. There is plenty of space wasted to simply discribe the performance plots (e.g. in "Sample efficient policy learning") instead of interpreting the results.

- Clarity and presentation issues:
	- The MIST definitions are spread across heavy notation that often obscures rather than clarifies the practical mechanism. Links between "masked states", "latent factors", and the "expandable state space" are never made concrete.

	- Multiple policies are introduced (exploration via skills, reward-optimizing policy on masked states, implied composition policy), but this is not explicitly explained and the optimization flow and interactions are not clearly described.

	- The experimental narrative mostly paraphrases plots instead of analyzing mechanisms or failure cases (e.g., underperformance on Reacher, same or worse performance with other backbones)


- Reproducibility concerns: Key mechanisms are vague and underspecified (how the state space expands, how skills are composed, what exactly is optimized by which loss). Some of which include:

    - In RoboSuite, "with access to ground-truth simulator states" is ambiguous. Were they used only for probing or also for training? The probing protocol (targets, regressors, metrics) needs to be specified.

	- The claimed "skill composition" is not described algorithmically. It is unclear how skills are sequenced or combined at test time.

	- Architectural and training specifics (encoders/heads, thresholds, when parameters are frozen,
	a list of all optimization processes, exact curriculum scheduler settings) are missing, which is insufficient for reproduction.


- Technical and notation issues that undermine confidence:

	- The Dreamer objective (Eq. 1) is malformed (conditioned on variables that are also predicted). If this is a shorthand, it still needs to be written correctly or deferred to a faithful appendix.

	- Inconsistent/undefined notation (e.g., what is [d_i]?; inconsistent use of pa(·); reuse of symbols like I_i^(t) for different entities).

	- Eq. 5 is presented as an "information gain" criterion for skill selection, but it clearly depends on encoder/policy parameters. It is not stated whether this loss updates those parameters or is used only for scoring. The optimization pathways are unclear.

**Questions:**

- MIST states and expansion: When and how are new state dimensions added per task? Is the "expandable state space" implemented via masking over a fixed large latent, or do you actually grow the latent dimensionality across tasks? How does this relate to the "latent factors" notation introduced around Line 269-270?

- Skill selection vs. single-task improvements: In Fig. 5, to what extent are gains in single-task learning due to the skill selection mechanism alone? Or, to ask differently: What remains of the framework in pure single-task settings (no curriculum, no cross-task expansion, a single mask,...)?

- Fig. 4a probing: What exactly is shown on the y-axis? Why is it a matrix rather than per-factor bars? I would have assumed you test how well the ground-truth states can be predicted from the full state, which would result in a bar plot. Does the y-axis go over segments of the full state? What do off-diagonal entries represent?

- Skill composition: How are "skills composed" at test time in RoboSuite/Franka-Kitchen/Meta-World? Is there a hierarchical policy, a scheduler, or simple concatenation of skill conditionings? Are you using an existing approach for this? The "composition" of skills alone would be an extra method to be analyzed by a full paper.

- Optimization flow for Eq. (5): Are the encoder and policy parameters updated by the contrastive separability loss used for skill selection, or are they fixed during selection? If updated, how is this coordinated with the other objectives?

---

> ### Author Response · Authors · 2025-11-19
> **Author Response (1/5)**
>
> Thank you for your detailed and thoughtful comments. These are great questions that have helped us improve the paper. We address your concerns point by point below, together with the corresponding revisions we have made. We also appreciate your positive feedback on our overall scope and goal, the technical components, and the empirical validation.
>
> ---
>
>
> > **W1**: Unfocused and incremental: The method is a collection of several incremental ideas (skill selection, MIST masking, curriculum) without a clear, central algorithmic idea or hypothesis. It consists of several known ingredients (DIAYN/MISL-style skills, Dreamer-v3 backbone with masking/MI, UED-like curriculum). The novelty feels diffuse rather than centered on one idea. Even with ablations, it remains unclear which piece drives which improvement and why.
>
>
> [**Response**] We respectfully disagree with this point. Our work is centered around a single, principled goal: learning task-specific minimal and sufficient world models. Achieving such representations inherently requires the synergy of (i) **the agent’s interaction strategy**, (ii) the **environment/task schedule**, and (iii) the **structure-aware world-model objective**. Each component has a distinct and necessary role: active interventions generate informative trajectories that reveal task-relevant factors; the adaptive curriculum exposes these factors in a progressively learnable order; and the structure-learning objective extracts a minimal, task-sufficient subspace from the Dreamer latents. These three parts form a coherent, closed loop rather than a collection of loosely combined modules.
>
>
> Regarding other novelty, beyond combining existing tools, each component is designed specifically for the central goal:
>
>
> (1) **Active intervention**. We go beyond standard MISL-style exploration by learning skills that maximize segment separability with respect to latent factors of the world model (`Eq. (5)`), i.e., skills are selected for how well they induce distinguishable observation trajectories, not merely for unsupervised discovery with MISL objectives.
>
>
> (2) **MIST structure learning**. Although we use Dreamer as a backbone, the proposed masking and sufficiency/minimality objectives over latent coordinates produce task-specific MIST states that are technically different from the original Dreamer representation (`Eq. (2)-(3)`, Approximated objectives in `Appendix B.1`).
>
>
> (3) **Curriculum design**. The curriculum is also centered on the core goal of learning MIST objective to reduce the worst-case world-model and reward prediction error across tasks.
>
>
> In summary, the guiding principle is **learn task-specific minimal and sufficient world models**, while the concrete realization naturally decomposes into interacting modules (active interventions, structured world-model learning, and curriculum). These are not independent “ingredients”, but complementary mechanisms required to implement this principle in practice.
>
>
> [**Related Revision**] To make this central idea and the roles of each component clearer, we have made the following changes:
>
>
> (1). **Revised Fig 1 for conceptual clarity**: We updated `Fig. 1` (`Page 2`) to better illustrate how the components form a single coherent pipeline, making the data flow between agent, curriculum, and structured world model explicit and clear.
>
>
> (2). **Clearer algorithmic explanations**: We expanded the descriptions of all three modules—active interventions, structure learning, and curriculum, in `§3.1.1, 3.1.2, and 3.2` (`Pages 4-6`). Newly added text (highlighted in green) provides explicit objective definitions, training signals, intermediate computation steps, and pointers to the approximate objectives in the appendix.
>
>
> (3). **`Algorithm 1`** We added a high-level pseudocode description (`Page 25`), showing how the modules connect and how data and gradients flow through the system.
>
>
> (4). **Additional supporting details in the Appendix** We added: (i) `A.2` (`Pages 16-17`): explanations of key concepts (unstructured latents, structure learning, MIST states, intervention skills, and segment separability), which are included in the main paper, but we provide more explanations. (ii) `B.4` (`Page 19`): details on expandable latents and how the dimensionality grows across tasks;
> (iii) `B.6` (Pages 19-20): a summary of all objectives, indicating which components receive gradients and which parameters are frozen during each stage of training (shown also by `Table A.2`, `Page 20`).
>
> ---

---

> ### Author Response · Authors · 2025-11-19
> **Author Response (2/5)**
>
> > **W2**: Missing analysis of mechanisms: Given the system complexity, the paper does not convincingly explain when and why each component helps. There is plenty of space wasted to simply describe the performance plots (e.g., in "Sample efficient policy learning") instead of interpreting the results.
>
> [**Response**] Thank you for raising this point. Our method is designed as a synergistic pipeline, and we indeed have ablations that isolate the contribution of each component. We now emphasise these insights much more clearly in the revision.
>
> In particular, the ablation in `Fig. 7` demonstrates that the three stages—interventional skills, structure learning, and curriculum, each address different failure modes and jointly enable learning minimal and sufficient world models. (i) **Interventional skills** are the most critical in terms of final results: replacing them with random or non-separable skills substantially degrades data quality. Without targeted interventions, the agent fails to uncover task-relevant latent factors, which in turn causes the MIST representation and downstream policy performance to drop (`Row 1: pink vs. others`, `Fig. 7`). (ii) **Structure learning** (masking, sufficiency, and minimality) also provides notable improvements. Removing the mask, the sufficiency term, or the MI-based minimality term consistently hurts representation quality and control performance (`Row 2`). (iii) **Curriculum** affects stability and learning order: removing it forces the agent to learn task-specific factors in an arbitrary sequence, often making the MIST subspace harder to recover and leading to performance drops (`Row 3`, `Fig. 7`).
>
> Regarding your comment that the original text devoted too much space to describing plots, we fully agree. In the revision, we reorganized all experimental sections to include explicit takeaways for each group of results—representation quality, policy behaviour, sample efficiency, and generalization. We now interpret what each result implies about the underlying mechanisms, why particular baselines behave as they do, and why our method works (also why some baselines work or do not work).
>
>
> [**Related Revision**] We revised `Sec. 5` (`Pages 8-10`), consolidating figure descriptions and adding clear mechanism-focused takeaways for all experiments (highlighted in green in the revision). We also added additional commentary explaining failure modes and the role of each component.
>
> ---
>
>
> > **W3: Clarity and presentation**
>
>
> >> **W3-1**: The MIST definitions are spread across heavy notation that often obscures rather than clarifies the practical mechanism. Links between "masked states", "latent factors", and the "expandable state space" are never made concrete.
>
> We agree that the formal definitions of MIST can feel notation-heavy (we did it for rigorous concept defs), and we have revised the text to make the underlying mechanism more concrete. Intuitively, MIST states are simply those state factors that directly drive rewards. In the factored DBN view (`Fig. 2(a)`), they correspond to (i) the nodes that are direct parents of the reward node, and (ii) their one-step parents in the previous time step. The mask $m^i$ that we learn in practice can therefore be understood as selecting exactly this set of reward-relevant state coordinates within the latent state $s^i_t$.
>
> The connection between *masked states*, *latent factors*, and the *expandable state space* is as follows. Dreamer provides an unstructured latent vector $s^i_t \in \mathbb{R}^{d_i}$ for each task $T_i$, whose coordinates we interpret as latent factors. As we move to new tasks and the environment introduces additional mechanisms, we allow the latent dimensionality $d_i$ to grow (expandable state space). For each task $T_i$, we then learn a soft mask $m^i \in (0,1)^{d_i}$ that gates these coordinates and defines the task-specific subspace $m^i \odot s^i_t$, which we call the MIST states. Thus: (i) *latent factors* are the coordinates of $s^i_t$,
> (ii) the *expandable state space* means that the dimensionality $d_i$ can increase across tasks, and
> (iii) the *masked states* $m^i \odot s^i_t$ are the subset of those factors that our structure-learning objectives identify as minimal and sufficient for predicting rewards.
>
>
> We made these clearer in the revision, as reflected below:
>
> [**Related Revisions**] In `Sec. 2`, we now explicitly walk through the concrete example in `Fig. 2(a)` after the formal definitions, highlighting which nodes are MIST states in the graph and why. In `Appendix A.2` ('Page 17'), we further unpack the notions of unstructured latents, structure learning, MIST states, intervention skills, and segment separability in more intuitive terms, and we explicitly restate the links between $s^i_t$, $m^i$, and the expandable latent dimensionality across tasks. Also, in `Sec 3.1.1` (`Lines 232-236`, `Page 5`), we provide more information on the expandable states and more details in `Appendix B.4` ('Page 19`).

---

> ### Author Response · Authors · 2025-11-19
> **Author Response (3/5)**
>
> >> **W3-3**: The experimental narrative mostly paraphrases plots instead of analyzing mechanisms or failure cases (e.g., underperformance on Reacher, same or worse performance with other backbones)
>
> This is closely related to our response and revisions for **W2**, so we kindly refer you there for the full discussion. Here we briefly clarify the main points, especially regarding failure cases.
>
>
> For the Reacher-Hard environment, MIST-WM underperforms TD-MPC2 by a small margin, which we attribute primarily to differing optimization objectives: TD-MPC2 optimizes a model-free control objective, while MIST-WM builds on Dreamer-style world-model objectives. This discrepancy appears only on Reacher-Hard, and the gap is relatively small (about 4%), while MIST-WM still substantially outperforms all Dreamer-based baselines. We explicitly discuss this underperformance and its likely cause in the revised `Sec. 5`. We wanted to point out that this is the only case in which we underperformed on the tasks we tested.
>
>
> Regarding “other backbones,” we believe this refers to the ablations using DINO-WM or R3M as visual encoders. In these experiments, we are **not** comparing against the original DINO/R3M control pipelines; instead, we plug their encoders into our framework in place of Dreamer’s encoder, while keeping the rest of MIST-WM unchanged. These results are intended to demonstrate that our structure-learning mechanism is compatible with multiple perceptual backbones and that performance remains comparable across encoders, which also helps justify our choice of Dreamer as the main backbone. We have clarified this in `Sec 5` (`Line 517`, `Page 10`)
>
> ---
>
> > **W4**: Reproducibility concerns: Key mechanisms are vague and underspecified (how the state space expands, how skills are composed, what exactly is optimized by which loss).
>
>
> >> **W4-1**: In RoboSuite, "with access to ground-truth simulator states" is ambiguous. Were they used only for probing or also for training? The probing protocol (targets, regressors, metrics) needs to be specified.
>
>
> [**Response**] We apologize for the ambiguity. By “with access to ground-truth simulator states,” we only mean that the simulator exposes the true underlying factors so that we can evaluate the learned representations. These ground-truth variables are **never** used to train the world model or the policy; they are used solely for probing.
>
>
> As for the details of probing, we assess representation quality by measuring how well the learned latents predict the ground-truth simulator factors. Since the latent space can be over-parameterized, a single factor may be encoded across multiple coordinates. We therefore first freeze all world-model parameters and use only the encoders to map observations to latent vectors $\mathbf{s}^i_t$, so no further learning happens in the world models during this evaluation.
>
>
> Given the assignment of latent dimensions to each ground-truth variable, we group the corresponding coordinates and attach a separate probe network to each group. Each probe is a two-layer MLP with hidden size 128, trained to predict the associated ground-truth factors from the grouped latents. We use mean-squared error for continuous factors and cross-entropy loss for categorical factors during probe training.
>
>
> We use the coefficient of determination ($R^2$) for continuous variables to quantify the “matching quality” between the learned representation and the ground-truth factors. For a ground-truth variable $x_i$ with predictions $\hat{x}_i$, we compute $R^2$ (`Eq (A15)`, `Page 20`) to evaluate the matching quality.
>
>
> [**Related Revision**] We now explicitly clarify how ground-truth factors are used in `Sec. 5`, and we include a full specification of the probing protocol in `Appendix B.7` (`Page 20`).
>
>
> >> **W4-2**: The claimed "skill composition" is not described algorithmically. It is unclear how skills are sequenced or combined at test time.
>
> [**Response**] As this point overlaps with our response to **W3-2**, we briefly summarize here. We do not introduce an additional composition policy; compositional generalization in our experiments is achieved purely through the learned latent space. Concretely, we maintain an expandable latent state space, and when a new task involves compositions of previously seen skills or objects, we initialize from the existing latent space and train a standard reward-optimizing policy on top of it with fewer samples. Thus, “composition” arises from reusing and extending the latent representation, rather than from a separate, explicitly designed composition policy. We clarify this explicitly in the revised `Sec. 5` (`Lines 483-485`, `Page 9`) and revise related parts in other sections.

---

> ### Author Response · Authors · 2025-11-19
> **Author Response (4/5)**
>
> >> **W4-3**: Architectural and training specifics (encoders/heads, thresholds, when parameters are frozen, a list of all optimization processes, exact curriculum scheduler settings) are missing, which is insufficient for reproduction.
>
> We have updated more details in the revised version, including `Table A.2` for the gradient flow w.r.t. the objective functions, `Table A4-A5` for intervention and MIST learning, `Table A6-A7` for policy leanring, and `Table A8` for curriculum learning.
>
> ---
>
> > **W5**： Technical and notation issues
>
>
> >> **W5-1**: The Dreamer objective (Eq. 1) is malformed (conditioned on variables that are also predicted). If this is shorthand, it still needs to be written correctly or deferred to a faithful appendix.
>
>
> [**Response**]: Thank you for the pointer. We have corrected this typo.
>
>
> [**Related Revision**]: `Eq (1)`
>
>
> >> **W5-2**: Inconsistent/undefined notation (e.g., what is [d_i]?; inconsistent use of pa(·); reuse of symbols like I_i^(t) for different entities).
>
>
> [**Response**] Thank you for pointing out these notation issues. We have clarified and streamlined the notation in the revised version. Specifically, we now explicitly define $[d_i]$ as the index set $\{1, \dots, d_i\}$ for the dimensions of $s^i$. The notation $\mathrm{pa}(\cdot)$ is used consistently throughout to denote parent sets in the task-specific DBN. For the sets previously denoted $I_i^{(t)}$, we now split them into two clearly distinguished parts: $I_{i,1}^{(t)}$ for one-step connections and $I_{i,2}^{(t)}$ for two-step connections.
>
>
> [**Related Revision**] These changes are reflected in `Sec`, in particular `Definitions 1–3`.
>
>
> >> **W5-3**: Eq. 5 is presented as an "information gain" criterion for skill selection, but it clearly depends on encoder/policy parameters. It is not stated whether this loss updates those parameters or is used only for scoring. The optimization pathways are unclear.
>
>
> [**Response**] The loss in Eq. (5) is used both to *score* skills and to *update* parameters. Concretely, it optimizes the trajectory encoder $\xi$ and the exploration (skill) policy $\varepsilon$. In other words, we not only evaluate which skills induce more separable trajectory segments, but also update the exploration policy to actively seek such informative trajectories. This makes Eq. (5) an information-gain *training* objective for the skill policy, rather than a purely post-hoc scoring function.
>
>
> [**Related Revision**] We make this emphasized in `Sec 3.1.2` (`Lines 300-302`, `Page 6`), `Algorithm 1` (`Page 25`) and `Table A.2` (`Page 20`).
>
> ---
>
> > **Q1**: MIST states and expansion: When and how are new state dimensions added per task? Is the "expandable state space" implemented via masking over a fixed, large latent, or do you actually grow the latent dimensionality across tasks? How does this relate to the "latent factors" notation introduced around Line 269-270?
>
>
> [**Response**] We do *not* assume that the ``exactly-matching'' latent dimensionality per task is known to the agent. Instead, we follow a simple over-parameterized heuristic that is consistent with practice and then rely on structure learning to select the task-relevant subspace. As to the relations between this to the "latent factors", these are exactly the latent factors for each task.
>
>
> Concretely, for each benchmark, we initialize the Dreamer latent dimensionality $d_1$ to a fixed value and keep it shared across tasks at the beginning (e.g., $d_1 = 512$ for Meta-World, RoboSuite, and Kitchen, which have high-dimensional visual observations, and $d_1 = 256$ for DMControl). When moving from task $T_i$ to $T_{i+1}$, we allow the latent dimensionality to grow by a fixed increment, $d_{i+1} = d_i + \Delta d$.
> where we set $\Delta d = 32$ in all experiments. The choice of $d_1$ and $\Delta d$ is therefore a heuristic design decision rather than an oracle choice of the ``correct'' dimension. In practice, this slight over-parameterization is benign because our structure-learning module (through the sufficiency and minimality objectives and the learned mask $m^i$) identifies and uses only the task-relevant subset of dimensions (the MIST states), while irrelevant coordinates are effectively suppressed.
>
>
> [**Related Response**] We have added this explanation to `§3.1.1` (`Lines 232-236`, `Page 5`) and provided further implementation details in `Appendix B.4` (`Page 19`).
>
> ---

---

> ### Author Response · Authors · 2025-11-19
> **Author Response (5/5)**
>
> > **Q2**: Skill selection vs. single-task improvements: In Fig. 5, to what extent are gains in single-task learning due to the skill selection mechanism alone? Or, to ask differently: What remains of the framework in pure single-task settings (no curriculum, no cross-task expansion, a single mask,...)?
>
>
> [**Response**] In the single-task setting, the framework reduces to two key components: (i) structure learning to obtain MIST states, and (ii) active intervention skills to collect informative data for learning these states. There is no curriculum and no cross-task expansion, and only a single mask is learned for the task. As a result, the single-task gains in Fig. 5 primarily reflect the effect of the skill-based exploration together with the MIST-style structure learning. We have clarified this decomposition explicitly in `Sec. 5` (`Lines 453-454`, `Page 9`) of the revised paper.
>
>
> [**Related Revisions**] `Sec 5` (`Lines 453-454`, `Page 9`)
>
> ---
>
> > **Q3**: Fig. 4a probing: What exactly is shown on the y-axis? Why is it a matrix rather than per-factor bars? I would have assumed you test how well the ground-truth states can be predicted from the full state, which would result in a bar plot. Does the y-axis go over segments of the full state? What do off-diagonal entries represent?
>
> [**Response**] For this question, please kindly refer to the probing details we provided in the response to **W4-1**. Here, the matrix in `Fig. 4a` visualizes the alignment between learned latent coordinates and individual ground-truth factors. Each entry corresponds to the $R^2$ score between a subset of latent dimensions (as selected by the learned assignment) and a particular ground-truth factor. Thus, the y-axis indexes ground-truth factors and the x-axis indexes groups of latent dimensions.
>
> Because we evaluate the mapping between groups of latents and each ground-truth variable, the result is naturally a matrix rather than a single bar per factor. Diagonal entries indicate how well the intended latent group captures its corresponding factor, while off-diagonal entries reflect unintended correlations with other factors (i.e., cross-factor leakage). This view is therefore more diagnostic than a single bar plot from the full state, as it reveals both correct alignments and cross-talk between latents and factors.
>
> [**Related Revision**] We have clarified this explanation and the axis semantics in the revised probing description in the appendix (`Appendix B.7`, `Page 20`).
>
> ---
>
> > **Q4**: Skill composition: How are "skills composed" at test time in RoboSuite/Franka-Kitchen/Meta-World? Is there a hierarchical policy, a scheduler, or simple concatenation of skill conditionings? Are you using an existing approach for this? The "composition" of skills alone would be an extra method to be analyzed by a full paper.
>
>
> [**Response**] Please kindly refer to our responses to **W4-2** and **W3-2**.
>
> ---
>
> > **Q5**: Optimization flow for Eq. (5): Are the encoder and policy parameters updated by the contrastive separability loss used for skill selection, or are they fixed during selection? If updated, how is this coordinated with the other objectives?
>
>
> [**Response**] Please kindly refer to our response to **W5-3**. The loss in Eq. (5) is used both to score skills and to update parameters. When we optimize it, other objectives, like world model ones are fixed for stability.
>
>
> ---
>
>
> Again, we thank you for this very detailed and thoughtful set of comments. They have greatly helped us strengthen and clarify the paper. In summary, W1 helped us to more clearly articulate our core contribution and unify the presentation; W2 and W4 led us to expand the technical explanations, add more details, and revise figures, algorithms, and notation; and W3 encouraged us to clarify the mechanisms behind our results and explicitly address edge cases and analyses. We hope the revisions and explanations above fully resolve your concerns. We are grateful for the opportunity to incorporate your suggestions, and please feel free to let us know if any point remains unclear.

---

> > ### Author Response · Authors · 2025-11-26
> > **Thank you for your thoughtful review**
> >
> > Dear Reviewer qDyg,
> >
> > Thank you once again for this very detailed and thoughtful set of comments. All of these have greatly helped us strengthen and clarify the paper. In summary, W1 pushed us to articulate our core contribution more clearly and better unify the presentation; W2 and W4 led us to expand the technical explanations, add missing details, and refine figures, algorithms, and notation; and W3 encouraged us to clarify the mechanisms behind our results and explicitly discuss edge cases and analyses. We hope that the revisions and explanations above address your concerns. We are really happy for the opportunity to incorporate your suggestions; as the discussion period is nearing its deadline, please feel free to let us know if any point remains unclear, and we will be happy to follow up.
> >
> > The Authors

---

> > > ### Comment · Reviewer_qDyg · 2025-11-27
> > >
> > > Thanks for the detailed rebuttal and the added clarifications. I appreciate the revisions made by the authors (fixed Eq. (1), list of take-aways, somewhat clearer module descriptions, expanded appendix). However, the impression remains that the paper is spread across too many interacting elements, where the causal attribution is hard to figure out and the core novelty remains unclear. I feel in its current form, the paper has substantial issues with clarity, focus, and reproducibility. I therefore maintain my recommendation to Reject.
> > >
> > >
> > > Below is a concise summary of what I find resolved and what remains lacking.
> > >
> > >
> > > **What your rebuttal clarified:**
> > >
> > > - Ground-truth states: Used only for probing/evaluation, not training. Probing protocol and $R^2$ matrix interpretation are clearer.
> > >
> > > - Optimization pathways: Eq. (5) is not just a selection score; it also trains the trajectory encoder and the exploration (skill) policy, while world-model parameters are frozen during that step.
> > >
> > > - Single-task scope: In single-task settings the framework reduces to skill-based exploration + structure learning (masking/minimality/sufficiency); no curriculum and no cross-task expansion.
> > >
> > > - Notation and objective fixes: Some notation clean-up and a corrected Dreamer objective.
> > >
> > >
> > > **Core concern**: While the authors spend a lot of time defending their work, the changes to the paper are minimal, whereas most of our previous concerns would require a larger rewrite. For example:
> > >
> > > 1. Contribution still feels diffuse despite reorganization: Even with the revised manuscript, the central technical novelty is still spread across several well-known components (MISL/DIAYN-style skills with a contrastive separability loss; Dreamer + masking + MI/sufficiency; UED-style curriculum), and the paper does not land on one self-contained idea. The ambition to learn “task-specific minimal and sufficient world models” is compelling, but the current implementation remains a collection of incremental ideas, despite the authors disagreeing.
> > >
> > >
> > > 2. Mechanism attribution remains weak
> > >      - Your additions to the text emphasizes ablations, but the causal attribution is still largely qualitative. In single-task settings, you explicitly state that improvements stem mainly from the skills + structure learning; the curriculum and expansion play no role. In multi-task settings, you claim synergy, but a quantitative decomposition (with compute- and data-matched controls) that isolates the contribution of each module is still missing.
> > >     - In practice, the separability objective (Eq. 5) reads like a standard contrastive objective over sequence encodings. It remains unclear to what extent it truly targets “task-relevant latent factors” (as opposed to just boosting general exploration diversity) and how much of the benefit would be captured by a strong MISL baseline plus a robust Dreamer training with careful data selection.
> > >
> > >
> > > 3. “Skill composition” claim walks back to “representation reuse”: The rebuttal clarifies there is no explicit composition policy or scheduler. What you call “compositional generalization” is primarily reusing/expanding the latent space when moving to new tasks, then training a standard control policy on top. That is materially different from skill composition and should be framed accordingly; several parts of the paper still read as if skills are composed directly at inference time. The language used by the authors is misleading.
> > >
> > >
> > > 4. Expandable latent dimensionality is a heuristic with limited analysis: You now state that the latent size grows by a fixed increment per task and that masking suppresses irrelevant coordinates. This is a reasonable heuristic, but there is no sensitivity analysis on the growth rate, mask sparsity targets, or failure modes (e.g., when the latent is under- or over-grown). Without such analysis, it is difficult to assess the robustness of the MIST claims.
> > >
> > >
> > > 5. Results interpretation still feels thin: The revised text offers more commentary, but the analysis remains surface-level in places where the story matters most (e.g., why performance gaps cluster across backbones; when curriculum materially helps vs. hurts; why Reacher-Hard underperforms and what that implies for the approach). The paper still spends a lot of space paraphrasing plots rather than extracting mechanistic insight.
> > >
> > >
> > > 6. Reproducibility is slightly improved but still short for a paper of this scope: The added tables and pseudocode in the appendix help, but the approach’s complexity (multiple objectives, staged training, growing latents, masking and MI schedules, curriculum scheduler) still requires a much more exhaustive specification. Given the breadth of benchmarks and claims, stronger reproducibility support would be necessary.

---

> ### Author Response · Authors · 2025-11-27
> **Further Response (1/3)**
>
> Thank you for your response! We are glad that some of your concerns have been clarified, and we appreciate the additional comments on the remaining points. We would like to take this opportunity to further elaborate on these remaining issues.
>
> ---
>
> > Contribution still feels diffuse despite reorganization
>
> We appreciate the follow-up and understand the concern. As we mentioned, our core contribution is indeed centered on a single idea: **learning task-specific minimal and sufficient world models (MIST)**. The three components you mention are not meant as loosely combined ingredients, but as the concrete mechanisms required to realize this goal in practice.
> Also, in terms of algorithmic novelty for each component: (i) Active interventions go beyond standard MISL-style exploration by selecting skills based on segment separability in the latent space (`Eq. (5)`), i.e., how well skills induce distinguishable trajectories that expose task-relevant factors. (ii) MIST structure learning adds masking with sufficiency/minimality objectives (`Eqs. (2)–(3)`, `Appendix B.1`), yielding task-specific latent subspaces that are qualitatively different from the original Dreamer representation. (iii) Curriculum orders tasks so that these MIST factors can be exposed and refined in a progressively learnable way, rather than relying on an arbitrary training order.
>
> We therefore see the method not as a collection of incremental ideas, but as a coherent recipe of the MIST objective, with each module playing a specific and necessary role in making “task-specific minimal and sufficient world models” operational.
>
> ---
>
> > Your additions to the text emphasizes ablations, but the causal attribution is still largely qualitative. In single-task settings, you explicitly state that improvements stem mainly from the skills + structure learning; the curriculum and expansion play no role. In multi-task settings, you claim synergy, but a quantitative decomposition (with compute- and data-matched controls) that isolates the contribution of each module is still missing.
>
> We agree that in single-task settings only two components are meaningful—there is no notion of curriculum or latent-space expansion when the task count is one, so the ablation naturally focuses on intervention skills and MIST structure learning, which are indeed the only mechanisms expected to matter there.
>
> For the multi-task case, we would like to clarify that we do provide module-level ablations with matched compute and data budgets. In Fig. 7:
>
> - Row 1 removes or replaces the intervention module (random policy, DIAYN-style skills, METRA, etc.), showing the effect of data quality while holding all other components fixed.
>
> - Row 2 ablates structure learning (removing minimality, removing sufficiency, removing MI), under identical training steps and backbone.
>
> - Row 3 removes the curriculum or changes the backbone, again under matched training budgets.
>
> These controls isolate each component’s contribution under the same compute and data constraints across tasks. While curriculum and expansion do not appear in single-task evaluations by design, in the multi-task regime the ablations demonstrate that each of the three components influences downstream performance, and the full model benefits from their combination.
>
> ---

---

> > ### Author Response · Authors · 2025-11-27
> > **Further Response (2/3)**
> >
> > > In practice, the separability objective (Eq. 5) reads like a standard contrastive objective over sequence encodings. It remains unclear to what extent it truly targets “task-relevant latent factors” (as opposed to just boosting general exploration diversity) and how much of the benefit would be captured by a strong MISL baseline plus a robust Dreamer training with careful data selection.
> >
> > The separability objective in Eq. (5) is indeed implemented as a contrastive objective over segment encodings, but its effect is more specific than promoting generic exploration diversity.
> >
> > The key point is that *segment separability is induced by underlying physical latent factors*. For example, two cubes with visually similar appearance but different masses generate distinguishable observation trajectories when lifted. Our skill policy is explicitly optimized to *create* such latent-revealing differences. This yields data that contains the variations required for minimality/sufficiency scoring, enabling the model to later isolate the task-relevant coordinates.
> >
> > This mechanism is reflected empirically. In `Fig. 4(a,c)` and Appendix `Table A.9`, the latent space learned under our objective aligns strongly with ground-truth physical variables, whereas MISL-style or intrinsic-motivation baselines (METRA, Plan2Explore, Curiosity Replay) do not reliably induce such factor-specific separability. For convenience, we put the probing results ($R^2$) here:
> >
> > | Methods          | Size | C-pos | C-Orient | G-pos | G-Orient | G-openness | C-Mass | Average |
> > |------------------|------|-------|----------|-------|----------|------------|--------|---------|
> > | Random           | 0.01 | 0.92  | 0.16     | 0.48  | 0.21     | 0.06       | 0.03   | 0.31    |
> > | METRA            | 0.03 | 0.95  | 0.85     | 0.51  | 0.49     | 0.20       | 0.55   | 0.59    |
> > | Curiosity Replay | 0.02 | 0.96  | 0.53     | 0.23  | 0.36     | 0.15       | 0.39   | 0.44    |
> > | Plan2Explore     | 0.01 | 0.93  | 0.59     | 0.41  | 0.26     | 0.10       | 0.40   | 0.45    |
> > | MIST-WM | 0.02 | 1.00  | 0.94     | 0.86  | 0.69     | 0.21       | 0.96   | 0.78    |
> >
> >
> > As shown, our intervention-driven separability yields substantially higher alignment with ground-truth variables. Even strong MISL baselines do not achieve comparable structure discovery.
> >
> > Therefore, the contrastive separability objective is not merely promoting exploration diversity; it is actively steering skills toward trajectories that expose latent causal factors, which is essential for learning minimal task-sufficient world-model representations.
> >
> > ---
> >
> > > “Skill composition” claim walks back to “representation reuse”: The rebuttal clarifies there is no explicit composition policy or scheduler. What you call “compositional generalization” is primarily reusing/expanding the latent space when moving to new tasks, then training a standard control policy on top. That is materially different from skill composition and should be framed accordingly; several parts of the paper still read as if skills are composed directly at inference time. The language used by the authors is misleading.
> >
> > We agree that our original wording could give the impression that we perform *policy-level* skill composition at inference time, which is not the case. Our focus is on **representation-level compositionality**, where the learned latent space expands and reuses previously acquired factors when encountering new tasks. The downstream control policy is then trained in the standard way on top of these representations.
> >
> > We have revised the manuscript to make this distinction explicit. In particular:
> >
> > - In the Introduction (Line 92), we now clarify that *our form of compositional generalization refers to compositional reuse and expansion of latent factors*, not explicit skill chaining.
> > - In the experimental analysis (Lines 483–485), we emphasize that improvements stem from representational composition across objects and skills, rather than composing skill policies at inference time.
> >
> > This framing is also consistent with prior work on representational or perceptual compositional generalization in RL and imitation learning [1–3], where the focus is on factoring and recombining latent structure rather than composing skill executors.
> >
> > We believe the revised text resolves the ambiguity and more accurately reflects the scope of our contribution.
> >
> > [1] Qi, Carl, et al. "EC-Diffuser: Multi-Object Manipulation via Entity-Centric Behavior Generation." ICLR 2025
> >
> > [2] Wang, Xinyue, and Biwei Huang. "Modeling Unseen Environments with Language-guided Composable Causal Components in Reinforcement Learning." ICLR 2025
> >
> > [3] Haramati, Dan, Tal Daniel, and Aviv Tamar. "Entity-centric reinforcement learning for object manipulation from pixels." ICLR 2024.

---

> ### Author Response · Authors · 2025-11-27
> **Further Response (3/3)**
>
> ---
>
> > Expandable latent dimensionality is a heuristic with limited analysis: You now state that the latent size grows by a fixed increment per task and that masking suppresses irrelevant coordinates. This is a reasonable heuristic, but there is no sensitivity analysis on the growth rate, mask sparsity targets, or failure modes (e.g., when the latent is under- or over-grown). Without such analysis, it is difficult to assess the robustness of the MIST claims.
>
> We appreciate this comment. The expandable latent dimensionality is indeed a heuristic, but it is motivated by a simple and practical principle: **use mild over-parameterization to allow new factors to emerge, and rely on the minimality/sufficiency objectives to suppress irrelevant ones.** In this setup, the fixed growth rate primarily determines how much unused capacity is available for future tasks, while the masking and structure losses control which coordinates remain active.
>
> Empirically, we found the method to be fairly insensitive to the exact growth rate. Before submission, we tested several increments (+/-50%) and did not observe meaningful differences in downstream performance, presumably because unnecessary dimensions are naturally pruned by the minimality and sufficiency pressures. That said, we agree that this sensitivity analysis is important. We rerun a more systematic sweep, and here are the results on Meta-World Generalization performances:
>
> | Environment           | 32 | 16 | 64 |
> |-----------------------|--------------------|--------------------------|--------------------------|
> | Box-close (Single)    | **0.94 ± 0.02**    | 0.89 ± 0.06              |0.92 ± 0.05              |
> | Assembly (Single)     | 0.79 ± 0.12        | **0.81 ± 0.04**              |0.87 ± 0.02              |
> | Coffee-push (Gen)     | **0.89 ± 0.05**    | 0.86 ± 0.04              |0.87 ± 0.03              |
>
> Results indicate that there is no significant difference across the choices of this hyper-parameter.
>
> ---
>
> > Reproducibility is slightly improved but still short for a paper of this scope
>
> As you pointed out, beyond the additional text in the rebuttal, the revised paper also includes: (i) a full gradient-flow table (`Table A.2`), (ii) `Algorithm 1` summarizing the end-to-end training loop, (iii) detailed hyperparameters and training times, and (iv) an anonymized code release linked in the general response. We hope this substantially lowers the barrier to reproducing and extending the method.
>
> ---
>
> Again, we sincerely appreciate your careful reading and critical feedback, and we hope these additional clarifications better convey both the intent and the technical substance of our work. Some of the points above extend what we already tried to express in the initial rebuttal, but your comments helped us see where the exposition was still unclear and improve it accordingly. If there are any remaining concerns, we would be very happy to discuss them further. Thank you again for your time and thoughtful engagement with our paper.

---

### Official Review · Reviewer_d8kx · 2025-10-31

**Soundness:** 3
**Presentation:** 1
**Contribution:** 2
**Rating:** 2
**Confidence:** 3

**Summary:**

This paper presents a world-model learning system with an active data collection strategy. The world model is trained using a standard variational objective augmented with proposed sufficiency scores and a minimality loss. The data collection policy is learned by maximizing information gain. Experiments on simulated manipulation and locomotion benchmarks show improved policy performance with the learned world model.

**Strengths:**

1. This paper introduces a novel paradigm of world-model learning with active data collection strategy.
2. Experimental results show improved task completion performance when compared with other baselines.

**Weaknesses:**

1. The paper is difficult to follow. It introduces uncommon concepts without sufficient explanation, presents a complex system design, and lacks clarity in the demonstration of results.

2. The system relies heavily on external information and sub-modules, such as reward design and state mask learning, which could limit its practical applicability in open environments.

3. While experiments are conducted on established benchmarks like DMControl and MetaWorld, newer and more reliable benchmarks, such as SimplerENV[1], are not considered. Additionally, real-world experiments are essential to validate the world model's generalization performance.

For further details, please refer to the questions.


[1] Li X, Hsu K, Gu J, et al. Evaluating Real-World Robot Manipulation Policies in Simulation[C]//Conference on Robot Learning. PMLR, 2025: 3705-3728.

**Questions:**

1. Some concepts introduced in this paper, such as structural learning, unstructured latent states, and segment separability, are not properly explained. The authors could consider remove redundant concepts or provide clearer explanations. Additionally, terms like "informative" (explained on Line 275) should be defined upon first mention.
2. Some terms are used imprecisely. For example, "intervention" in robotics typically refers to altering a running system’s dynamics to correct or prevent danger. However, in this context, it seems to refer to the data collection phase.
3. No efficiency metrics, such as inference or training time, are provided to demonstrate the efficiency and generalizability of the policy shown in Figure 1.
4. The legend in Figure 6 is missing.

---

> ### Author Response · Authors · 2025-11-19
> **Author Response (1/3)**
>
> Thank you for your comments and for recognizing the novelty of our approach. Your questions and suggestions are very helpful. We address each of them point by point below, together with the corresponding revisions in the paper.
>
>
> ---
>
>
> > **W1**: The paper is difficult to follow. It introduces uncommon concepts without sufficient explanation, presents a complex system design, and lacks clarity in the demonstration of results.
>
>
> [**Response**] Thank you for pointing this out. As you noted in the strengths (novel paradigm of world-model learning with active data collection), our framework necessarily requires the synergy between several interacting modules, which can make the presentation dense. Based on your feedback, we have carefully revised the paper to clarify the key concepts and to better structure the experimental discussion.
>
>
> **(1) Uncommon Concepts**
>
>
> - **Unstructured Latent States** We denote by $s_t \in \mathbb{R}^d$ the latent state of a standard Dreamer-style world model, learned purely from reconstruction and prediction objectives (Eq.~(1)). At this stage, $s_t$ is *unstructured*: no factorization or task-specific subset is identified.
>
>
> - **Structure Learning** Given unstructured latents $s_t^i$ for task $T_i$, we learn a mask $m^i \in (0,1)^d$ and associated heads that define a factored, task-specific subspace $\mathcal{S}^{\mathrm{min}}_i = \{m^i \odot s_t^i\}$.  This is done by optimizing the *sufficiency* and *minimality* scores, which favor masked latents that are both predictive of task information and as low-dimensional as possible.
>
>
> - **MIST States** For each task $T_i$, we define index sets $U_i \subseteq [d_i]$ that collect those latent factors with direct or one-step parents into the reward in the underlying DBN (`Def. 1--3`). The corresponding *MIST states* are the coordinates $(s_t^i)_{U_i}$.
>
>
> - **Active Intervention Skills** We maintain a library of latent skills $z \sim p_\eta(z)$ and an exploration policy $\pi_\varepsilon(a \mid s, z)$, learned using a MISL-style objective and contrastive objectives that can identify the informative trajectories for learning the world models with MIST states.
>
>
> - **Segment Separability** For a latent factor $s^{(n)}_t$ taking values $v$ and $v'$, skills that help distinguish trajectories starting from $s^{(n)}_t = v$ versus $s^{(n)}_t = v'$ provide high information gain. We formalize this via an InfoNCE-style contrastive loss over encoded segments in `Eq. (5)`. Minimizing this loss encourages *segment separability*, i.e., skills that make observation segments for different latent values easily distinguishable, which in turn helps identify MIST states.
>
>
> **(2) Clarification of Results Presentations**
>
>
> We have also revised the experimental sections to focus less on paraphrasing plots and more on interpreting mechanisms and failure cases. Each group of results (representation quality, policy behaviour, sample efficiency, and generalization) now has explicit ``Takeaway'' statements that summarize what the experiments show about the roles of the different components and why certain baselines succeed or fail.
>
>
> [**Related Revision**]
> - **Concepts**
>
>
> In `§2.2` and `Fig. 2` (`Page 3`), we more clearly connect the formal MIST definitions to an intuitive DBN example (which nodes and edges correspond to MIST states).
>
>
> In `§3.1–3.2` (`Pages 4-6`), we tightened the first mention of each concept and added short intuitive explanations alongside the equations.
>
> In `Appendix A.2` (`Page 17`), we added a dedicated subsection that restates these five concepts in a self-contained way with minimal notation and simple formulas (as summarized above).
>
> - **Results**
>
> In `§5` (`Pages 8-9`), we restructured the experimental narrative to replace plot paraphrasing with explicit takeaways and mechanism-level discussion (highlighted in the revision).
>
> ---

---

> ### Author Response · Authors · 2025-11-19
> **Author Response (2/3)**
>
> > **W2**: The system relies heavily on external information and sub-modules, such as reward design and state mask learning, which could limit its practical applicability in open environments.
>
> > **W3**: While experiments are conducted on established benchmarks like DMControl and MetaWorld, newer and more reliable benchmarks, such as SimplerENV, are not considered. Additionally, real-world experiments are essential to validate the world model's generalization performance.
>
> [**Response**] Our work is centered around a single, principled goal: learning **task-specific minimal and sufficient world models**. Achieving such representations naturally requires the synergy of (i) the **agent’s interaction strategy**, (ii) the **environment/task schedule**, and (iii) the **structure-aware world-model objective**. Each component has a distinct and necessary role and works together closely: active intervention skills generate informative trajectories that reveal task-relevant factors; the adaptive curriculum exposes these factors in a progressively learnable order; and the structure-learning objective extracts a minimal, task-sufficient subspace from the Dreamer latents. These parts form a coherent closed loop, rather than a collection of loosely combined complex modules. **Hence, we argue the method is not overly complex to deploy in open environments; rather, it is specifically designed for them.**
>
> Regarding "reward design": all signals we use are standard and available in typical RL setups. For exploration, the skill policy is trained with intrinsic rewards based on mutual information and separability (`Eq. (4)`–`(5)`), and does not require any oracle labels or environment-specific engineering. For control, we use the usual task reward from the benchmark. The state masks are not hand-designed; they are learned end-to-end from data using the sufficiency and minimality scores (`Eq. (2)`–`(3)`), and updated jointly with the world model. This design is precisely intended to make the framework applicable in open environments: skills are task-agnostic and aim to uncover salient latent factors, while the mask adapts to whichever subset of latents is sufficient for the current task.
>
> Per your suggestion, we are also testing on SimplerEnv are promising testbed. We are currently running experiments on this using the Dreamer-v3 backbone and will update you with the results in this rebuttal thread. As SimplerEnv is built from real-world image data, we view it as a natural stepping stone toward real-robot evaluation. Due to space and resource constraints, we were not able to include real hardware experiments during this rebuttal, but we will leave this as an important direction for future work.
>
> ---
>
> > **Q1**: Some concepts introduced in this paper, such as structural learning, unstructured latent states, and segment separability, are not properly explained. The authors could consider removing redundant concepts or provide clearer explanations. Additionally, terms like "informative" (explained on Line 275) should be defined upon first mention.
>
> [**Response**] Thank you for this suggestion. We have revised the relevant sections to clarify these concepts; a detailed list of changes is provided in our response to **W1**. In particular, we now give concise, self-contained explanations of *structural learning*, *unstructured latent states*, and *segment separability* and connect them directly to the MIST construction and the overall pipeline.
> Regarding the term *informative*, we now define it at first use in `Sec. 3.1.2`: in our context, “informative data” refers to trajectories that expose how latent factors influence observations and rewards, rather than passively reflecting a narrow slice of the state space. This notion is tightly related to segment separability, since informative trajectories are precisely those that make different latent valuations distinguishable in observation space.
>
>
> [**Related Revision**] We added an explicit explanation of “informative data” in `Sec. 3.1.2` (`Lines 244-248`, `Page 5`) and clarified all related concepts as listed in the response to **W1**.
>
> ---

---

> ### Author Response · Authors · 2025-11-19
> **Author Response (3/3)**
>
> > **Q2**: Some terms are used imprecisely. For example, "intervention" in robotics typically refers to altering a running system’s dynamics to correct or prevent danger. However, in this context, it seems to refer to the data collection phase.
>
> [**Response**] Thank you for pointing this out. In our work, we use the term *intervention* in the sense of intervening on latent factors through actions, rather than in the safety-oriented sense commonly used in robotics. Concretely, the skill policy selects actions that actively perturb the environment so as to induce distinguishable trajectories for different latent factors (e.g., lifting two visually similar objects and observing different motion due to mass), thereby “intervening” on the underlying latent variables of the world model. This notion follows the usage of intervention in the latent-variable and causal generative modeling literature [1-4], where interventions are applied to hidden factors rather than directly to low-level dynamics. We have added a short explanation to clarify this usage in Sec. 3.1.2.
>
>
> [**Related Revision**] Explanation in `Sec 3.1.2` (`Lines 251-253`, `Page 5`).
>
> ---
>
> > **Q3**: No efficiency metrics, such as inference or training time, are provided to demonstrate the efficiency and generalizability of the policy shown in Figure 1.
>
> [**Response**] We indeed reported training-time efficiency in the original submission. As summarized in `Appendix C.4` and `Table A.3` in our original submission, our method adds only modest overhead relative to Dreamer-v3. During generalization (test-time), the only additional computation compared to the backbone is the inference of MIST states, which involves the soft mask and the MINE estimator. These components together account for less than 10% of full inference time, since exploration skills and curriculum are used only during training and not at test time.
>
> For completeness, we report below the training-time costs on 4× NVIDIA A100 GPUs (80GB). While our method does introduce some overhead from structure learning, skill learning, and curriculum, the total cost remains within a reasonable range relative to Dreamer-v3.
>
> | Task Suite      | Ours | Dreamer-v3 |
> |-----------------|------|------------|
> | DM Control      | 0.35 | 0.25       |
> | Meta-World      | 0.75 | 0.50       |
> | Franka Kitchen  | 0.35 | 0.30       |
> | RoboSuite       | 0.75 | 0.60       |
>
> ---
>
> > **Q4**: The legend in Figure 6 is missing.
>
>
> [**Response**] We apologize for the confusion and have corrected this.
>
>
> [**Related Revision**] `Fig. 6`
>
>
> ---
>
>
> Thank you again for your constructive feedback and suggestions. We believe that the revisions addressing W1 (together with the listed changes in the main text and appendices) substantially improve the clarity of the presentation, and that our response to W2 clarifies the algorithmic design and its practical complexity. For Q1–Q3, we hope the additional details on concepts, terminology, and efficiency metrics address your concerns.
>
>
> As suggested, we are also running experiments on SimplerEnv and will report these results in the discussion once they are ready. We are very grateful for your comments, which have helped us significantly improve the paper, and we are happy to clarify any remaining issues.
>
> --
>
> **References**
>
> [1] Schneider, Nora, et al. "Generative intervention models for causal perturbation modeling." ICML 2025.
>
>
> [2] Ahuja, Kartik, et al. "Interventional causal representation learning." ICML 2023.
>
>
> [3] Lippe, Phillip, et al. "Citris: Causal identifiability from temporal intervened sequences." ICML 2022.
>
>
> [4] Brehmer, Johann, et al. "Weakly supervised causal representation learning." NeurIPS 2022.

---

> ### Author Response · Authors · 2025-11-24
> **Updates on SimplerEnv Results**
>
> Thanks again for your thoughtful review and insightful comments. Below we include the latest results on `SimplerEnv`, which we just finished running.
>
> Since `SimplerEnv` provides a real-to-sim translation of real-world demos such `Bridge-v2`, it is primarily intended for imitation learning and VLA models. To meaningfully evaluate MIST in this setting, we adapted the setup to demonstrate:
>
> - Our framework can operate in these more complex manipulation environments,
> - MIST can also leverage task information other than reward here, task instructions to identify the compact but sufficient representations.
>
> **Setup**. We follow the `Bridge-v2` task definitions and consider four tasks: `Put Spoon on Towel`, `Put Carrot on Plate`, `Stack Green Block on Yellow Block`, and `Put Eggplant in Yellow Basket`. We train MIST on the `SimplerEnv` version of `Bridge-v2` using the same architecture as in the Meta-World experiments. To learn the latent dynamics, we encode pixels into latent states and train an RSSM on both policy-induced transitions (RT-1 provided by SimplerEnv, 50k) and actively explored transitions (50k) for each task.
>
> **Learning MIST**.
> Because these tasks lack dense rewards, we use task instructions as the task variable $g$. Each instruction is embedded with RoBERTa (768-dim). We then estimate mutual information between $g$ and masked latent states to obtain the MIST representation.
>
> **Evaluation protocol**.
> After training the world model, we follow the spirit of the SIMPLER paper: evaluate the gap between policy performance measured on the original simulator vs. our latent-space simulator, and compare against Dreamer-v3. **Conceptually, this forms the chain
> real → SIMPLER simulator → MIST latent simulator**, allowing us to assess how well MIST can serve as a compact world model in this benchmark.
>
> **Results**.
> We show the comparison between our latent simulator and Dreamer-v3 below. In the parentheses, we report the evaluation gap relative to SIMPLER. So, smaller values indicate a better latent simulator.
>
> | WidowX+Bridge Evaluation Setup      | Policy      | Put Spoon on Towel |                      | Put Carrot on Plate |                      | Stack Green Block on Yellow Block |                      | Put Eggplant in Yellow Basket |                      |
> |-------------------------------------|-------------|---------------------|----------------------|----------------------|----------------------|-----------------------------------|----------------------|-------------------------------|----------------------|
> |                                     |             | Grasp Spoon         | Success              | Grasp Carrot         | Success              | Grasp Green Block                 | Success              | Grasp Eggplant                | Success              |
> | **SIMPLER Eval (Visual Matching)**  | Octo-Small  | 0.778               | 0.472                | 0.278                | 0.097                | 0.403                             | 0.042                | 0.875                         | 0.569                |
> | **Latent Simulator (Dreamer-v3)**   | Octo-Small  | 0.052 (0.726)     | 0.000 (0.472)      | 0.060 (0.218)      | 0.000 (0.097)      | 0.040 (0.363)                   | 0.000 (0.042)      | 0.070 (0.805)               | 0.000 (0.569)      |
> | **Latent Simulator (Ours: MIST)**   | Octo-Small  | 0.525 (0.253)     | 0.240 (0.232)      | 0.420 (0.142)      | 0.182 (0.085)      | 0.285 (0.118)                   | 0.090 (0.048)      | 0.556 (0.319)               | 0.325 (0.244)      |
>
> As expected, there are still some gaps between ours and SIMPLER visual-matching due to the compression into latent space (there is always some kind of information loss due to the neural network approximation), while Dreamer-v3 largely collapses on these tasks (success rates near zero). Hence, our latent model preserves non-trivial task-level signal and demonstrates that MIST can learn meaningful latent dynamics from instruction-conditioned SimplerEnv demonstrations.

---

> > ### Author Response · Authors · 2025-11-26
> > **Thank you for your thoughtful review**
> >
> > Dear Reviewer d8kx,
> >
> > We again thank you for your constructive feedback and suggestions. We believe the rebuttal and revisions addressing W1 (together with the listed changes in the main text and appendices) substantially improve the clarity of the presentation, and that our response to W2 clarifies the algorithmic design and its practical complexity. For Q1–Q3, we hope the additional details on concepts, terminology, and efficiency metrics address your concerns. We have also added evaluations on SimplerEnv as you suggested. If you have any further questions or comments, we would be very happy to continue the discussion.
> >
> > The Authors

---

### Official Review · Reviewer_AWLT · 2025-11-01

**Soundness:** 2
**Presentation:** 2
**Contribution:** 3
**Rating:** 4
**Confidence:** 4

**Summary:**

The authors introduce MIST-WM, a new way to learn world models with latent representations which are minimal, but task-sufficient. This is achieved by a co-design of learning useful interventions on the side of the agent, and an adaptive curriculum on the side of the environment. For the agent side, it builds on Dreamer-v3, and adds a projection of its latent state with a filter of learned masks to get to the minimal yet sufficient factored representation. Interventions are performed with a skill library that is learned using DIAYN and METRA in a way that maximizes mutual information about selected individual values of the latent representation while minimizing it about the other values. The curriculum  optimizes the ordering of tasks and environment, aiming to minimize world-model learning error and maximize cumulative rewards across environments and tasks. Evaluation of various aspects is performed on DMControl, RoboSuite, and Meta-World. Results demonstrate the improved sample-efficiency on most tasks of the robotic manipulation and locomotion tasks tested, and improved generalisation compared to baselines. Different ablation studies are included as well.

**Strengths:**

* the problem of learning compact yet well-performing world models which generalize well to new situations is highly relevant and timely
* the authors provide concise definitions of task-sufficient and minimal representations, as well as loss functions which encourage learning them
* empirical results show an improved sample-efficiency for most of the tested benchmarks
* analysis reveals that the learned representations have higher correlations with ground truth variables that those of other baselines
* improved compositional and unseen generalization is notable

**Weaknesses:**

* The paper tries to pack too many things into one paper and as a result, sacrifices clarity and reproducibility in the process.
* More focus on clear explanations of the whole approach, including pseudo-code and details of all necessary intermediate steps would be helpful.
* As is, it is not clear how latent states of the Dreamer model are projected into the factored representation of MIST-WM.

**Questions:**

* Regarding the projection of the Dreamer latents to MIST-WM: how is the dimensionality $d_{i}$ of the $\mathbf{s}_{t}^{i}$ initialized? When is it expanded and by how much? Who decides on that, and is the right dimensionality per task assumed to be known?
* What do the ensemble member functions $f_{\theta}^{m}$ in Eq. (7) model exactly and how are they trained?
* Is there a specific reason you use PPO for some and SAC for other environments?
* The KL-divergence in Eq. (1) looks suspicious, since $s_{t}^{i}$
appears on both sides of the conditioning bar of $q_{\theta}$ and $p_{\theta}$. Is this a mistake?
* What do the error bars in figures 6 and 7 signify?
* Will source code be published to reproduce results?

Minor point:

* The claim that "humans do not predict future pixels or track redundant detail when planning" (l. 43 on p. 1) seems rather strong. Do you have any references to back this up? If not, maybe this claim is not really necessary and should be left out.

---

> ### Author Response · Authors · 2025-11-19
> **Author Response (1/3)**
>
> We would like to thank you for the thoughtful and constructive feedback. We really appreciate the detailed questions and suggestions on clarity, presentation, and technical details, and we are glad that you found the problem setting, the MIST-WM definitions, the overall framework, and the empirical results to be positive. Below, we respond to each point in the Weaknesses and Questions sections in turn and provide the corresponding clarifications and revisions.
>
> ----
>
>
> > **W1**: The paper tries to pack too many things into one paper and as a result, sacrifices clarity and reproducibility in the process.
>
>
> > **W2**: More focus on clear explanations of the whole approach, including pseudo-code and details of all necessary intermediate steps would be helpful.
>
> [**Response**] We answer these two weaknesses together, as they concern closely related issues of the modular design and related clarity.
>
> First, we respectfully argue that the components in our framework are not an attempt to “pack too many things” into a single paper, but rather arise naturally from the goal of learning task-specific minimal and sufficient world models. Achieving such representations inherently requires a synergy between (i) the agent’s **interaction strategy**, (ii) the **environment/task schedule**, and (iii) the **structure-aware world-model** learning objective. Each component plays a distinct and necessary role: the active interventions provide the informative trajectory data needed to identify minimal sufficient factors; the adaptive curriculum ensures that tasks expose these factors in a progressively learnable order; and the world-model objective extracts the minimal sufficient subspace. These three parts form a coherent closed loop rather than independent modules being combined.
>
>
> As for reproducibility, our implementation follows a modular design and is constructed from standard objectives, so the overall recipe is not too complex. We provide an anonymized code repository  [here](https://anonymous.4open.science/r/mist-wm-C82C), along with more detailed technical breakdowns in the revised paper: more details on each component, explicit algorithmic steps for each module, more architecture and hyperparameter tables, and environment/task specifications. Together, these clarifications are intended to make it straightforward to use or extend our method.
>
> In summary, the guiding principle is to learn task-specific minimal and sufficient world models; the concrete implementation of this principle naturally decomposes into several interacting modules (active interventions, structured world-model learning and curriculum). These are not independent add-ons “packed” into the paper, but rather a coherent recipe whose components are each required to realize this principle in practice.
>
> [**Related Revisions**] To improve readability and reproducibility, we have made several targeted revisions throughout the paper. Here we summarize the main updates (more detailed clarifications are provided in our responses to later questions):
>
> (1). **Clearer algorithmic explanations**: We expanded the descriptions of all three modules: active interventions, structure learning, and curriculum, in `§3.1.1, 3.1.2, and 3.2` (`Pages 4-6`). Newly added/organized text (highlighted in green) provides more explicit objective definitions, training signals, and intermediate computation steps and links to the approximated objectives in the appendix. These edits also directly address several technical questions raised later in your other comments.
>
> (2). **Revised Fig 1 for conceptual clarity**: We updated `Fig. 1` (`Page 2`) to better illustrate how the components form a coherent recipe rather than independent modules. The revised figure visually conveys the data flow and how the agent, curriculum, and structured world-model learning interact.
>
> (3). **New `Algorithm 1` in the Appendix (`Page 25`)** We added a high-level pseudocode description, showing how the modules connect, the order of operations, and how gradients flow through the system.
>
> (4). **Additional supporting details in the Appendix** We added: (i) `A.2` (`Page 16-17`): explanations of key concepts (unstructured latents, structure learning, MIST states, intervention skills, and segment separability), which are included in the main paper, but we provide more explanations. (ii) `B.4` (`Page 19`): details on expandable latents and how the dimensionality grows across tasks;
> (iii) `B.6` (`Page 19-20`): a summary of all objectives, indicating which components receive gradients and which parameters are frozen during each stage of training (shown also by `Table A.2`, `Page 20`).
>
> ---

---

> ### Author Response · Authors · 2025-11-19
> **Author Response (2/3)**
>
> > **W3**: As is, it is not clear how latent states of the Dreamer model are projected into the factored representation of MIST-WM.
>
> [**Response**] We clarify that the ``projection’’ corresponds to the *structure learning* of our method.
> At a high level, Dreamer first produces an unstructured latent state $s_t \in \mathbb{R}^d$. MIST-WM then learns a task-specific factored representation by applying a soft mask $m^i \in (0,1)^d$ to select
> the relevant subspace for task $T_i$, yielding the MIST states $m^i \odot s^i_t$.
>
> The mask is learned through the optimization of objectives, including the sufficiency and minimality scores defined
> in `Eq. (2)-(3)` (`Sec 3.1.1`), which encourages sufficiency for reward prediction and minimality with respect to the task.
>
> Both terms provide gradients to the mask $m^i$. To enable differentiability, we implement $m^i$ as a soft, gated following the continuous relaxation in `Appendix B.1` (`Page 17`). For the minimality loss $\mathcal{R}^{(i)}_{\text{min}}$, estimating the mutual information term requires a differentiable estimator, for which we use MINE (details also provided in `Appendix B.1`; `Page 17`). The resulting gradients update only the mask parameters, while the Dreamer encoder remains frozen during structure learning to ensure that projection acts purely on the learned latent space.
>
> [**Related Revisions**] We have revised `§3.1.1` (`Pages 4-5`) to make this projection process explicit and added clear pointers to `Appendix B.1`, where we provide the full implementation details (soft-gated mask, MINE-based MI estimator).
>
> ---
> > **Q1**: Regarding the projection of the Dreamer latents to MIST-WM: how is the dimensionality $d_i$ of the $s^i_t$ initialized? When is it expanded and by how much? Who decides on that, and is the right dimensionality per task assumed to be known?
>
> [**Response**] We do *not* assume that the ``right'' latent dimensionality per task is known to the agent. Instead, we follow a simple over-parameterized heuristic that is consistent with practice and then rely on structure learning to select the task-relevant subspace.
>
> Concretely, for each benchmark, we initialize the Dreamer latent dimensionality $d_1$ to a fixed value and keep it shared across tasks at the beginning (e.g., $d_1 = 512$ for Meta-World, RoboSuite, and Kitchen, which have high-dimensional visual observations, and $d_1 = 256$ for DMControl). When moving from task $T_i$ to $T_{i+1}$, we allow the latent dimensionality to grow by a fixed increment, $d_{i+1} = d_i + \Delta d$.
> where we set $\Delta d = 32$ in all experiments. The choice of $d_1$ and $\Delta d$ is therefore a heuristic design decision rather than an oracle choice of the ``correct'' dimension. In practice, this slight over-parameterization is benign because our structure-learning module (through the sufficiency and minimality objectives and the learned mask $m^i$) identifies and uses only the task-relevant subset of dimensions (the MIST states), while irrelevant coordinates are effectively suppressed.
>
>
> [**Related Response**] We have added this explanation to `§3.1.1` (`Lines 232-236`) and provided further implementation details in `Appendix B.4` (`Lines 989-1006, Page 15`).
>
> ---
>
> > **Q2**: The ensemble member functions $f_{\theta}^m$ in Eq.~(7) are one–step latent
> transition predictors that model the RSSM dynamics used in Dreamer.
>
> [**Response**] Following prior work on ensemble-based uncertainty estimation for world models [1-3], we train an
> ensemble of $5$ such models, each parameterized independently. Each
> $f_{\theta}^m$ takes as input the current latent state and action $(s_t, a_t)$ and predicts the next latent state.
>
>
> Training is performed jointly with the world model: for every latent transition $s_{t+1} \sim p_\theta(s_{t+1} \mid s_t, a_t)$ generated by the Dreamer RSSM, we optimize each ensemble member with a standard one–step prediction loss. Thus, the ensemble is trained on the same latent transitions as the main world model, and the disagreement term of the dynamics part in `Eq. (7)` is computed as the variance across the ensemble predictions $\{ f_{\theta}^m(s_t, a_t)\}_{m=1}^5$.
>
> [**Related Revision**] We have added these details to Appendix `B.3` (highlighted in green, `Lines 983-987`, `Page 19`).
>
> ---
> **References**
>
> [1] Ball, Philip, et al. "Ready policy one: World building through active learning." ICML 2020.
>
>
> [2] Sekar, Ramanan, et al. "Planning to explore via self-supervised world models." ICML 2020.
>
>
> [3] Mendonca, Russell, et al. "Discovering and achieving goals via world models." NeurIPS 2021.
>
>
> ---

---

> ### Author Response · Authors · 2025-11-19
> **Author Response (3/3)**
>
> > **Q3**: Is there a specific reason you use PPO for some and SAC for other environments?
>
> [**Response**] There is no methodological dependence on a particular RL algorithm in our framework. We use PPO for DMControl and Kitchen, and SAC for Meta-World and RoboSuite primarily for empirical convenience: in these settings, the respective algorithms are widely used and known to be stable. Importantly, our contributions do not modify the policy learning components of either PPO or SAC; the choice of RL algorithm is orthogonal to MIST-WM. For all baselines within each environment, we use the same policy architecture and the same RL algorithm to ensure fair comparison. To support this, we also include the full hyperparameter configurations for both PPO and SAC in `Appendix Tables A6 and A7`.
>
> [**Related Revisions**] We added hyperparamerters in Appendix `Table A6 and A7` (`Page 24` & `Page 26`).
>
>
> ---
>
>
> > **Q4**: The issue in Eq. (1)
>
> [**Response**] Thank you for catching this! We apologize for the confusion. It is a typo and intended to be the standard Dreamer world-model objective. We have corrected the equation in the revised version.
>
> [**Related Revision**] `Eq. (1)`
>
> ---
>
> > **Q5**: What do the error bars in figures 6 and 7 signify?
>
> [**Response**] Error bars indicate the standard deviation across 5 random seeds. We have clarified this in the captions of Figures 6 and 7 (and also the figures in the appendix).
>
> [**Related Revision**] `Fig. 6-7`, `Appendix Fig. A4`, `Appendix Fig. A7`
>
> ---
>
> > **Q6**: Will source code be published to reproduce results?
>
> [**Response**] Yes, we will open-source the code upon publication. In the meantime, please find the anonymized code repository [here](https://anonymous.4open.science/r/mist-wm-C82C).
>
> ---
>
>
> > **Q7**: The claim that "humans do not predict future pixels or track redundant detail when planning" (l. 43 on p. 1) seems rather strong. Do you have any references to back this up? If not, maybe this claim is not really necessary and should be left out.
>
> [**Response**] Thank you for pointing this out. We have revised the statement to a more precise and better-supported version: `By contrast, humans typically do not plan using pixel-level predictions or by tracking redundant perceptual details; instead, our planning relies on compact task-relevant representations (Mastrogiuseppe & Ostojic, 2018; Ho et al., 2022; Rajalingham et al., 2022; Nayebi et al., 2023).`
>
> Works [4,6,7] collectively show that both human and non-human primate neural activity, as well as computational RNN models trained on dynamic tasks, exhibit low-dimensional latent dynamics that capture task-relevant variables rather than high-dimensional visual details. Ho et al. (2022) [5], in particular, demonstrate that human planning behavior depends on simplified internal representations that are usually without irrelevant perceptual information. We believe this statement aligns well with existing evidence while still motivating our focus on task-relevant latent representations.
>
>
> [**Related Revision**] `Sec. 1` (`Lines 41-43`)
>
> ---
> **References**
>
> [4] Francesca Mastrogiuseppe and Srdjan Ostojic. Linking connectivity, dynamics, and computations in
> low-rank recurrent neural networks. Neuron, 99(3):609–623, 2018.
>
>
>
> [5] Mark K Ho, David Abel, Carlos G Correa, Michael L Littman, Jonathan D Cohen, and Thomas L
> Griffiths. People construct simplified mental representations to plan. Nature, 606(7912):129–136, 2022.
>
>
> [6] Rishi Rajalingham, Aida Piccato, and Mehrdad Jazayeri. Recurrent neural networks with explicit
> representation of dynamic latent variables can mimic behavioral patterns in a physical inference
> task. Nature Communications, 13(1):5865, 2022.
>
>
> [7] Aran Nayebi, Rishi Rajalingham, Mehrdad Jazayeri, and Guangyu Robert Yang. Neural foundations of mental simulation: Future prediction of latent representations on dynamic scenes. Advances in Neural Information Processing Systems, 36:70548–70561, 2023.
>
>
>
>
> ---
>
>
> We really appreciate all of your comments and are glad we had the chance to clarify and strengthen the paper. We hope the revisions address your concerns: for **W1–W2**, we clarify the overall structure and improve explanations; for **W3**, we provide a clearer description of how Dreamer latents map to MIST states; for **Q1–Q3** and **Q6**, we add detailed methodology, hyperparameters, and training pathways; for **Q4–Q5**, the typos and missing details are now fixed; and for **Q7**, we provide supporting citations and a more careful formulation. If any additional questions arise, we would be happy to clarify further.

---

> > ### Author Response · Authors · 2025-11-26
> > **Thank you for your thoughtful review**
> >
> > Dear Reviewer AWLT,
> >
> > Thank you again for your insightful comments and the time you invested in our paper. We have made clarifications and revisions accordingly. For W1–W2, we clarified the overall structure and improved the high-level explanations; for W3, we now more clearly describe how Dreamer latents are projected to MIST states; for Q1–Q3 and Q6, we added detailed methodology, hyperparameters, and training pathways; for Q4–Q5, the typos and missing details have been corrected; and for Q7, we provide supporting citations and a more carefully formulated statement. Should there be any leftover questions, please feel free to let us know and we will make every effort to address them during the subsequent discussion period.
> >
> > The Authors

---

### Author Response · Authors · 2025-11-26
**General Response (1/2)**

Thank you again for all your thoughtful comments and constructive suggestions. Those are very helpful for improving the paper. We are encouraged by your recognition of our framework and goals as a “novel paradigm” (`d8kx`), with an “ambitious scope and goal” (`qDyg`), and as addressing a problem that is “highly relevant and timely” (`AWLT`). We also appreciate your positive feedback on our “concise definitions” (`AWLT`), the “novel formulation of the MIST states” (`HNgN`), as well as the acknowledgement of our empirical results and evaluations from all of you.

Below, we summarize the main common concerns raised in the reviews and the corresponding changes in our revision.

---

1. **About the overall goal (focus, “too many components”, and novelty).**

Our goal is to learn **task-specific minimal and sufficient world models**. Achieving such representations naturally requires a synergy between (i) the agent’s **interaction strategy**, (ii) the **environment / task schedule**, and (iii) the **structure-aware world-model objective**. Each component plays a distinct and necessary role: active interventions provide informative trajectories that reveal task-relevant factors; the adaptive curriculum ensures that tasks expose these factors in a progressively learnable order; and the structure-learning objective extracts a minimal, task-sufficient subspace from the Dreamer latents. These three parts form a coherent closed loop rather than a collection of independent modules.

In short, the guiding principle is fixed: task-specific minimal and sufficient world models; and the implementation naturally decomposes into the three interacting modules above (intervention, structure learning, curriculum). They are not “packed in” arbitrarily, but are each required to realize this principle in practice. Empirically, we also quantify the individual impact of each module through careful ablations.

**Related Responses to questions**: **W1-W2** from `AWLT`; **W1-W2** from `qDyg`; **W2** from `HNgN`

**Revisions**: We updated `Fig. 1` (Page 2) to better illustrate how the components form a coherent recipe rather than independent modules. The revised figure visually conveys the data flow and how the agent, curriculum, and structured world-model learning interact.

---

2. **About clarity of concepts and algorithms**

We have revised the manuscript to clarify the overall pipeline and how the components interact. Specifically:

- Clearer algorithmic explanations. We expanded the descriptions of the main modules—projection, active interventions, structure learning, and curriculum, in `§3.1.1`, `§3.1.2`, and `§3.2` (`pp. 4–6`). The newly added and reorganized text (highlighted in green) provides more explicit objective definitions, training signals, and intermediate computation steps, together with links to their approximations in the appendix.

- In `Sec. 2`, we now explicitly walk through the concrete example in `Fig. 2(a)` after the formal definitions, highlighting which nodes are MIST states in the graph and why.

- New `Algorithm 1` in the appendix. We added a high-level pseudocode description (`Algorithm 1`, `p. 25`) that shows how the modules connect, the order of operations, and how gradients flow through the system.


- Additional supporting details in the appendix.

  - `A.2` (`pp. 16–17`): explanations of key concepts (unstructured latents, structure learning, MIST states, intervention skills, and segment separability), which complement the formal definitions in the main paper.


  - `B.4` (`p. 19`): details on the expandable latent space and how the dimensionality grows across tasks.


  - `B.6` (`pp. 19–20`): a summary of all objectives, indicating which components receive gradients and which parameters are frozen at each stage of training (also summarized in `Table A.2`, `p. 20`).


All of these changes are highlighted in the revised manuscript and referenced at the relevant points in the main text (for those in appendix).

---

---

> ### Author Response · Authors · 2025-11-26
> **General Response (2/2)**
>
> 3. **Empirical Complexity and Experimental Details**
>
> - **Complexity**: The framework is modular rather than intrinsically complicated. Each stage is trained with a relatively simple objective while the remaining components are kept frozen. We make this explicit in the revised `Algorithm 1` and in `Appendix B.6` (`Table A.2`).
>
> - **More hyperparameters and implementation details**: PPO/SAC (**S2** from `HNgN`; `Table A.6-A.7` (`pp. 24; 26`).); Anonymized code repository  [here](https://anonymous.4open.science/r/mist-wm-C82C); Skill composition (See responses to **W4-2/Q4** from `qDyg`; revised in `Sec. 5` (`L483-485`, `p. 9`)); Projection of MIST and how to increase dimensionality across tasks? (See responses to **W3-1/Q1** from `qDyg`; **W3** from `AWLT`; Revision `§3.1.1` (`pp. 4-5`), `Appendix B.1/B.4`). For the full suite of hyperparameters, see `Table A.2` for the gradient flow w.r.t. the objective functions, `Table A4-A5` for intervention and MIST learning, `Table A6-A7` for policy leanring, and `Table A8` for curriculum learning.
>
> ---
>
> 4. **More Evaluation**
>
> We additionally include ablations using DIAYN (see our response to **W4-5** from `HNgN`) and evaluations on the real-to-sim robotic manipulation benchmark SimplerEnv (see our response to `d8kx`).
>
> ---
>
> For the other comments, we provide point-by-point responses and have incorporated the corresponding edits into the revised manuscript. Thank you again for your time and thoughtful feedback. We hope these clarifications and revisions address your concerns, and we are happy to further clarify any remaining questions during the discussion period.

---

### Meta-Review · Area_Chair_iymB · 2025-12-28

**Summary:**

### Summary

The paper proposes an iterative framework that alternates between (1) learning compact yet task-sufficient state representations for world modeling and control, and (2) selecting the next task/environment to maximize information gain, thereby co-designing interventions and a curriculum. The approach is an integration of several components: world-model learning, compact latent representation extraction, exploratory policy learning (via unsupervised skill discovery), and task/environment curriculum design.

### Reviewers' concerns

* **Clarity and diffused contributions (qDyg, AWLT)**: The paper is difficult to follow given the number of moving parts. The writing and figures do not clearly communicate the high-level concept and the roles of each component. More importantly, the manuscript places emphasis on many components, making it hard to identify the primary novelty; the "main contribution" feels spread across multiple submodules rather than being stated crisply and defended throughout. This creates uncertainty about what is new versus what is assembled from existing techniques, and the presentation does not yet meet the expected ICLR standard for accessibility given the method’s complexity.

* **Marginal curriculum gains vs added complexity (HNgN)**: The empirical benefit attributed to the proposed curriculum appears small relative to the additional complexity introduced by curriculum selection and its supporting machinery. Given the integrated nature of the approach, stronger evidence and cleaner isolations are needed to justify the curriculum mechanism as a meaningful improvement rather than an incremental add-on.

* **Incomplete related work coverage (HNgN)**: The related work is missing.

* **Limited experimental scope (d8kx)**: Experiments are conducted on a small set of relatively simple tasks. This limits confidence in the central claim that curriculum co-design yields robust benefits in broader settings. Supporting the curriculum claim would require evaluation on larger-scale and more diverse benchmarks (e.g., SIMPLER, LIBERO-style suites) or real-world experiments, or at least a clearer limitations discussion acknowledging that scalability/generalization is not established.

* **Reproducibility and heuristic design choices (qDyg, AWLT)**: Despite the method’s complexity, implementation details and key design choices are not specified clearly enough, and several choices appear heuristic without rigorous verification (e.g., sensitivity analyses or principled ablations). This raises concerns about whether the results can be reproduced reliably and which components are truly necessary.

**Reviewer Concerns:**

* **Clarity and diffused contributions (qDyg, AWLT)**: The revised manuscript remains difficult to follow, and the overall presentation (including figures) still does not meet the ICLR standard given the method’s complexity. The core contribution is still not stated crisply and appears spread across several components.

* **Marginal curriculum gains vs added complexity (HNgN)**: The rebuttal does not convincingly address this concern.

* **Incomplete related work coverage (HNgN)**: Although the related work section was included in the updated manuscript, it still does not sufficiently cover prior work combining exploration or unsupervised skill learning with model-based RL, or representation learning for model-based RL.

* **Limited experimental scope (d8kx)**: The added results on SIMPLER are a step forward, but the evaluation remains limited to a single task level, and does not show the strength of the full pipeline of the proposed approach, which is on the multi-task curriculum learning setup.

* **Reproducibility and heuristic design choices (qDyg, AWLT)**: The revision and rebuttal include many missing implementation details. But, this is still not sufficient to understand and reproduce the proposed method end-to-end.

**Reviewer Scores:**

Given the unclear contributions and missing technical details, I expect the reviewers would likely maintain their scores and lean toward rejection.

---

### Decision · Program_Chairs · 2026-01-26

Reject